# Parenteral BCG vaccine induces lung-resident memory macrophages and trained immunity via the gut–lung axis

Mangalakumari Jeyanathan[1], Maryam Vaseghi-Shanjani[1], Sam Afkhami[1], Jensine A. Grondin[2], Alisha Kang[1], Michael R. D'Agostino[1], Yushi Yao[1,3], Shreya Jain[1], Anna Zganiacz[1], Zachary Kroezen[4], Meera Shanmuganathan[4], Ramandeep Singh[1], Anna Dvorkin-Gheva[1], Philip Britz-McKibbin[4], Waliul I. Khan[2] & Zhou Xing [1]✉

Aside from centrally induced trained immunity in the bone marrow (BM) and peripheral blood by parenteral vaccination or infection, evidence indicates that mucosal-resident innate immune memory can develop via a local inflammatory pathway following mucosal exposure. However, whether mucosal-resident innate memory results from integrating distally generated immunological signals following parenteral vaccination/infection is unclear. Here we show that subcutaneous Bacillus Calmette–Guérin (BCG) vaccination can induce memory alveolar macrophages (AMs) and trained immunity in the lung. Although parenteral BCG vaccination trains BM progenitors and circulating monocytes, induction of memory AMs is independent of circulating monocytes. Rather, parenteral BCG vaccination, via mycobacterial dissemination, causes a time-dependent alteration in the intestinal microbiome, barrier function and microbial metabolites, and subsequent changes in circulating and lung metabolites, leading to the induction of memory macrophages and trained immunity in the lung. These data identify an intestinal microbiota-mediated pathway for innate immune memory development at distal mucosal tissues and have implications for the development of next-generation vaccine strategies against respiratory pathogens.

There is growing recognition of the importance of innate immune memory and trained innate immunity (TII) in host defense and vaccinology[1–3]. Epidemiological studies have shown that parenteral immunization with live attenuated vaccines (including Bacillus Calmette–Guérin (BCG)) offers protection against both the target and unrelated pathogens[4]. Thus, antituberculosis (TB) BCG vaccination reduces all-cause mortality and/or respiratory infections in young children and the elderly[5–7]. Such systemic/parenteral microbial-, inflammation- or vaccine-induced TII is mediated primarily through centrally trained circulating leukocytes including monocytes resulting from metabolic/epigenetic rewiring of myeloid progenitors in the bone marrow (BM)[8–13].

[1]McMaster Immunology Research Centre, M. G. DeGroote Institute for Infectious Disease Research and Department of Medicine, McMaster University, Hamilton, Ontario, Canada. [2]Farncombe Family Digestive Health Research Institute and Department of Pathology and Molecular Medicine, McMaster University, Hamilton, Ontario, Canada. [3]Department of Immunology, Zhejiang University, Zhejiang, China. [4]Department of Chemistry and Chemical Biology, McMaster University, Hamilton, Ontario, Canada. ✉e-mail: xingz@mcmaster.ca

Until recently, little was known about whether mucosal-resident macrophages can directly be trained to store lasting innate memory[3]. We and others have discovered that respiratory mucosal exposure to microbes/vaccines can induce airway memory macrophages with TII[14,15] or immune-regulatory or tolerized property[16–18]. Respiratory adenoviral-vectored vaccination/infection induced a persisting memory phenotype in resident alveolar macrophages (AMs), independent of circulating monocytes. TII associated with such memory macrophages enhances innate protection against both the intended target and heterologous bacterial pathogens in the lung[14,19].

Hence, growing evidence supports a paradigm of compartmentalization in the genesis of resting-state innate immune memory resulting from the recent history of immunological imprinting/training. This paradigm ascribes trained hematopoietic progenitors and circulating monocytes to systemic/parenteral microbial exposure/vaccination, while it attributes the memory phenotype of barrier mucosal-resident macrophages to the local microbial exposure/vaccination[3,20,21]. The latter is in line with the current concept of macrophage niche of tissue residence and its adaptation to local inflammation[22]. It has remained unclear whether, in the absence of local inflammation, innate memory at barrier tissues may develop as a way of integrating and adapting to distally generated immunological signals following systemic/parenteral microbial exposure/vaccination[23]. Recent studies show that subcutaneous viral infection/vaccination triggers a widespread immunological alert across multiple tissue sites[24,25] and that local tissue injury can activate resident macrophages in remote tissue sites[26]. One prototypic modality of the immunological cross-talk between tissue sites is the gut–lung axis whereby intestinal microbiota dysbiosis alters immune responses in the lung[27–34]. However, it is not well understood whether parenteral vaccination affects the intestinal microbiome and whether/how intestinal dysbiosis induces lung-resident innate memory[23]. As BCG is administered via the skin, like most current human vaccines (including coronavirus disease 2019 (COVID-19) vaccines), addressing these questions has far-reaching implications.

Here using an experimental model, we investigated whether and how subcutaneous BCG vaccination induces tissue-resident memory macrophages and TII in the lungs. Aside from its effect on myeloid progenitors, parenteral BCG independently induces memory AM and TII against *Mycobacterium tuberculosis* infection. This process occurs via the initial mycobacterial spread and a gut–lung axis involving a time-dependent alteration in intestinal microbiota, barrier function and metabolites. Our study thus identifies an intestinal microbiota-mediated pathway to innate memory development at distal mucosal sites and has implications for the development of next-generation vaccines against respiratory pathogens[35,36].

## Results

### Subcutaneous BCG induces a memory phenotype in lung macrophages

Subcutaneous BCG vaccination induces trained monocytes in the BM/blood[9,13]. It is unclear whether it trains lung-resident AM. Thus, mice were vaccinated subcutaneously with BCG. At 5 weeks postvaccination, the bronchoalveolar lavage (BAL) and lung mononuclear cells were analyzed with (S) or without (US) stimulation by *M. tuberculosis* whole cell lysates (WCL) (Fig. 1a). AMs were identified as Ly6G⁻CD11b⁻CD11c^hiCD64^hiSiglec-F^hi myeloid cells[14]. Approximately, 95% of airway cells of BCG-vaccinated hosts were AMs. Compared to control AMs (PBS-US), BCG AMs (BCG-US) expressed higher levels of baseline and augmented MHC II upon stimulation (PBS-S versus BCG-S) (Fig. 1b). The stimulated AMs also exhibited significantly increased toll-like receptor 2 (TLR2) (Fig. 1c). Significantly higher frequencies of stimulated BCG AM produced IL-6, while tumor necrosis factor (TNF)-producing cells were comparable (Fig. 1d,e). Similar immune profiles were observed among lung tissue AM populations from control and BCG-vaccinated hosts (Extended Data Fig. 1a–c).

Eleven-analyte Luminex analysis of secreted cytokines/chemokines shows that at baseline (US), levels of cytokines/chemokines were low in control and BCG AM cultures (Fig. 1f–l), whereas upon stimulation, compared to control AMs, BCG AMs produced significantly higher levels of IL-6, IL-12p40, MCP-1, MIP-1α and RANTES (Fig. 1g,h,j–l) with no difference observed in TNF production (Fig. 1i).

We next analyzed the metabolic state of airway AM. BCG AM demonstrated significantly increased glycolysis compared to their mildly changed rates of oxidative phosphorylation (Fig. 1m,n and Extended Data Fig. 1d). Because local viral infection-trained AM underwent low-rate in situ proliferation for maintenance[14], the proliferating capability of BCG AM was examined by in vivo BrdU incorporation (Extended Data Fig. 1e). BCG AM showed significantly increased BrdU incorporation over control AM (Extended Data Fig. 1f), and their increased MHC II was independent of their proliferating status because both BrdU⁺ and BrdU⁻ AMs expressed increased MHC II (Extended Data Fig. 1g).

We next examined the immunophenotype and metabolic state of airway AM at an earlier 2-week time point post-BCG (Extended Data Fig. 1h). In contrast to their trained phenotype acquired at 5 weeks, the phenotype of airway and lung tissue AM of BCG hosts was comparable to the controls (Extended Data Fig. 1i–k). The 2-week BCG AM also demonstrated significantly reduced glycolysis compared to their controls (Extended Data Fig. 1l). To assess the mechanisms for such time-dependent training of airway AM, we first examined its relationship to mycobacterial dissemination post-BCG. We observed a small extent of BCG dissemination to the mediastinal lymph nodes (MLN) (40 ± 20 colony forming unit (CFU)/MLN) but not to the lung at 2 weeks. Because BCG possesses a slow-doubling time, this finding thus implicated the time-dependent mycobacterial dissemination in memory AM induction. We next addressed the role of BCG viability/replication and dissemination in AM training by comparing subcutaneous injection of viable BCG with heat-inactivated BCG (BCG-ia). Contrary to trained AM in viable BCG-vaccinated hosts, BCG-ia AM exhibited a profile in MHC II and IL-6 expression similar to untrained AM (Extended Data Fig. 1m).

We also determined whether, besides the lung, BCG vaccination had a global effect on macrophages in the peritoneal cavity. The peritoneal macrophages (PM) were identified as CD11b⁺F4/80^hiSiglecF⁻ population[37]. BCG PM demonstrated a significantly altered immune phenotype with reduced F4/80 expression (CD11b⁺F4/80^LowSiglecF⁻) compared to PBS control (Fig. 1o) and a trained phenotype with constitutively increased MHC II expression without restimulation (US) and increased IL-6/TNF production upon ex vivo restimulation (S) (Fig. 1p). On the contrary, consistent with its failure to train AM (Extended Data Fig. 1m), BCG-ia also failed to train PM (Fig. 1p).

The above data indicate that parenteral BCG leads to a time-dependent induction of memory AM. Such memory AM is characterized by increased MHC II and TLR2 expression, glycolysis and cytokine production upon stimulation. BCG vaccination also globally trains macrophages in the peritoneal cavity.

### Distinct gene profile and microbial control by memory lung macrophages

We next examined the transcriptional profile of memory AM in BCG hosts. Airway BAL BCG or control cells were cultured with (S) or without (US) stimulation, and isolated RNA was sequenced (Fig. 2a). The principal component analysis (PCA) shows that each of the groups clustered into its own pattern, indicating a unique gene expression profile in each (Fig. 2b and Extended Data Fig. 2a). A total of 248 genes were differentially expressed (DE) between BCG and control AM populations at baseline and after WCL restimulation (Extended Data Fig. 2b,c). BCG AMs with and without stimulation were enriched in gene sets involved in cell cycle and division (Fig. 2c and Extended Data Fig. 2d). For instance, *Mki67* (*Ki67*), a cellular proliferation marker, *Kif11, Kif15, Kif23* encoding kinesin-like proteins involved in chromosome segregation and spindle formation and *Rad51, Rad54b* involved in DNA

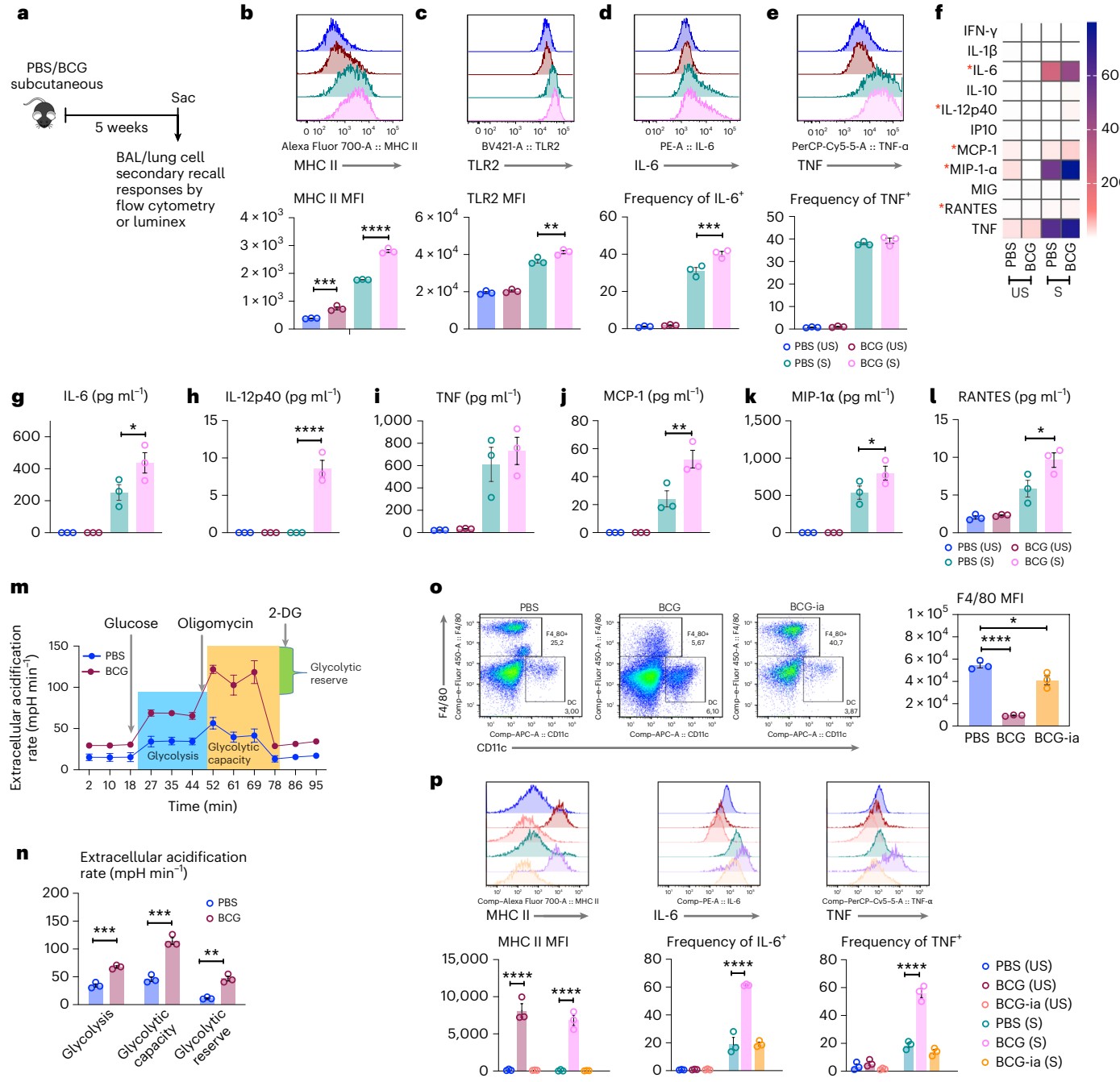

**Fig. 1 | Subcutaneous BCG vaccine induces memory AM in distal respiratory mucosa. a**, Experimental schema. **b,c**, Histograms of surface expression of MHC II (**\*\*P = 0.0027, \*\*\*\*P < 0.0001**) (**b**) or TLR2 (**\*\*P = 0.0050**) (**c**) on airway-resident AM. **d,e**, Histograms and frequencies of IL-6- (**\*\*\*P = 0.0007**) (**d**) or TNF-producing airway AM (**e**). **f**, Heatmap of cytokine/chemokine protein levels (geometric means) in supernatants of airway AM cultured with (S) or without (US) stimulation. Red asterisks denote significantly increased cytokine/chemokine production upon stimulation by airway AM of BCG hosts. **g–l**, Concentrations of IL-6 (**\*P = 0.0230**) (**g**), IL-12p40 (**\*\*\*\*P < 0.0001**) (**h**), TNF (**i**) and MCP-1 (**\*\*P = 0.0031**) (**j**), MIP-1α (**\*P = 0.0408**) (**k**) and RANTES (**\*P = 0.0137**) (**l**) in supernatants of airway AM cultured with and without stimulation. **m**, Real-time ECAR in airway AM at 8 weeks post-BCG immunization. 2-DG: 2-deoxy-glucose. **n**, Glycolysis (**\*\*\*P = 0.0006**), glycolytic capacity (**\*\*\*P = 0.0007**) and glycolytic reserve (**\*\*P = 0.0013**) in airway AM at 8 weeks post-BCG immunization. **o**, Dotplots of frequencies of cells gated out of total live peritoneal cells expressing F4/80 surface marker and the median fluorescent intensity (MFI) of F4/80 expression by PM at 5 weeks post-BCG vaccination (**\*P = 0.0231, \*\*\*\*P = <0001**). **p**, Histograms of and the MFI of MHC II and frequencies of IL-6- or TNF-producing PM at 5 weeks postvaccination with viable BCG or inactivated BCG (BCG-ia) or PBS with (S) or without (US) stimulation (**\*\*\*\*P = <0.0001**). Data in **b, d** and **e** are representative of three independent experiments. Data in **f, g–l** and **o–p** are representative of two independent experiments. Data in bar graphs are presented as mean ± s.e.m. and represent individual data points of biologically independent samples, *n* = 3 mice per group. One-way ANOVA was used for multiple comparison testing with Bonferroni's test for data in **b–l**, two-tailed *t*-test for data in **n** and one-way ANOVA followed by Dunnett's multiple comparison test for data in **o** and **p**.

repair, and the genes involved in sister chromatin segregations were upregulated in BCG AM. BCG AM also upregulated defense response genes, particularly those involved in chemotaxis of T cells (*Cxcl10, Ccl5*), monocytes (*Ccl7*) and downregulated cell activation genes *Lck, Bcl2, Il7r* and *Slamf7* (Fig. 2d). Furthermore, the predefined gene sets related to TII[1] and antigen (Ag) processing and presentation[14], glycolysis, mTOR

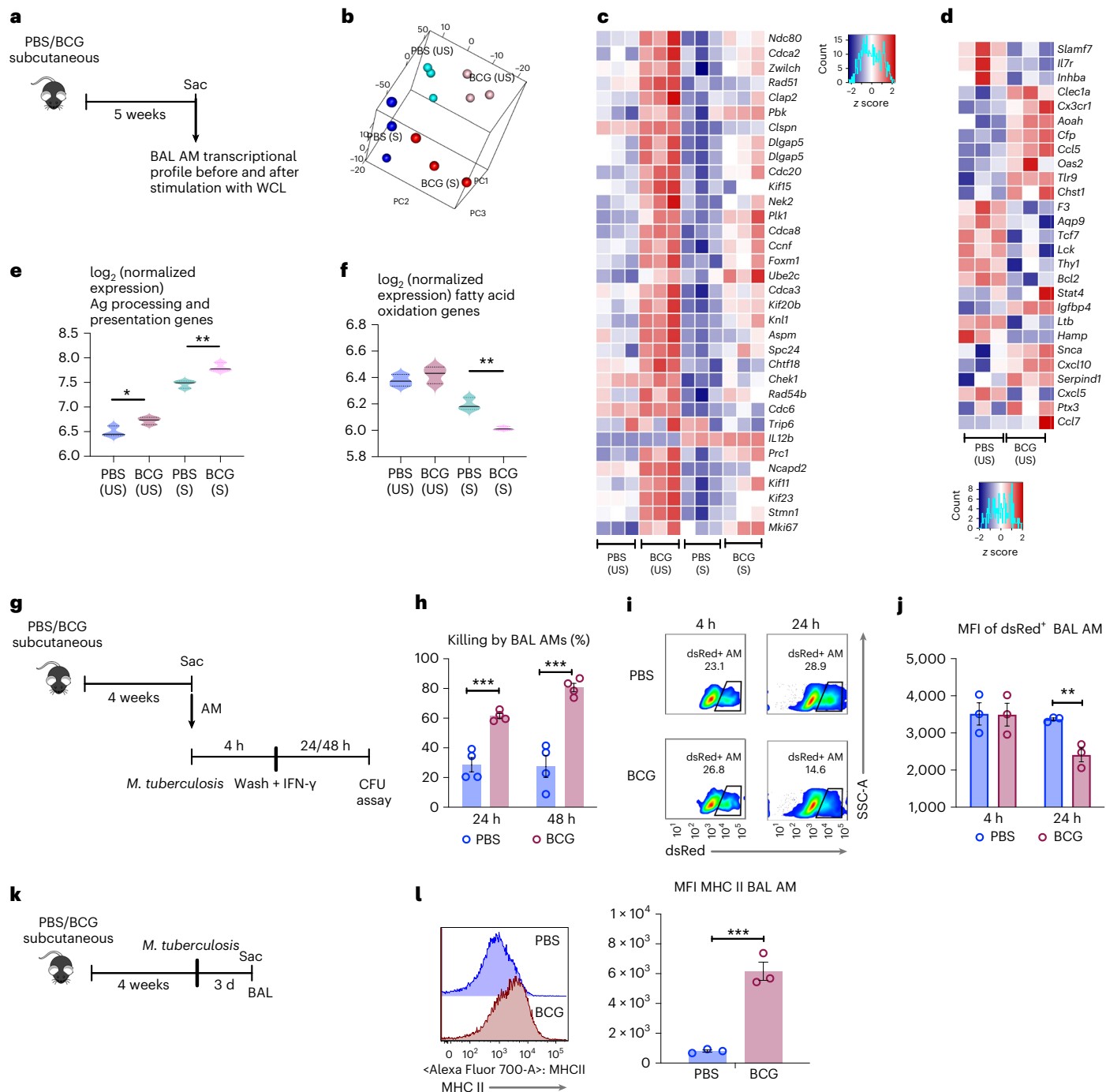

**Fig. 2 | BCG vaccine-induced memory AM transcriptional and antimicrobial responses. a**, Experimental schema. **b**, PCA of gene expression in airway AM. $n = 3$ mice per group. **c**, Heatmap of DEG involved in cell cycle in airway AM from PBS and BCG hosts with (S) and without (US) stimulation. **d**, Heatmap of DEG related to immune processes in airway AM comparing PBS and BCG groups with (S) and without (US) stimulation. **e,f**, Signature scores of genes involved in Ag processing and presentation (*$P = 0.0260$; **$P = 0.0054$) (**e**) and fatty acid oxidation (**$P = 0.0076$) (**f**) in airway AM. Horizontal lines in violin plots denote medians and dotted lines denote lower and upper quartiles. **g**, Experimental schema of ex vivo phagocytosis/killing of *M. tuberculosis* by airway AM. **h**, Percent of killing of phagocytosed *M. tuberculosis* bacilli by airway AM assessed at 24 (***$P = 0.0005$) and 48 h (***$P = 0.0006$). Each point represents biologically independent

samples ($n = 4$ mice in PBS group; $n = 3$ mice in BCG group). Representative of two independent experiments. **i**, Representative flow cytometric plots of frequencies of dsRed+ airway AM at 4 h (phagocytosis) and 24 h (killing) postinfection with BCG-dsRed. **j**, MFI of dsRed signal within airway AM at 4 h and 24 h (**$P = 0.0072$) postinfection with BCG-dsRed. Each point represents biologically independent samples ($n = 3$ mice per group). **k**, Experimental schema. **l**, Histograms and MFI of MHC II expression (***$P = 0.0010$) on airway AM before *M. tuberculosis* infection. Each point represents biologically independent samples ($n = 3$ mice per group). Data in bar graphs are presented as mean ± s.e.m. Adjusted $P$ values are presented for violin plots and obtained using limma package and BH correction (Methods) (**e,f**). Statistical analysis was determined for data in **h**, **j**, and **l** by a two-tailed $t$-test, comparing BCG (US) with PBS (US) and BCG (S) with PBS (S).

pathway and fatty acid oxidation (Supplementary Table 1) were compared PBS (US) versus BCG (US) and PBS (S) versus BCG (S). In keeping with increased MHC II (Fig. 1b), expression levels of the genes involved

in antigen presentation significantly increased in BCG AM at baseline (US) and upon stimulation (S) (Fig. 2e), while those in glycolysis and mTOR pathways did not significantly differ (Extended Data Fig. 2e,f).

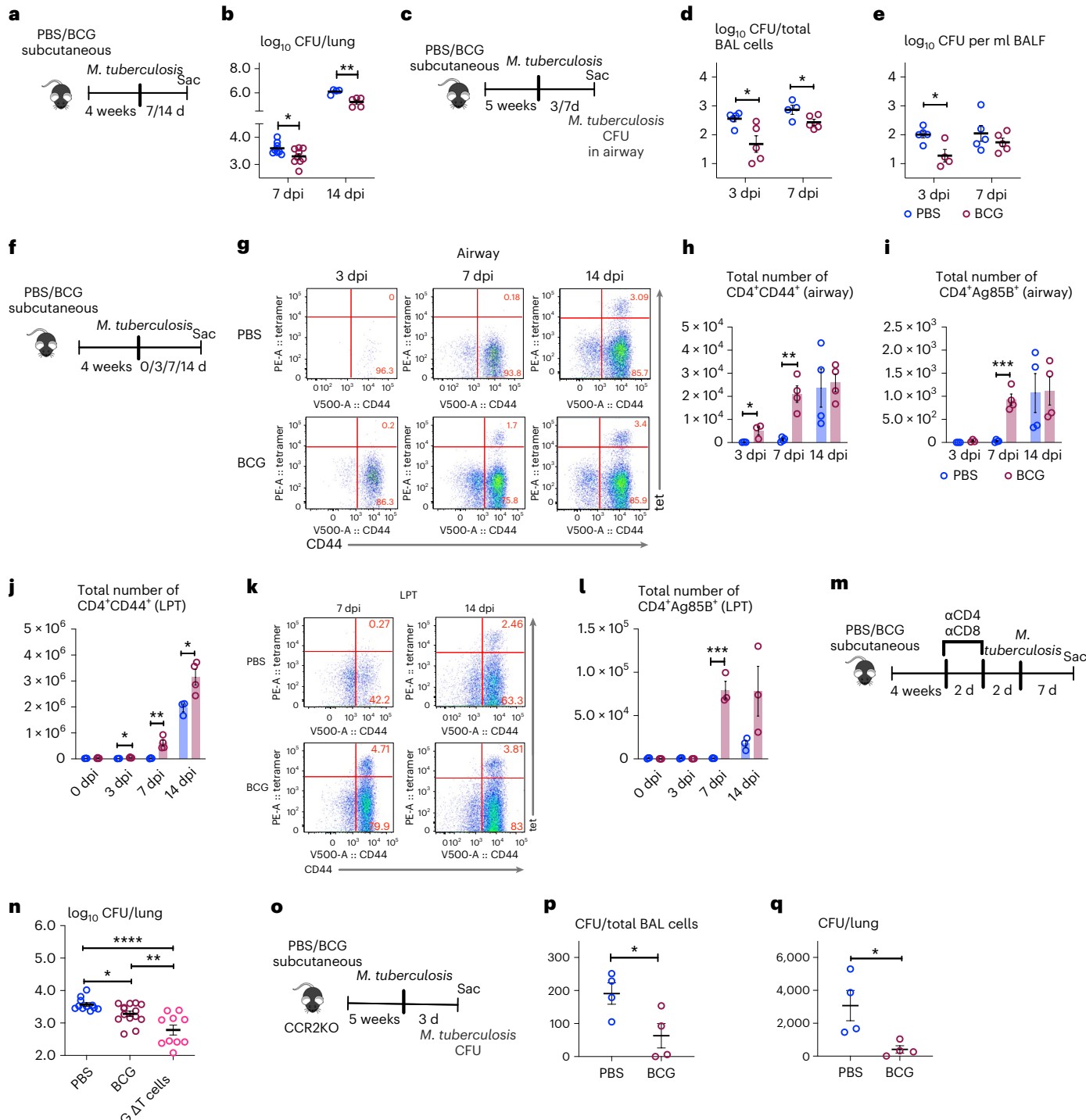

**Fig. 3 | Memory AMs confer trained immunity to pulmonary TB early in infection. a**, Experimental schema. **b**, Lung tissues CFU at 7 and 14 d post-*M. tuberculosis* infection (dpi) (*P = 0.0342; **P = 0.0047). Biologically independent samples pooled from two independent experiments for 7 dpi (n = 8 mice in PBS group; n = 9 mice in BCG group) and for 14 dpi (n = 4 mice in PBS group; n = 5 mice in BCG group). **c**, Experimental schema. **d,e**, Intracellular (BAL cells) *M. tuberculosis* CFU in the airway at 3 dpi (n = 5 mice per group; *P = 0.0208) and 7 dpi (n = 4 mice in PBS group; n = 5 mice in BCG group; *P = 0.0476) (**d**), and extracellular (BALF) *M. tuberculosis* CFU bacilli in the airway at 3 dpi (n = 5 mice in PBS group; n = 4 mice in BCG group; *P = 0.0152) and 7 dpi (n = 5 mice per group) (**e**). **f**, Experimental schema. **g**, Representative dotplots of airway CD4 T cells positive for activation marker CD44 and Ag85B tetramer at 3, 7 and 14 dpi. **h,i**, Total numbers of CD4+CD44+ (*P = 0.0490; **P = 0.0016) (**h**) and CD4+tet+ T cells (***P = 0.0003) (**i**) in the airway at 3 (n = 3 mice per group), 7 (n = 4 mice per group) and 14 (n = 4 mice per group) dpi. **j**, Total numbers of CD4+CD44+ T cells in LPT at

0, 3 (*P = 0.0162), 7 (**P = 0.0082) and 14 (*P = 0.0260) dpi. n = 3 mice in PBS group and n = 4 mice in BCG group for all time points. **k,l**, Representative dotplots (**k**) and total numbers of CD4+tet+ T cells in LPT postinfection (***P = 0.0002) (**l**). n = 3 mice per group for all time points. **m**, Experimental schema. **n**, Lung *M. tuberculosis* CFU at 7 dpi (*P = 0.0438; **P = 0.0013; ****P ≤ 0.0001). Biologically independent samples were pooled from three independent experiments. n = 11 mice in PBS group, 14 mice in BCG group and 10 in BCG ΔT cells group. **o**, Experimental schema. **p,q**, Numbers of intracellular (BAL cells) *M. tuberculosis* CFU in the airway (*P = 0.0404) (**p**) and lung tissue (*P = 0.0308) (**q**) at 3 dpi, n = 4 mice per group per tissue. The horizontal line in scatterplots denotes the mean with s.e.m. error bars. Data in bar graphs are presented as mean ± s.e.m. Statistical analysis was determined by two-tailed *t*-test for **b**, **d**, **e**, **h**–**j**, **l**, **p** and **q** comparing BCG with PBS groups, and data in **n** were analyzed by one-way ANOVA, followed by Fisher's least significant difference (LSD) test.

On the contrary, expression of genes related to fatty acid oxidation significantly decreased in stimulated BCG AM (Fig. 2f and Supplementary Table 1), consistent with a shifted metabolism from oxidation toward glycolysis (Fig. 1m,n)[12]. These data suggest that BCG vaccination leads to a unique transcriptional profile in memory AM.

We then used the data in the current study on differentially expressed genes (DEG) by BCG AM versus PBS AM and analyzed them against the DEG in AM of intranasally adenoviral (Ad)-vaccinated mice[14]. About five times more genes (1,309 versus 248 genes) were differentially expressed by trained Ad AM (Extended Data Fig. 2g) compared to BCG AM (Extended Data Fig. 2b). However, both shared similar features in predefined gene sets related to antigen presentation, glycolysis, mTOR pathway and fatty acid oxidation (Extended Data Fig. 2h). The transcriptomic features distinguishing Ad AM from BCG AM were the upregulated cell activation genes, *Lck*, *Bcl2*, *Il7r* and *Slamf7* and no enrichment for cell cycle and division-associated genes. These data suggest that although some features of trained AM are shared between certain vaccine strategies, the trained AM are unique depending on the vaccine type and delivery route.

Given that BCG-trained AM displayed enhanced MHC II and antigen presentation/processing genes, we assessed their capability of antigen presentation to T cells ex vivo by using transgenic *M. tuberculosis* Ag85B-specific CD4 T cells cocultured with Ag85B-laden BCG AM (Extended Data Fig. 3a). T cell proliferation rates were calibrated as the extent of CFSE dilution by FACS. While ~30% of T cells cultured with BCG AM underwent at least three generations of proliferation (G), only 15% of those cultured with control AM underwent mostly one-generation proliferation (Extended Data Fig. 3b,c), indicating enhanced antigen presentation by BCG AM.

To examine their antimicrobial activity, BCG and control airway AM were infected ex vivo with *M. tuberculosis* and mycobacterial inhibition/killing rates were determined by CFU assay (Fig. 2g). Compared to control AM, BCG AM exhibited a significantly greater ability to control *M. tuberculosis* at both 24-h and 48-h postinfection (Fig. 2h) or to control *Mycobacterium bovis* expressing a fluorescent protein (BCG-dsRed) (Fig. 2i,j). Augmented *M. tuberculosis* control was also seen in CD11C⁺/CD11b⁺ antigen-presenting cells (APC) from BCG lung tissue (Extended Data Fig. 3d). Both trained and control AM exhibited similar phagocytosis (Fig. 2i,j) and cell death/apoptosis (Extended Data Fig. 3e) rates. In keeping with their increased ex vivo MHC II-mediated *M. tuberculosis* antigen presentation (Extended Data Fig. 3b,c), trained AM rapidly further upregulated MHC II expression upon in vivo *M. tuberculosis* infection (Fig. 2k,l). These data indicate that besides their memory phenotype in BCG hosts, trained AMs show

a distinct transcriptional profile, increased antigen presentation and antimycobacterial activities.

## Trained immunity by memory lung macrophages against pulmonary TB

Anti-TB host defense has long been attributed solely to adaptive Th1 immunity induced by parenteral BCG vaccination. Airway AMs are known to harbor most of *M. tuberculosis* bacilli and contribute to its dissemination in the early stages of TB[38]. Enhanced mycobacterial control/responses by BCG-trained AM in ex vivo settings (Fig. 2h–l) suggest that such AM could offer TII against TB independent of T cell immunity in vivo. To begin examining whether memory AM offers anti-TB TII, 4-week BCG-vaccinated animals were infected with *M. tuberculosis* and lung CFU was assessed at days 7 and 14 (Fig. 3a). Compared to unvaccinated control, BCG lungs had ~0.3 log and ~1 log-reduced *M. tuberculosis* CFU at days 7 and 14, respectively (Fig. 3b). Consistent with airway AM being the primary *M. tuberculosis* reservoir within the first 7–9 d[38], BCG airway cells (BAL cells) at days 3 and 7 (Fig. 3c) contained significantly reduced *M. tuberculosis* CFU compared to nonvaccinated counterparts (Fig. 3d), coincided with significantly reduced *M. tuberculosis* CFU in cell-free fluid (BALF) of BCG hosts at day 3 (Fig. 3e). These data suggest that BCG-trained AM can better control *M. tuberculosis* in vivo.

We next investigated the potential role of BCG-activated Th1-cells in enhanced AM control of *M. tuberculosis* in BCG hosts. We first characterized the kinetics of antigen-specific CD4 T-cells in the airway and lung parenchymal tissue (LPT) at early time points post-*M. tuberculosis* (Fig. 3f) by using Ag85B-CD4 T-cell tetramers (tet)[39]. The T-cells within LPT were differentiated from intravascular counterparts via intravascular CD45.2 immunolabeling[40] (Extended Data Fig. 4a). Substantially activated CD4⁺CD44⁺ T cells appeared in BCG airways as early as day 3 postinfection, whereas, they did not populate the control airways until day 7 (Fig. 3g,h). Likewise, tet⁺CD4 T-cells (CD4⁺Ag85B⁺) were seen only in BCG airways at day 7 while being absent in the control airways (Fig. 3g,i). However, both CD4⁺CD44⁺ and CD4⁺Ag85B⁺ T-cells became comparable at day 14 between BCG and control hosts (Fig. 3g–i). Similarly, the BCG LPT had significantly greater numbers of activated CD4⁺CD44⁺ T-cells at days 3, 7 and 14 compared to the controls (Fig. 3j and Extended Data Fig. 4a) and also contained more CD4⁺Ag85B⁺ T-cells, particularly at day 7 (Fig. 3k,l). These data suggest the enhanced *M. tuberculosis* control by trained AM to be accompanied by increased Ag-specific T cells within the airway early during *M. tuberculosis* infection.

To address the direct relationship of trained AM to enhanced *M. tuberculosis* control, we depleted the T-cells in BCG hosts before

---

**Fig. 4 | Induction of memory AMs is independent of trained circulating monocytes and T cell-derived signals. a**, Experimental schema. **b**, Representative dotplots of myeloid (MMP3) and lymphoid (MMP4) progenitors and frequencies of MMP3 out of total multipotent progenitors in the BM (*P = 0.0120) (*n* = 3 mice in PBS group, *n* = 4 mice in BCG group). **c,d**, MFI of MHC II on circulating Ly6C^high (*P = 0.0228) (**c**) and Ly6C^low (*P = 0.0510) (**d**) monocytes with (S) and without (US) stimulation (*n* = 3 mice per group per cell type). **e**, Frequencies of Ly6C^low TNF⁺ monocytes with (S) and without (US) stimulation (*P = 0.0157) (*n* = 3 mice per group). **f**, Heatmap of cytokine/chemokine protein levels (geometric means) in the plasma from whole blood culture samples with (S) and without (US) stimulation. Red asterisks denote significant differences upon stimulation of airway AM of BCG hosts. **g,h**, Representative dotplots of SiglecF⁺Ly6C⁻ airway AM (**g**) and lung tissue (**h**). MDM and IM in lung tissue were identified as SiglecF⁻Ly6C⁺ and SiglecF⁻Ly6C⁻, respectively. The total numbers of macrophage subsets in airway and lung tissue are presented in the bar graph. Representative of two independent experiments (*n* = 3 mice per group per tissue). **i**, MFI of MHC II on airway AM from BCG-vaccinated or PBS-treated CCR2KO mice with (S) and without (US) stimulation and cytokine/chemokine levels in culture supernatant of airway AM with stimulation (S) (*P = 0.0280 for TNF; *P = 0.0335 for IL-6; *P = 0.0239 for IL-10). *n* = 3 mice in PBS group, *n* = 4 mice in BCG group. **j**, MFI of MHC II on lung tissue AM from BCG-vaccinated or PBS-

treated CCR2KO mice with (S) and without (US) stimulation (US: *P = 0.0238; S: *P = 0.0246) and frequencies of lung tissue AM producing IL-6 (US: *P = 0.0284; S: *P = 0.0297) and TNF with and without stimulation. **k**, Experimental schema. **l**, Representative histograms of PKH-labeled AM in the airway of BCG-vaccinated or PBS-treated WT and CCR2KO animals, compared to naïve mouse AM without PKH-labeling (no PKH). *n* = 3 mice per group. **m,n**, Signature scores for embryonic origin (AM) (**m**) and circulating monocyte genes (**n**) in airway AM in PBS and BCG-vaccinated hosts. Horizontal lines in violin plots denote medians and dotted lines denote lower and upper quartiles. **o**, Experimental schema. **p**, Heatmap of cytokine/chemokine protein levels (geometric means) in culture supernatants of AM with stimulation, comparing PBS, BCG-vaccinated and BCG/T cell depletion (dep) groups. Red asterisks denote significant differences upon stimulation of airway AM of BCG hosts. **q,r**, MFI of MHC II on airway AM (**P = 0.0066; ****P ≤ 0.0001) (**q**) and frequencies of IL-6-producing airway AM (*P = 0.0159; ***P = 0.0003; ****P ≤ 0.0001) (**r**) with (S) and without (US) stimulation, comparing PBS, BCG-vaccinated and BCG/IFN-γ-depleted (anti-IFNγ) groups. *n* = 3 mice per group. Data in bar graphs are presented as mean ± s.e.m. Statistical analysis was determined by two-tailed *t*-test for **b**–**e**, **i**, and **j**, comparing BCG with PBS. Data in **q** and **r** were analyzed by one-way ANOVA, followed by multiple comparisons with Bonferroni's test.

*M. tuberculosis* infection by using mAbs (Fig. 3m) and compared day 7 *M. tuberculosis* CFU in BCG lungs (BCG ΔT cells) with those in unvaccinated (PBS) and control Ab-treated BCG hosts (BCG). Consistent with the earlier data (Fig. 3b), BCG lungs (BCG) contained significantly reduced *M. tuberculosis* CFU (Fig. 3n). Depletion of T-cells in BCG hosts (BCG ΔT cells) did not compromise enhanced protection, but it rather further reduced *M. tuberculosis* CFU (Fig. 3n). Depletion of T-cells in unvaccinated (PBS) hosts did not affect *M. tuberculosis*

CFU (4.27 ± 0.06 in PBS versus 4.16 ± 0.08 in PBS ΔT cells). On the contrary, the naive animals receiving the adoptively transferred BCG AM (BCG-AM) had moderately reduced lung *M. tuberculosis* CFU compared to those receiving the control AM (PBS-AM; Extended Data Fig. 4b). We next examined the role of recruited monocyte-derived AM (MDM) in BCG hosts[41] by using the CCR2KO model of *M. tuberculosis* infection lacking classical Ly6C$^{hi}$ monocytes (Fig. 3o). BCG-vaccinated CCR2KO mice remained significantly better protected against

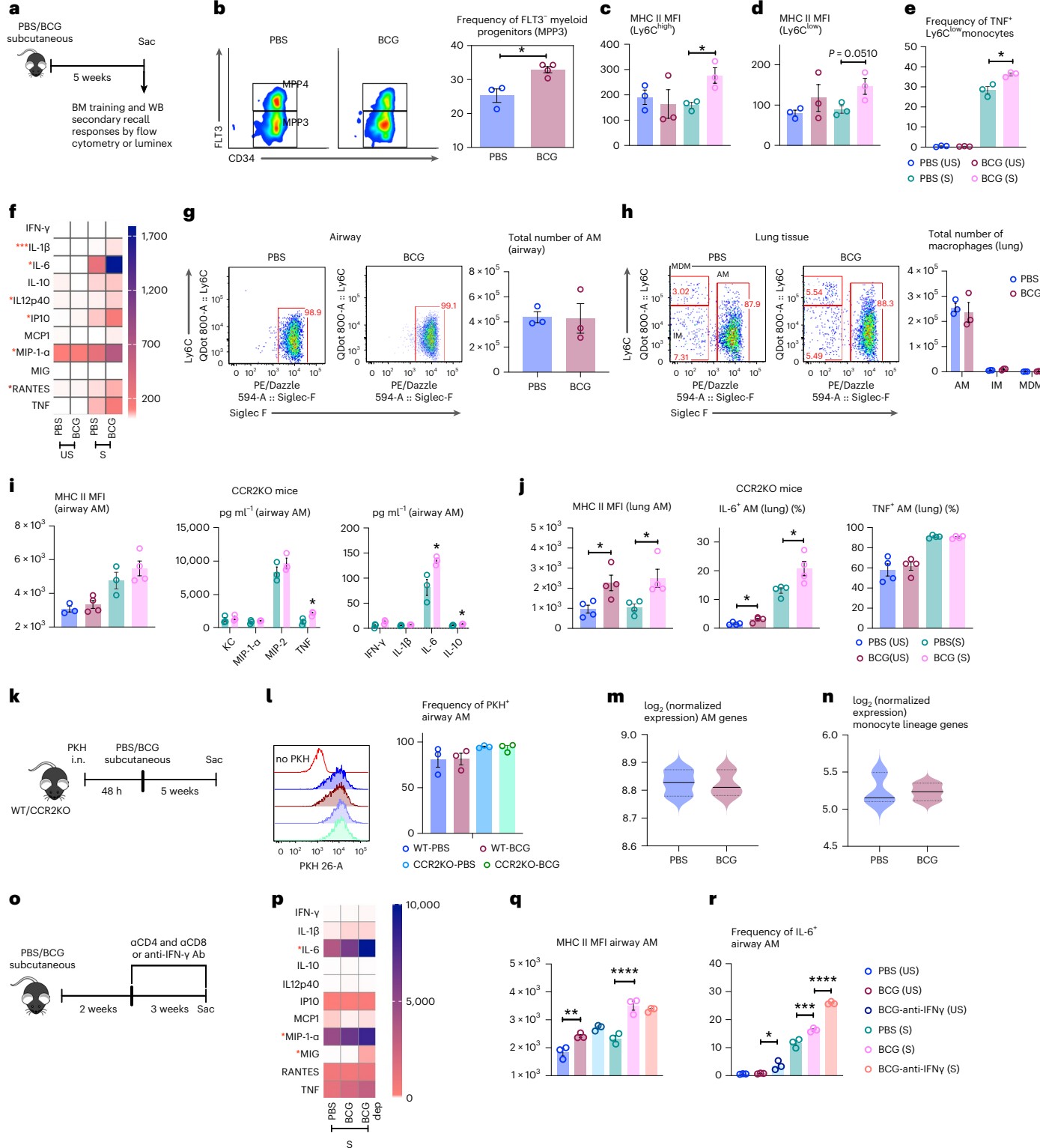

*M. tuberculosis* both in airway macrophages (Fig. 3p) and lung tissue (Fig. 3q) than the unvaccinated control. The above data together indicate the enhanced early TB protection in BCG hosts to be independent of T cells or circulating monocytes, further supporting the role of trained AM.

## Independence of monocytes and T cells for memory macrophage induction

Because parenteral BCG vaccination was shown to train circulating monocytes via imprinting the BM myeloid progenitors[9,13], we examined whether the BM myeloid cells and circulating monocytes were trained in our model (Fig. 4a). Indeed, significantly increased frequencies of MPP3 myeloid progenitors were observed in BCG hosts compared to the control (Fig. 4b). Using a gating strategy (Extended Data Fig. 5a), both Ly6C$^{hi}$ and Ly6C$^{low}$ monocytes in the peripheral blood of BCG hosts were found to express higher levels of MHC II upon stimulation compared to their controls (Fig. 4c,d and Extended Data Fig. 5b); however, they did not differ in their ability to produce IL-6 (Extended Data Fig. 5c). On the contrary, significantly increased Ly6C$^{low}$ monocytes from BCG hosts produced TNF upon stimulation (Fig. 4e) while TNF$^+$Ly6C$^{hi}$ monocytes were comparable (Extended Data Fig. 5d). Consistent with enhanced activation of circulating monocytes, IL-1β, IL-6, IL-12p40, IP-10, MIP-1α and RANTES production in stimulated whole blood cultures from BCG hosts significantly increased compared to the controls (Fig. 4f and Extended Data Fig. 5e). These data show that subcutaneous BCG vaccination leads to increased myelopoiesis in the BM and trained circulating monocytes.

The circulating monocytes may contribute to the pool of AM, particularly under inflammatory conditions in the lungs[21,22]. To address the relationship of trained circulating monocytes to the genesis of memory AM in BCG hosts, we first assessed the levels of pro-inflammatory cytokines/chemokines in the airway and found all of them to be undetectable. We next examined the airway macrophage and monocyte surface markers. At 2 weeks (Extended Data Figs. 6a) and 5 weeks (Fig. 4g) post-BCG, most airway (BAL) macrophages in both BCG and control hosts were Siglec-F$^+$ resident AM. Furthermore, frequencies and total numbers of major macrophage populations, monocyte-derived-macrophages (MDM) (Siglec-F$^-$Ly6C$^+$), interstitial macrophages (IM) (Siglec-F$^-$Ly6C$^-$) and AM (Siglec-F$^+$Ly6C$^-$) were similar in lung tissues of both groups (Extended Data Figs. 6b and 4h). These data thus do not support a substantial contribution of circulating monocytes to the induction of airway-resident memory AM in BCG hosts.

To investigate this further, CCR2KO mice lacking classical Ly6C$^{hi}$ monocytes were BCG-vaccinated for 5 weeks. Compared to the controls, MHC II expression remained elevated in both airway and lung tissue AM of BCG-vaccinated CCR2KO animals with and without stimulation (Fig. 4i,j). Furthermore, upon stimulation, compared to the controls, BCG AM from CCR2KO hosts produced significantly higher levels of IL-6 and TNF (Fig. 4i). Like the cytokine profile in wild-type (WT) BCG AM (Fig. 1d,e), there were significantly higher frequencies of stimulated CCR2KO BCG AM producing IL-6 while TNF-producing cells were comparable (Fig. 4j). The trained phenotype/immunity of AM from BCG CCR2KO animals was further supported by the functional data that these cells, upon ex vivo infection with *M. tuberculosis*, significantly better controlled mycobacterial infection (Extended Data Fig. 6c), consistent with ex vivo-infected WT BCG AM (Fig. 2g–j) and anti-TB TII in the lung of BCG CCR2KO animals (Fig. 3o–q). These findings together indicate the circulating monocyte-independent induction of functional memory AM by BCG vaccination.

Using a different approach, we delivered a stable fluorescent dye PKH26 to label airway-resident AM via phagocytosis[42] in both WT and CCR2KO mice, and any contribution of circulating monocytes toward BCG-trained AM would have diluted PKH26 within airway AM (Fig. 4k). In nonvaccinated WT (WT-PBS) and CCR2KO (CCR2KO-PBS) hosts, the majority of airway AM remained stably labeled by PKH over 5 weeks (Fig. 4l). There was no loss/dilution of PKH from AM of both WT and CCR2KO BCG hosts (Fig. 4l), suggesting a minimum contribution of circulating monocytes to BCG-trained AM. Because the autonomously induced AM would remain similar to their steady-state counterparts in their embryonic (AM) and monocytic gene signatures[14], we compared these genes in control and BCG and found no substantial differences (Fig. 4m,n), thus further supporting the independence of circulating monocytes for memory AM induction.

Because BCG activates Th1-cells (Fig. 3g,l,k,l) which produce IFN-γ involved in training monocytes/macrophages[13,14], we determined the role of T-cells and IFN-γ in BCG-trained AM. T-cells or IFN-γ were depleted by using mAbs from 2 weeks post-BCG when there was a lack of trained AM (Extended Data Fig. 1h–l) and their depletion was maintained over the next 3 weeks (Fig. 4o). While in keeping with earlier observations (Fig. 1f), BCG AM produced greater amounts of cytokines upon stimulation (S), and T-cell depletion did not compromise such enhanced responses by BCG AM to restimulation but it rather led to further increased IL-1β, IL-6, MIP-1α and MIG production (Fig. 4p). Similarly, IFN-γ neutralization did not impair the trained AM phenotype with elevated MHC II (Fig. 4q), IL-6 (Fig. 4r) and TNF (Extended Data Fig.

**Fig. 5 | Alterations in intestinal microbiota and microbial metabolites after BCG vaccination. a,b,** Alpha diversity comparison between 5-week PBS and BCG-vaccinated groups based on OTU richness using Chao1 diversity measure (*t*-test) (*P* = 0.0540) (**a**) and beta diversity comparison using PCoA ordination method and Jenson–Shannon divergence distance method (PERMANOVA) (*P* < 0.0040) (**b**) in cecum microbiota. *n* = 6 mice in PBS group, *n* = 5 mice in BCG group. **c,d,** Pie charts for relative abundance profile of top eight bacterial families (**c**) and bar graph comparing frequencies of top four abundant bacterial families in CM (*P* = 0.0505) (**d**). *n* = 6 mice in PBS group, *n* = 5 mice in BCG group. **e,f,** Representative micrographs of H&E-stained colon sections (*n* = 6 mice in PBS group, *n* = 5 mice in BCG group). Sloughed epithelium (**a**), inflammatory infiltrates in submucosal areas and lymphoid aggregates (**b**) in the colon of 5-week BCG hosts are marked. **g,** Histologic scoring of architectural changes, epithelium alterations and inflammatory infiltrates of the colon at indicated time points post-BCG. *n* = 5/2, 6/5, and 4/8 weeks. **h,** Immunohistochemical staining of ZO-1 protein in colonic epithelium. Red asterisk identifies irregular/disrupted distribution of ZO-1 in the colonic epithelium of 5-week BCG hosts. *n* = 6 mice in PBS group, *n* = 5 mice in BCG group. **i,** Comparison of intestinal permeability measured as an optical density of orally delivered FITC-dextran translocated into the circulation (*\*P* = 0.0012). *n* = 4 mice in PBS group, *n* = 3 mice in BCG group. **j,** PLS-DA analysis on metabolic profiles in the cecal tissue samples of 5-week PBS and BCG hosts. *n* = 16 mice per group. **k,** Comparison of deoxycarnitine

levels (relative peak area per gram wet weight) in cecal tissue samples of 5-week PBS and BCG hosts (*\*P* = 0.0312). *n* = 16 mice per group. **l,** PLS-DA analysis on metabolic profiles in the serum samples of 5-week PBS and BCG hosts. *n* = 13 mice in PBS group, *n* = 14 mice in BCG group. **m,** Proportions of butyrate SCFA were calculated out of total concentration (mM g$^{-1}$) of SCFAs in cecal tissue samples of 5-week PBS and BCG hosts (*P* = 0.0512). *n* = 16 mice per group. **n,** Comparison of PLS-DA analysis on metabolic profiles in the lung tissue samples from PBS, 2-week and 5-week BCG hosts. *n* = 10 mice per group. **o,p,** Comparison of carnitine products, butyryl carnitine (*\*\*\*P* = 0.0007) (**o**) and hexanoyl carnitine (*\*P* = 0.0268) (**p**; relative peak area per gram wet weight) in lung tissue samples of PBS, 2-/5-week BCG hosts. *n* = 10 mice per group per time. **q,** Comparison of colon length between 2-week PBS and BCG hosts (*\*\*P* = 0.0025). *n* = 4 mice in PBS group, *n* = 5 mice in BCG group. **r,** Numbers of BCG CFU in the MLN, cell-free PW in peritoneal cell fraction (PCL) of 2-week BCG hosts. *n* = 4 mice per group. Data in **a**–**d** are representative of two independent experiments. Data in **j**–**m** are from two pooled experiments (*n* = 16 mice per group). Horizontal lines in box plots denote medians and the length of the box denotes lower and upper quartiles, and the whiskers denote minimum and maximum values. Data in bar graphs are presented as mean ± s.e.m. Numeric numbers on pie charts represent median frequencies. Statistical analysis was determined by two-tailed *t*-test for data in **d**, **i**, **k**, **m** and **q**, comparing BCG with PBS. Data in **o** and **p** are analyzed by one-way ANOVA, followed by multiple comparisons with Bonferroni's test.

6d) production. These results are consistent with the fact that T-cell depletion in BCG-vaccinated hosts did not impair increased protection in the early stages of TB (Fig. 3n). The above data indicate that BCG induction/maintenance of airway-resident memory AM is independent of trained circulating monocytes, T cell help or IFN-γ.

## Alterations in intestinal microbiota, metabolites and barrier function

Recent evidence suggests that gut microbiome may undergo changes in response to distal infections and such changes can alter immune responses in the lung via the gut–lung axis[27–32,34]. To address whether the gut–lung axis was involved in mucosal-resident memory AM induction in our model, we first characterized the cecal microbiome at 5 weeks. BCG vaccination significantly reduced the cecum size at

5 weeks (Extended Data Fig. 7a). Alpha diversity comparison based on the operational taxonomic unit (OTU) revealed a significantly lower microbial richness or alpha-diversity index in BCG cecum (Fig. 5a). Actual abundance at the rank of bacterial phylum was also reduced in BCG animals with the mean abundance of top four phyla being 20,329 versus 33,125 ($P = 0.06$) in control animals (Extended Data Fig. 7b). The intestinal microbiota clusters from BCG hosts were significantly separated from those in control animals despite some overlap by principal coordinate analysis (PCoA) ($P < 0.004$, PERMANOVA; Fig. 5b). Moreover, at rank of bacterial family, while *Muribaculaceae*, also known as *S24-7*, predominated both in control and BCG hosts, a significantly increased frequency of *Lactobacillaceae* was present in the gut microbiome of BCG hosts (Fig. 5c,d). Nineteen specific OTUs were differentially changed in the BCG host (false discovery rate (FDR) < 0.05;

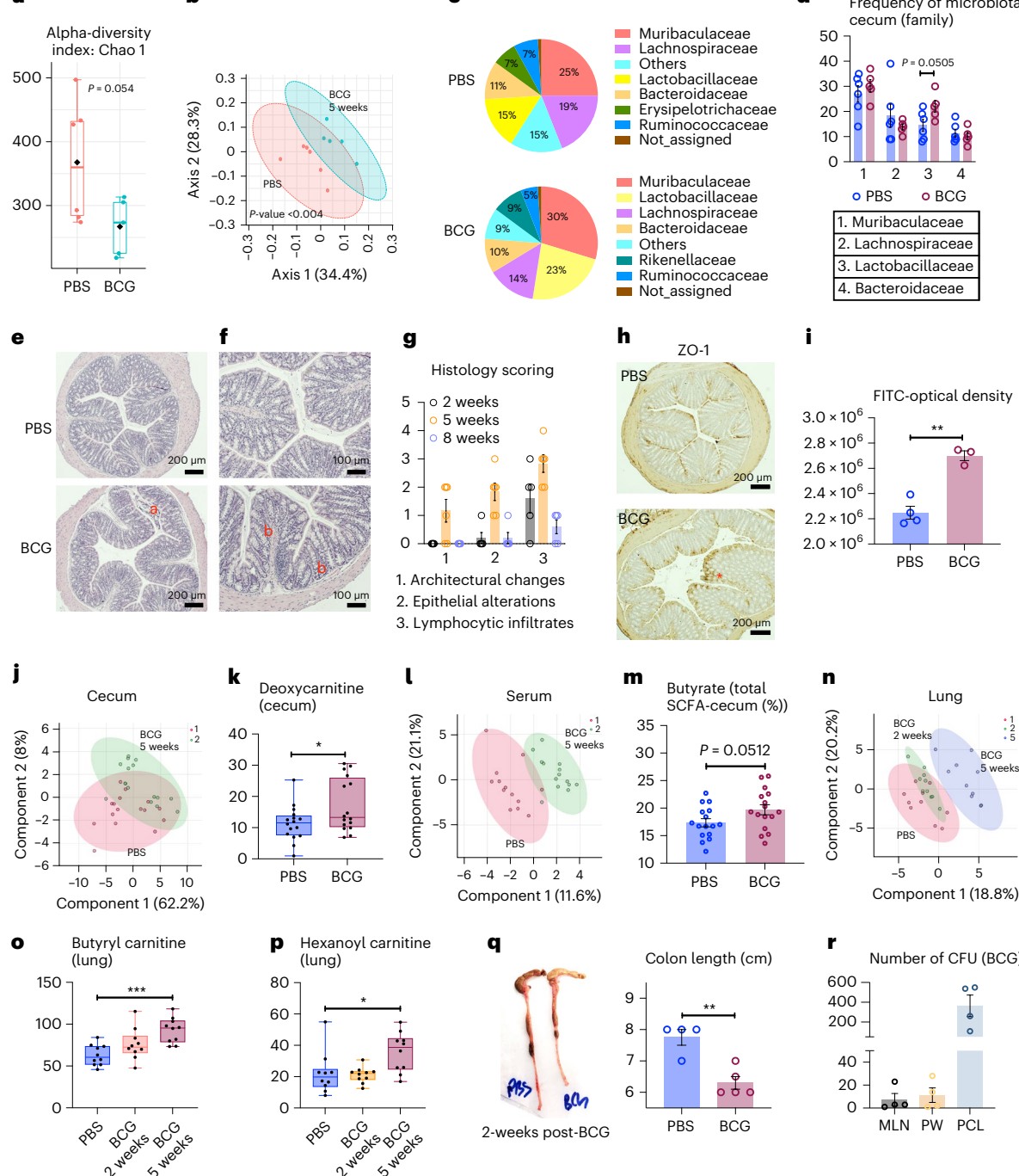

Supplementary Table 2). Because significant induction of TII in AM was observed at 5 weeks (Fig. 1a–p) but not at 2 weeks (Extended Data Fig. 1h–l), we also characterized the intestinal microbiome at 2 weeks post-BCG. Indeed, 2-week intestinal microbiome did not change much, resembling the control gut microbiome (Extended Data Fig. 7c–f), despite their partial overlap (Extended Data Fig. 7e) and 14 specific OTUs being differentially changed (Supplementary Table 3). These results indicate a time-dependent development of intestinal dysbiosis postparenteral BCG vaccination.

We next evaluated whether intestinal dysbiosis in BCG hosts was accompanied by microscopic histologic changes at 5 weeks postvaccination. Compared to the control, marked changes were observed in the distal colon of BCG hosts characterized by irregular villi, shortened crypts, enlarged lumen, epithelial disruption/sloughing and inflammatory infiltrates in the mucosa (Fig. 5e,f and Extended Data Fig. 7g). Although these changes are distinct, they were mild in severity according to the guidelines on murine intestinal inflammation[43]. To address whether BCG vaccination-triggered colitis was time-dependent and self-limited, we examined colon histology at 2 weeks and 8 weeks post-BCG and compared it to 5 weeks when the colitis was overt. Consistent with limited changes in intestinal microbiome at 2 weeks (Extended Data Fig. 7c–f), colon histology remained unchanged except for the low-degree lymphocytic infiltration (Fig. 5g and Extended Data Fig. 7h). Contrast to the colitis at 5 weeks (Fig. 5e,f), by 8 weeks postvaccination, colitis largely resolved and the colon architecture restored (Fig. 5g and Extended Data Fig. 7h). These findings indicate that parenteral BCG triggers time-dependent but self-limited intestinal inflammation. Given the colitis at 5 weeks, we immunohistochemically examined the expression of epithelium tight junction proteins, zonula occludens (ZO-1) and occludin. Irregular and disrupted the distribution of ZO-1 (Fig. 5h) and occludin (Extended Data Fig. 7i) was seen in the colonic epithelium of BCG hosts, compared to their even/intact distribution in control hosts. As a result, there was significantly increased intestinal permeability in BCG hosts shown by a fluorescein isothiocyanate (FITC)-labeled dextran method (Fig. 5i). These findings suggest an association of intestinal dysbiosis with intestinal structural changes and increased translocation of intestinal luminal molecules across the epithelium.

As intestinal dysbiosis is often linked to changes in its metabolites[28], we profiled the metabolome in the cecum, colon and serum from 5-week BCG animals. Partial least square-discriminant analysis (PLS-DA) indicated an intergroup clustering of cecal metabolites with some overlap (Fig. 5j). Compared to control hosts, deoxycarnitine/γ-butyrobetaine, a precursor of L-carnitine (Fig. 5k), and creatinine (Extended Data Fig. 8a) levels increased significantly in the cecum of BCG hosts. Metabolites in the colon of each group also clustered out into their own patterns (Extended Data Fig. 8b) with substantial increased lactic acid levels in BCG hosts (Extended Data Fig. 8c). Besides changes in intestinal microbial metabolites, the PLS-DA model showed a clear between-group clustering of serum metabolites (Fig. 5l). Based on variable importance in projection (VIP) scores (>1.5), ten discriminating serum metabolites were rank-ordered, showing decreased concentrations of all metabolites, except creatine, in BCG hosts compared to the controls (Extended Data Fig. 8d, colored boxes on the right). Pathway analysis of the murine serum metabolome revealed the arginine metabolic pathway to be predominantly impacted in BCG hosts compared to controls (Extended Data Fig. 8e). Because short-chain fatty acids (SCFAs) are among the major metabolites of gut microbiota generated upon dietary fiber breakdown and have inflammatory/metabolic impacts within and beyond the gut[28], we quantified the major SCFAs. While acetate in the cecum was not impacted by BCG-induced intestinal dysbiosis, the relative proportions of propionate and butyrate were altered (Extended Data Fig. 8f) with butyrate levels significantly increased (Fig. 5m). Using PICRUSt[44] as a predictive tool, we also explored the functional potential of the intestinal microbiome based on its differences between BCG and control animals. Nineteen predictive functional molecules significantly differed ($P < 0.05$), categorized into energy metabolism, transporters, signaling and cellular processes and genetic information processing (Supplementary Table 4). In keeping with elevated butyric acid levels in the cecum (Fig. 5m), acetolactate synthase I/II/III large subunit involved in butonate (salts and esters of butyric acid) metabolism was significantly increased in BCG hosts. Because besides metabolites, other luminal molecules including microbial-associated molecular patterns (MAMPs) may also translocate into the circulation due to increased permeability, we measured serum LPS levels but found them not to differ (<0.01 EU per ml; Extended Data Fig. 8g).

Given the metabolomic changes in the gut/serum of BCG hosts, we examined the metabolome in the lung at 2 and 5 weeks. Consistent with little and marked changes in gut microbiome/metabolome at 2 weeks (Extended Data Figs. 7c–f) and 5 weeks (Fig. 5a–d,j–m), respectively, the profile of lung metabolites significantly differed only at 5 weeks, but not at 2 weeks (Fig. 5n). Based on VIP scores (>1.5), seven discriminating lung metabolites were rank-ordered (Extended Data Fig. 8h). In keeping with increased deoxycarnitine levels in the cecum at 5 weeks (Fig. 5k), the carnitine products, butyrylcarnitine and hexanoylcarnitine, significantly elevated in the lung at 5 weeks post-BCG (Fig. 5o,p). These data demonstrate a time-dependent association in metabolomic changes between the gut, serum and lung.

Considering that besides lung AM, BCG vaccination had a global training effect on PM (Fig. 1o,p) and there exists a biological connection between the peritoneal cavity/macrophages and the organs including the gut within the peritoneal cavity[45,46], to address how BCG vaccination mediated changes in the gut we examined the possibility of BCG translocation to the peritoneal cavity at 2 weeks post-BCG. Indeed, not only was the length of the colon significantly different between control and BCG hosts with the latter being shortened by ~2 cm, suggesting mild colitis (Fig. 5q), but substantial BCG CFU was detected in the MLN, cell-free peritoneal washes (PW) and total macrophage fraction (peritoneal cell lysate (PCL)) only from BCG hosts (Fig. 5r), not from unvaccinated controls. These data establish a mechanistic linkage between distal BCG vaccination and the marked alterations in the gut.

The above data collectively suggest that via mycobacterial dissemination to the gut-associated sites, BCG vaccination leads to time-dependent alterations in intestinal microbiota, metabolome and barrier function which, in turn, result in metabolomic changes in serum and lung.

## BCG vaccine-conditioned intestinal microbiota induces memory lung macrophages

We next used a microbiota transplant approach to address the relationship of BCG-induced intestinal dysbiosis to induction of memory AM. Naïve mice were treated with broad-spectrum antibiotics for 10 d to perturb the original microbiota before transplantation with cecal microbiota (CM) from control (PBS-CM) or BCG (BCG-CM) hosts and allowed to colonize over the next 5 weeks (Fig. 6a). Consistent with the trained phenotype of BCG AM (Fig. 1a–n), airway AM of those colonized with BCG-CM demonstrated elevated MHC II and IL-6 and TNF production at baseline (US) and upon stimulation (S) compared to those colonized with PBS-CM (Fig. 6b–d). A similar trained phenotype was also seen with AM in BCG-CM, but not PBS-CM, lung tissue (Fig. 6e–g). These data, thus, establish a causal relationship between BCG-conditioned microbiota and memory AM induction.

As expected, unlike in BCG-vaccinated hosts (Fig. 5e,f and Extended Data Fig. 7g), BCG-CM colonization of naïve animals did not cause major colonic architectural changes (Fig. 6h). However, it led to epithelial hyperplasia with reduced goblet cells and mild epithelium disruption (Fig. 6h) and reduced epithelial mucin-2 (muc-2) production compared to PBS-CM controls (Fig. 6i). Because gut dysbiosis is linked to changes in BM myeloid hematopoiesis[47], we examined the

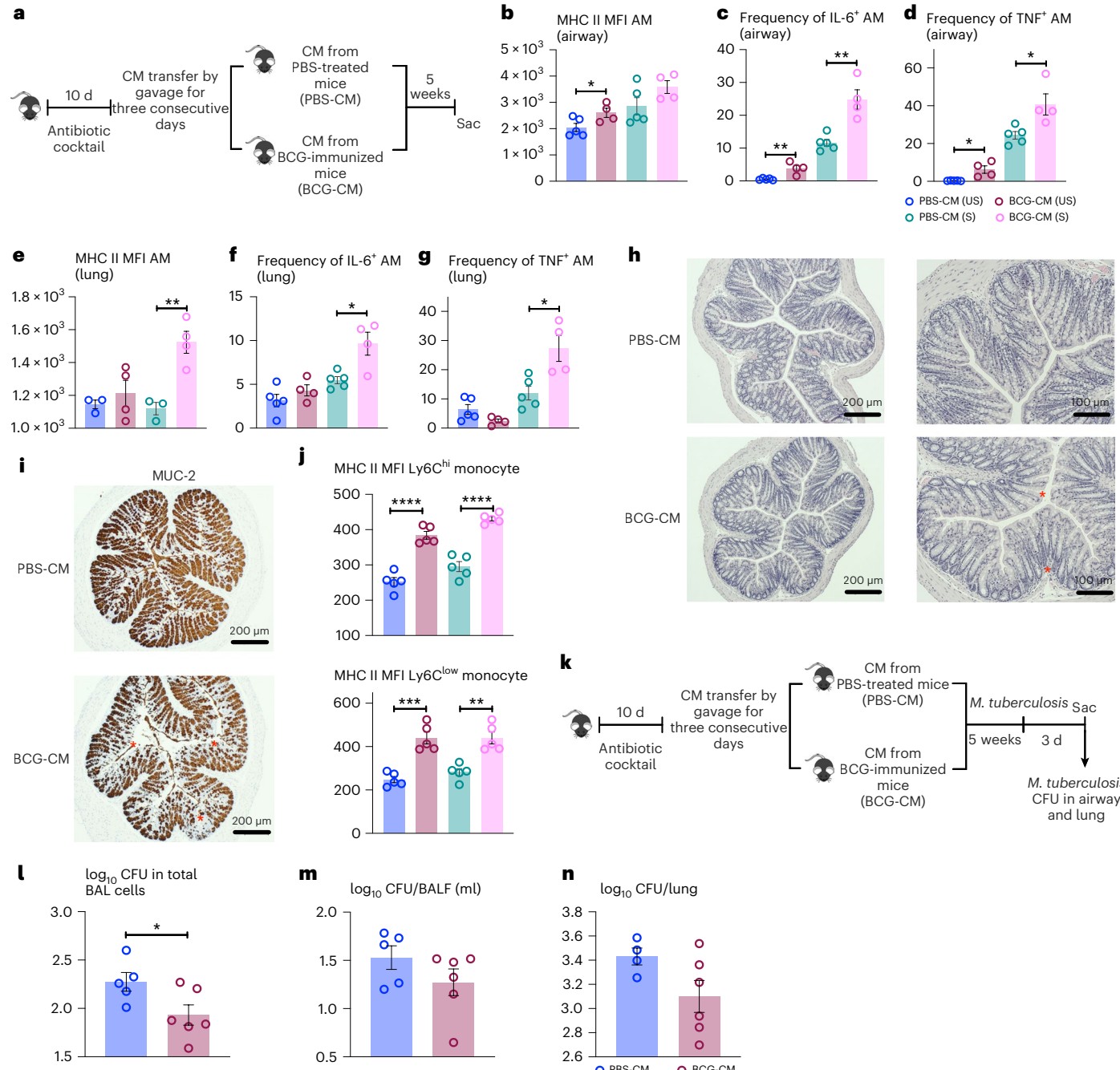

**Fig. 6 | Transplantation of BCG vaccine-conditioned intestinal microbiota induces memory AMs and trained immunity. a**, Experimental schema. **b,c,d**, Trained innate immune characteristics by airway AM from BCG-CM group: MFI of MHC II (*P = 0.0477) on airway AM (**b**), frequencies of IL-6 (**P = 0.0046; **P = 0.0025) (**c**) and TNF (*P = 0.0109, *P = 0.0192) (**d**) producing airway AM with (S) and without (US) stimulation. n = 5 mice in PBS-CM group, n = 4 mice in BCG-CM group. **e,f,g**, Trained innate immune characteristics by lung tissue AM from BCG-CM group: MFI of MHC II (**P = 0.0052) on lung AM (**e**), frequencies of IL-6 (P = *0.0122) (**f**) and TNF (*P = 0.0144) (**g**), producing lung AM with (S) and without (US) stimulation. Representative of two independent experiments. n = 5 in PBS-CM group, n = 4 mice in BCG-CM group. **h**, Representative micrographs of H&E-stained colon sections (n = 5 mice in PBS-CM group, n = 4 mice in BCG-CM group) showing epithelium hyperplasia, reduced goblet cells and mild epithelium disruption in the colon of BCG-CM mice. Red asterisks identify epithelium disruption and hyperplasia. **i**, Representative micrographs of colon sections

(n = 5 mice in PBS-CM group, n = 4 mice in BCG-CM group) immuno-histochemically stained for MUC2 protein. Red asterisks identify significantly reduced MUC2 staining in the colonic epithelium of BCG-CM mice compared to PBS-CM mice. **j**, MFI of MHC II on circulating Ly6C$^{hi}$ (****P < 0.0001) and Ly6C$^{low}$ (***P = 0.0002, **P = 0.0011) monocytes from BCG-CM mice with and without stimulation, compared to those from PBS-CM mice. n = 5 mice per group. **k**, Experimental schema. **l,m,n**, Numbers of intracellular (BAL cells) (*P = 0.0431) (**l**) and extracellular (BALF) (**m**) *M. tuberculosis* CFU in the airway (n = 5 in PBS-CM group, n = 6 mice in BCG-CM group) and those in lung tissue (**n**) (n = 4 mice in PBS-CM group, n = 6 mice in BCG-CM group) of BCG-CM mice, compared to PBS-CM mice, at 3 d postinfection. Data in bar graphs are presented as mean ± s.e.m. Statistical analysis was determined by two-tailed t-test for all data in **b–g**, **j** and **l**, comparing PBS-CM (US) versus BCG-CM (US) and PBS-CM (S) versus BCG-CM (S).

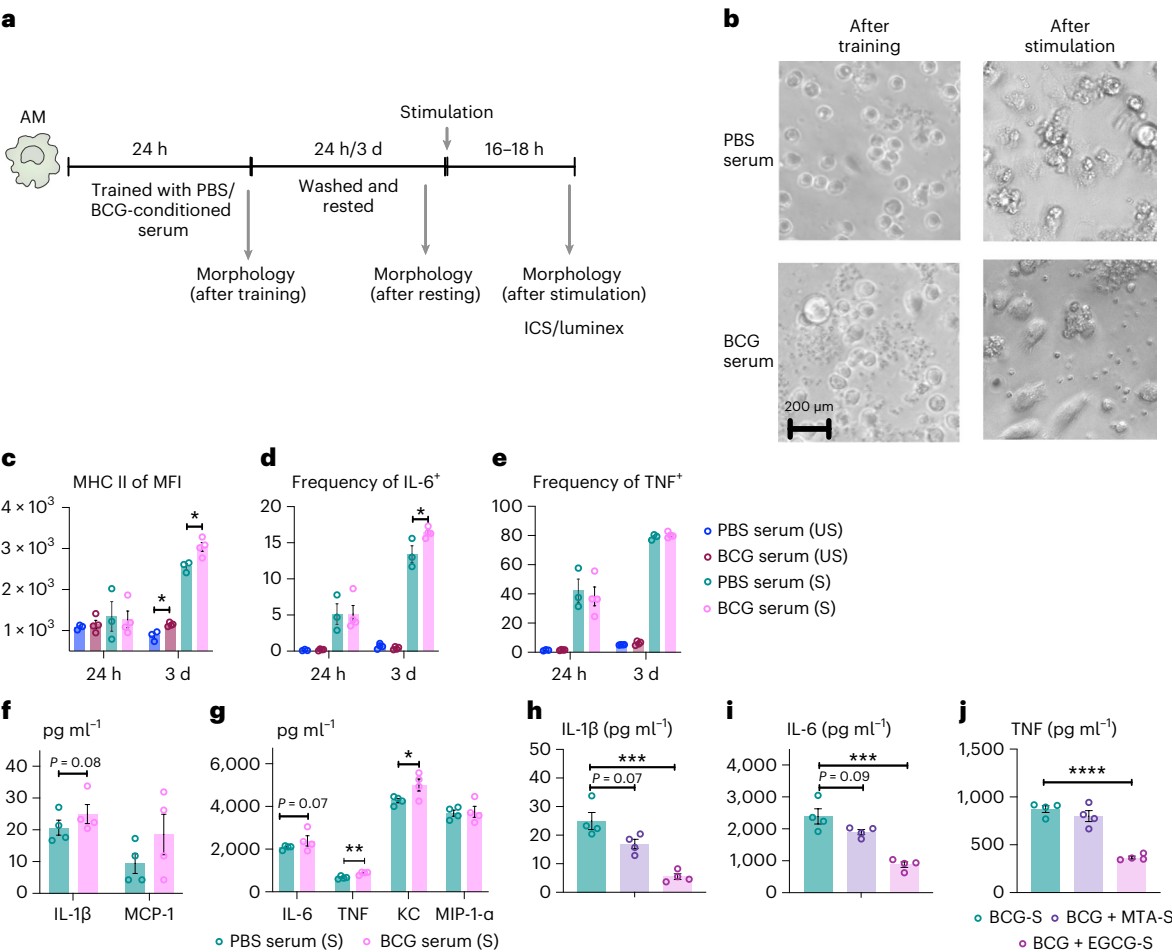

**Fig. 7 | Circulating microbial metabolites in BCG-vaccinated hosts leads to innate immune training of AMs. a**, Experimental schema of in vitro innate training of AM. **b**, Representative bright-field microscopic images of AM after training with BCG-S or treatment with control serum or after restimulation. Representative of two independent experiments (*n* = 4 wells per condition). **c**–**e**, Increased median fluorescence intensity (MFI) of MHC II (*P* = 0.0132 and *P* = 0.0182) (**c**) and frequencies of IL-6 (*P* = 0.0362) (**d**), but not TNF (**e**), producing AM trained with BCG-S and upon re-stimulation after 24-h or 3-d resting. Each data point represents *n* = 3 wells per PBS serum and *n* = 4 wells per BCG serum. **f**,**g**, Cytokine/chemokine protein contents in culture supernatants of AM trained with BCG-S and upon restimulation after 3-d resting. Each data point represents *n* = 4 wells per condition. TNF, **P* = 0.0047; KC, *P* = 0.0491. **h**–**j**, Inhibition of innate immune training of AM by BCG-S upon histone methylation and acetylation blockade with methyltransferase inhibitor MTA or acetyltransferase inhibitor EGCG. Data shown are IL-1β (***P* = 0.0008) (**h**), IL-6 (***P* = 0.001) (**i**) and TNF (****P* < 0.0001) (**j**) protein contents produced by AM upon restimulation. Each data point represents *n* = 4 wells per condition. Data in bar graphs are presented as mean ± s.e.m. Statistical analysis was determined by two-tailed *t*-test for data in **c**–**e**, comparing PBS serum (US) versus BCG serum (US) and PBS serum (S) versus BCG serum (S) within each time point poststimulation for data in **f** and **g**, comparing PBS serum (S) versus BCG serum (S) for **h**–**j**, comparing to BCG-S.

circulating monocytes and found BCG-CM colonization to significantly activate circulating Ly6C^high and Ly6C^low monocytes over PBS-CM controls (Fig. 6j).

To investigate whether BCG-CM colonization-trained AM might translate to lung TII as seen in BCG-vaccinated hosts (Fig. 3b,d,e,n,p,q), the CM-colonized naive animals were infected with *M. tuberculosis* and CFU was assessed at 3 d (Fig. 6k). Indeed, airway BCG-CM AM (BAL) contained significantly reduced mycobacterial bacilli compared to PBS-CM controls (Fig. 6l). Correspondingly, CFUs in cell-free BALF and lung tissue were also trending smaller in BCG-CM animals (Fig. 6m,n). The above data together indicate that transplantation of BCG-conditioned intestinal microbiota alone can induce intestinal barrier changes, memory AM and anti-TB TII in the lung.

**Role of circulating microbial metabolites in training lung macrophages**

To examine the relationship of circulating metabolites to BCG-trained AM, we adapted an in vitro model well-established for monocyte

training[48]. Naive airway AMs were incubated in a culture medium supplemented with serum from BCG-vaccinated or control animals (training). After 24-h training and 24-h or 3-d resting, cells were stimulated, microscopically analyzed and immunophenotyped (Fig. 7a). Different from control serum, BCG-conditioned serum (BCG-S) caused remarkable morphologic changes of AM after training or resting and upon restimulation (Fig. 7b and Extended Data Fig. 9a). These AM congregated in clusters and were larger with cytoplasmic inclusions (after training; Fig. 7b). Upon resting, there appeared increased cell divisions and spreading (after resting), consistent with their enriched genes involved in cell division in AM from BCG hosts (Fig. 2c and Extended Data Fig. 2b–d). The most marked morphologic changes were seen upon restimulation (after stimulation), accompanied by significantly elevated MHC II and IL-6 production (Fig. 7c–e). This trained immunophenotype was similar to memory AM in BCG-vaccinated hosts (Fig. 1b,d,e) and was observed only with the AM rested for 3 d but not with those rested for 24-h after training, consistent with previous observations[48]. Furthermore, production of

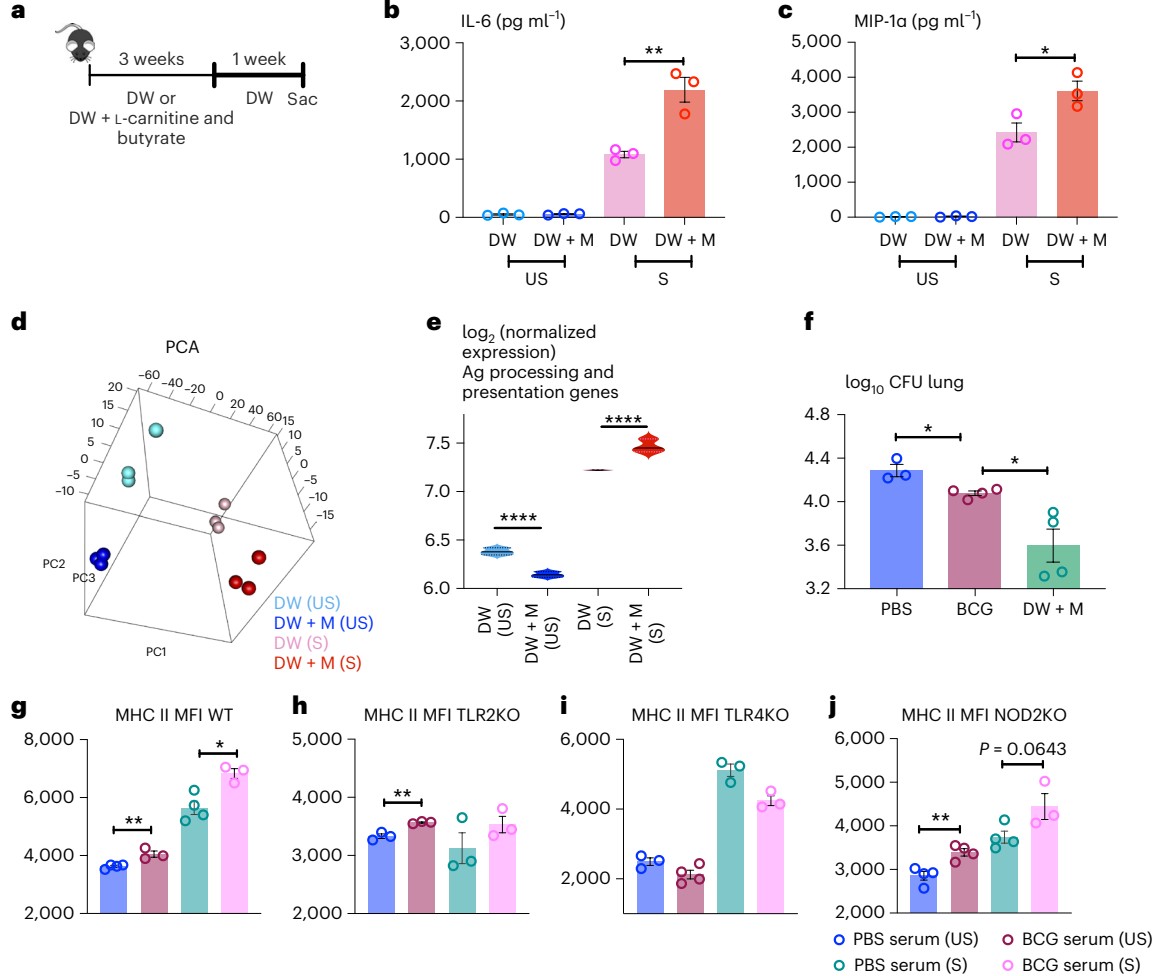

**Fig. 8 | Water supplemented with microbial metabolites in BCG-vaccinated hosts leads to innate immune training in AMs in naïve mice. a**, Experimental schema of in vivo continuous supplementation of circulating SCFAs butyrate and L-carnitine via (DW + M) given to naïve mice for 3 weeks. Mice were then placed on regular DW for 1 week. The control animals were on regular DW. **b**,**c**, IL-6 (*P = 0.0071) (**b**) and MIP-1α (*P = 0.0381) (**c**) protein production by airway AM isolated from animals on DW + M or DW, with (S) or without (US) re-stimulation. n = 3 mice per group. **d**, PCA of gene expression in airway AM from metabolite-supplemented (DW + M) and control (DW) hosts with (S) or without (US) stimulation. n = 3 mice per group. **e**, Signature scores of genes involved in Ag processing and presentation in airway AM comparing control (DW) with metabolite-supplemented (DW + M) animals with (S) and without (US) stimulation (P = 4.34 × 10⁻⁶). Horizontal lines in violin plots denote medians and dotted lines denote lower and upper quartiles. **f**, Numbers of *M. tuberculosis* CFU in the lung tissue of naïve animals given metabolite-supplemented water (DW + M) compared to BCG-vaccinated (DW + M versus BCG, *P = 0.0203) (BCG versus PBS *P = 0.0115) n = 3 mice in PBS group, n = 4 mice in BCG group, 4/DW + M. **g–j**, Comparing MFI of MHC II expression in airway AM from WT (**P = 0078 and *P = 0.0118) (**g**), TLR2KO (**P = 0.0043) (**h**), TLR4KO (**i**) and NOD2-KO (**P = 0.0062) (**j**) mice trained ex vivo with BCG-conditioned or control serum and upon restimulation after 3-d resting. Data in bar graphs are presented as mean ± s.e.m. Statistical analysis was done by two-tailed t-test for data in **b**, **c** and **f**. Adjusted P values are presented for violin plots (**e**) and obtained using limma package and BH correction (Methods). Statistical analysis was determined by two-tailed t-test for data in **g–j**, comparing PBS serum (US) versus BCG serum (US) and PBS serum (S) versus BCG serum (S).

training biomarkers IL-1β, IL-6 and TNF along with MCP-1 and KC significantly increased from the AM exposed to BCG-S over the controls (Fig. 7f,g). These data suggest a role for circulating soluble factors in BCG-trained AM.

Because innate training involves epigenetic reprogramming via histone methylation/acetylation[1], we assessed whether histone modification was involved in the observed training effect of BCG-S on AM. Thus, during the training with BCG-S, the culture medium was supplemented with either histone methyltransferase inhibitor, 5'-deoxy-5'-methylthioadenosine (MTA) or histone acetyltransferase inhibitor epigallocatechin-3-gallate (EGCG)[48]. Inhibition of histone modification enzymes, particularly histone acetyltransferase (BCG + EGCG-S), significantly reduced IL-1β, IL-6 and TNF production by BCG-S-trained AM upon restimulation (Fig. 7h–j). A relatively minor inhibitory effect was observed with methyltransferase inhibitor (BCG + MTA-S). These data

suggest the involvement of epigenetic modification in AM training by circulating metabolites in BCG hosts.

Given the immunomodulating role of intestinal microbiome-derived circulating SCFAs[28] and our observed increases in deoxycarnitine and butyrate in BCG-vaccinated hosts (Fig. 5j–p), we determined the relationship of these SCFAs to memory AM induction. A mix of L-carnitine and butyrate hydrochloride was introduced to the drinking water (DW + M) of naive animals for 3 weeks[49] and the control animals received the regular drinking water (DW) (Fig. 8a). DW + M AM demonstrated enhanced IL-6 and MIP-1α production among the cytokines examined over the control (DW) upon stimulation (S) (Fig. 8b,c and Extended Data Fig. 9b), similar to the profile of BCG-trained AM (Fig. 1g,k). However, unlike the circulating monocytes in BCG hosts (Fig. 4f and Extended Data Fig. 5), the DW + M monocytes did not assume a trained phenotype, displaying a suppressed immune

profile with reduced Ly6C and MHC II expression (Extended Data Fig. 9c,d). Their ability to secrete cytokines/chemokines also remained comparable (Extended Data Fig. 9e). Furthermore, as expected, unlike in BCG-vaccinated hosts (Fig. 5e,f and Extended Data Fig. 7g), the metabolite supplementation (DW + M) did not cause colonic architectural changes except a mild lymphocytic infiltration (Extended Data Fig. 9f).

We next compared the transcriptomic profiles in trained AM by metabolite supplements and BCG vaccination. DW + M airway AM (Fig. 8a) were subjected to transcriptional analysis with (S) and without (US) stimulation. Each of the groups was found to cluster into its own pattern, suggesting the transcriptional alteration in AM following metabolite treatment (Fig. 8d). A total of 265 genes were differentially expressed in DW + M AM compared to DW controls (Extended Data Fig. 9g,h). Like BCG AM (Fig. 2c and Extended Data Fig. 2d), the genes associated with cell differentiation/proliferation (*Nov*, *Hbegf*, *Kitl* and *Six5*) were also upregulated in DW + M AM compared to controls (Extended Data Fig. 9g). The gene, *Snca*, a microphage/microglial activation gene required for inflammatory responses was also upregulated in both AM (Fig. 2d and Extended Data Fig. 9g). Also similar to BCG AM (Figs. 1b and 2c,e), the immune genes including HLA genes (Extended Data Fig. 9h) and antigen presentation genes (Fig. 8e) were significantly increased in stimulated DW + M AM. Furthermore, the levels of the genes associated with fatty acid oxidation, glycolysis and mTOR pathway in DW + M AM (Extended Data Fig. 9i–k) were in general similar to those of BCG AM (Fig. 2f and Extended Data Fig. 2e,f). Of importance, there was significantly enhanced protection in the early stages of *M. tuberculosis* infection in DW + M lungs, even to a greater extent than in BCG hosts (Fig. 8f). These findings indicate that supplementation of BCG-altered metabolites alone could induce trained immunophenotype, transcriptomic changes and TII in AM similarly as BCG vaccination.

Given that serum LPS was undetectable (Extended Data Fig. 8g), we further investigated the role of intestinal MAMPs potentially translocated into the circulation in BCG-vaccinated hosts. Considering microbial peptidoglycan, LPS and muramyl dipeptide are ligands for TLR2, TLR4 and NOD2, respectively, TLR2-, TLR4- and NOD2-deficient (KO) AM were trained ex vivo with BCG-S or PBS-conditioned serum using our ex vivo AM training model (Fig. 7a). While BCG-S induced significantly increased MHC II, an innate training marker, on WT AM without (US) and with (S) stimulation (Fig. 8g), it also induced increased MHC II on un-stimulated TLR2- or NOD2-KO AM (Fig. 8h,j). Stimulation with *M. tuberculosis* WCL enhanced MHC II further on NOD2-KO AM, particularly on those trained by BCG-S (Fig. 8j). Interestingly, TLR4-KO AM expressed reduced MHC II which did not differ between control and BCG serum before stimulation (Fig. 8i). Although the stimulation increased MHC II in these cells, it did not differ between control and BCG-S, suggesting the inherent requirement of TLR4 signaling for MHC II expression by AM. The above data suggest that even if present at heightened circulating levels, these MAMPs do not contribute significantly; however, circulating microbial metabolites have a critical role in AM training by BCG vaccination.

## Discussion

It remains unclear whether, in the absence of local inflammation, lung-resident innate memory may occur following integrating and adapting to distally generated immunological signals postparenteral vaccination. Here, we show that subcutaneous BCG vaccination induces memory AM and TII in a time-dependent manner besides its training effects on circulating monocytes. Such memory AM develops and self-sustains independently of circulating monocytes. BCG vaccination does so via the gut–lung axis involving mycobacterial translocation, intestinal dysbiosis and increased permeability and changes in local/systemic metabolites (Extended Data Fig. 10).

Our study, thus, reveals a new intestinal microbial metabolic pathway for innate memory/TII development at a distal mucosal

site postparenteral vaccination (Extended Data Fig. 10), and it changes the current view that genesis of innate immune memory is compartmentalized according to the route of immunologic exposure[2,3,20]. Thus, parenteral vaccination could trigger a long-range immunological alert across multiple tissue sites, resulting in macrophage memory formation. Such knowledge shall enhance our understanding of host defense mechanisms by parenteral vaccines. It indicates that following immunization with a properly designed parenteral vaccine and upon respiratory entry of pathogens, trained mucosal-resident macrophages act as the first line of host defense which can be reinforced via the recruitment of trained circulating monocytes, a mechanism referred to as 'canonical tissue trained immunity'[20]. It also offers an additional mechanism for enhanced nonspecific innate protection in the lung of BCG-vaccinated humans[5–7,9]. It is noteworthy that different from parenteral BCG vaccination, parenteral adenoviral-vectored vaccination is unable to train AM[19], suggesting the importance of choices of vaccine platform and route of delivery to mucosal-resident TII induction. The inability of BCG-ia to train AM suggests that the replicability of the parenteral vaccine is required for its widespread immunological alert and global macrophage-training effects, which is supported further by our finding that BCG spread appears required to initiate the gut–lung axis. That BCG replicates slowly may explain a slow build-up of its spread and a time-dependent manifestation of intestinal dysbiosis, colitis and metabolomic shifts. Of importance, mild colitis is self-limited as by 8 weeks it is largely resolved. Although two recent reports show the changes in intestinal microbiome following parenteral BCG vaccination[50,51], there have not been any clinical reports on parenteral BCG-related colitis, let alone its linkage to metabolomic shifts in the gut and TII induction in the lung. The mild/transient nature of BCG vaccination-related colitis could explain its clinical insignificance. Furthermore, because it is only a proportion of parenteral BCG-vaccinated humans that develop innate protection against *M. tuberculosis* in the lung[52], induction of lung-resident anti-TB TII via the gut–lung axis is likely genetically determined and ensues only in some human BCG vaccinees.

Our study also offers evidence that parenteral vaccination can induce intestinal dysbiosis-associated local/systemic metabolomic changes. We further demonstrate that induction of lung-resident innate memory via the gut–lung axis is independent of T cells or IFN-γ, different from their central role in the genesis of lung mucosal-resident macrophages via a local inflammatory pathway[14]. Besides its effects on the lung, intestinal dysbiosis/metabolites could also train circulating monocytes as shown in our current study, likely through influencing BM myelopoiesis[47]. As gut microbiota-derived SCFAs possess immune-modulatory properties[23,33], a decline in acetate, a predominant SCFA, in prior flu-experienced mice affected AM bactericidality[30] and deficient SCFA production hampered microglia maturation[49]. Our finding that elevated deoxycarnitine and butyrate levels in gut–lung are linked to AM training significantly adds to our understanding of innate regulatory properties of SCFAs. Our approach of supplementing via drinking water of SCFAs to induce lung TII presents a potential immunotherapeutic strategy. Future studies shall address whether parenteral BCG-triggered intestinal dysbiosis may train macrophages in other mucosal tissues than the lung and peritoneal cavity.

Our findings also highlight the plasticity of tissue-resident AM in lung homeostasis and host defense[3,21]. The trainability, durability and autonomy of AM are in keeping with their ability to patrol the alveoli via crawling and to kill bacteria at a greater-than-neutrophil rate[53]. Thus, we show memory AM to enhance TB protection in BCG-vaccinated hosts independent of T cells or circulating monocytes. This is a plausible mechanism underlying the innate clearance of *M. tuberculosis* observed in a substantial proportion of BCG-vaccinated humans[52]. Having trained AM at the site of *M. tuberculosis* entry in BCG vaccinees is of importance to early control of infection given the ability of *M. tuberculosis* to hijack

airway macrophages for its dissemination[19,38,39]. Besides *M. tuberculosis*, we are currently investigating if parenteral BCG-induced lung TII can provide protection against heterologous bacterial infection.

In conclusion, our study reveals a new parenteral vaccine-triggered intestinal microbiota-mediated pathway to innate memory development in distal mucosal tissues. The work shows that parenteral immunization with a live vaccine can both centrally and peripherally induce TII. Such knowledge shall help design the next-generation vaccines against respiratory pathogens such as *M. tuberculosis* and SARS-CoV-2 (refs. [35,36]).

## Online content

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

                    

## Methods

### Mice

WT female C57BL/6 mice were purchased from Charles River Laboratories (Saint Constant) or the Jackson Laboratory (Bar Harbor). Female chemokine (C-C motif) receptor 2 (CCR2) (B6.129S4-$Ccr2^{tm1Ifc}$/J), TLR2 (B6.129-$Tlr2^{tm1Kir}$/J), TLR4 (B6(Cg)-$Tlr4^{tm1.2Karp}$/J), NOD2 (B6.129S1-$Nod2^{tm1Flv}$/J) knock-out and P25 TCR-Tg transgenic mice containing CD4 T cells expressing Ag85B receptor (H2-K$^b$-Tcra,-Tcrb)P25Ktk/J) on a C57BL/6 background were purchased from the Jackson Laboratory. All mice were 6–8 weeks of age upon arrival. Mice were housed in a specific pathogen-free level B facility or at the biosafety level 3 facility with ad libitum access to food and water, 12 h light cycle, 50–60% humidity and at 20–25 °C room temperature at McMaster University. Age-matched mice housed in the same room were used in each experiment. Control mice were administered subcutaneously with PBS used for the preparation of BCG. Animals were assigned experimental groups at random. All experiments were carried out in accordance with the institutional guidelines from the Animal Research and Ethics Boards of McMaster University (AUP 210822).

### Subcutaneous immunization with BCG

Mice were immunized subcutaneously with 4–5 × 10$^4$ CFU of BCG Pasteur strain[54]. Heat-BCG-ia was prepared by incubating 4–5 × 10$^4$ CFU of BCG doses at 85 °C water bath for 45 min. Inactivation was verified by colony formation assay.

### Pulmonary infection by *M. tuberculosis*

Infection doses of virulent (H$_{37}$R$_v$; ATCC 27,294) and avirulent (H$_{37}$Ra; ATCC 25,177) *M. tuberculosis* were prepared as previously described[19]. Mice were infected with 0.5 × 10$^6$ CFU per mouse of *M. tuberculosis* H$_{37}$Ra or 1 × 10$^4$ CFU per mouse of *M. tuberculosis* H$_{37}$Rv. *M. tuberculosis* H$_{37}$Rv burden was assessed at designated endpoints by plating the serial dilution of lung homogenates, BAL fluid or BAL cells disrupted with sterile water to release intracellular mycobacteria in triplicates.

### Bronchoalveolar and peritoneal lavages and mononuclear cell isolation

Mice were killed by exsanguination. In some instances, intravascular staining was carried out 3 min before exsanguination by injecting i.v. anti-CD45.2 antibody (clone 104) (BD Pharmingen)[40]. Cells in BAL and lung tissue were isolated as previously described[14,40]. The peritoneal cavity was lavaged as previously described[37]. Briefly, 3 ml of total wash solution (PBS containing 2 mM EDTA, 1 mM HEPES) was injected into the peritoneal cavity and the peritoneum was massaged gently for 30 s. Lavage fluid was collected with a pipette tip after making a small cut in the body wall. Spleen mononuclear cells were obtained as previously described[40]. BM cells were obtained by crushing the spine, femur and tibia bones in a mortar in PBS. BM cells were then filtered through a 40-μm basket filter (BD Biosciences). After lysing red blood cells, isolated cells were resuspended in either complete RPMI 1640 medium (RPMI 1640 supplemented with 10% FBS and 1% L-glutamine, with or without 1% penicillin/streptomycin) for ex vivo culture or in PBS for flow cytometry staining. When the BAL and lung cells were stimulated for intracellular cytokine staining (ICS) or cultured to measure cytokine/chemokine levels in culture supernatants, cells were resuspended in complete RPMI 1640 medium containing 2% FBS.

### Immunostaining, in situ cell proliferation and flow cytometry

Cell immunostaining and flow cytometry were performed as previously described[14]. Specifically, to determine alveolar macrophage activation levels and intracellular cytokine production, 250,000 mononuclear cells from BAL and 2 × 10$^6$ mononuclear cells from lung tissue were plated in a flat bottom 48-well plate and incubated for 3 h at 37 °C for macrophage adherence and tempering the irrelevant pro-inflammatory activities of freshly isolated AM. At the end of incubation, nonadherent

cells were washed off and fresh media was added with and without *M. tuberculosis* WCL at a concentration of 1.6 μg ml$^{-1}$. To determine levels of trained circulating monocytes and intracellular cytokine production, whole blood was collected into EDTA blood tubes (Sarstedt) via cardiac puncture and diluted with an equal volume of RPMI 1640. Diluted whole blood was aliquoted to 300 μl and incubated with or without *M. tuberculosis* WCL at a concentration of 1.6 μg ml$^{-1}$. GolgiPlug (5 mg ml$^{-1}$) (BD Biosciences) was added to BAL and lung cells and to diluted whole blood cultures 1 h after adding the stimulant. Cells were incubated for further 12–14 h. To determine activation levels of PM, 1 × 10$^6$ mononuclear cells were plated in a U-bottom 96-well plate with and without WCL at a concentration of 1.6 μg ml$^{-1}$(ref. [55]). GolgiPlug was added 1 h after adding the stimulant, and the cells were incubated for further 5 h. At the end of stimulation, BAL and lung cells were lifted from the wells by incubating in ice-cold FACS buffer (0.5% bovine serum albumin) (Sigma Aldrich) in the fridge for 15 min. Whole blood mononuclear cells were obtained for immunostaining after incubating with EDTA (Sigma-Aldrich) (30 μl of 20-mM EDTA per tube) for 15 min at room temperature and lysing red blood cells using BD Pharm Lyse (BD Biosciences). After staining with Aqua dead cell staining kit (ThermoFisher Scientific), cells were washed and blocked with anti-CD16/CD32 (clone 2.4G2) and then fixed and permeabilized with BD Cytofix/Cytoperm (BD Biosciences) according to the manufacturer's instructions. In some instances, BAL and lung mononuclear cells were immunostained without culturing for surface maker expression and immunophenotyping.

Fluorochrome-labeled mAbs used for the characterization of TII in macrophages and monocytes are listed in Supplementary Table 5. A panel of mAbs was used to identify multipotent progenitors polarized toward myeloid (MMP3) and lymphoid (MMP4) progenitors[13] (Supplementary Table 5). For ICS of T cells, BAL, lung and spleen cells were cultured in the presence of GolgiPlug (5 mg ml$^{-1}$ brefeldin A; BD Pharmingen) with or without a mixture of crude BCG and *M. tuberculosis* culture filtrate (2 μg per well)[54]. Stimulated cells were stained with cell surface antibodies, followed by fixation/permeabilization by using fixation/permeabilization solution kit (BD Biosciences) according to the manufacturer's instructions. Cells were then stained with anti-IFN-γ-APC mAb in Perm/Wash buffer (BD Biosciences) for 30 min on ice. Fluorochrome-labeled mAbs used for T cell surface and ICS were listed in Supplementary Table 5. For tetramer immunostaining, a tetramer for the immunodominant CD4 T cell peptide (FQDAYNAA-GGHNAVF) of Ag85B bound to the C57/Bl6 MHC class II allele (I-A(b)) conjugated to PE fluorochrome (Ag85B:H-2I-A$^b$) (NIH Tetramer Core, Atlanta, GA) was used[39].

For the determination of in situ AM proliferation, APC BrdU flow kit (552598; BD Biosciences) was used. Intranasal administration of BrdU was performed repeatedly at 5-week post-BCG immunization for a total of 9 d at a concentration of 0.5 mg per mouse in a total volume of 50 μl[14]. BrdU incorporated into DNA was then detected with an anti-BrdU-APC mAb (clone B44)[56].

Unless otherwise indicated, all mAbs and reagents were purchased from BD Biosciences. All antibodies were validated and titrated for optimal conditions before their applications in the experiments. Immunostained cells were processed according to the BD Biosciences instructions for flow cytometry and run on a BD LSR II or BD LSRFortessa flow cytometer using FACSDiva software. Data were analyzed using FlowJo software (version 10.8.1; Tree Star).

### Chemokine and cytokine quantification

To measure cytokines and chemokines produced by BAL AM, cells suspended in complete RPMI containing 2% FBS were plated in a 96-well flat bottom plate at 100,000 cells per well and rested for 3 h. After washing cells, fresh complete RPMI containing 2% FBS with and without 1.6 μg ml$^{-1}$ of *M. tuberculosis* lysates was added to wells and incubated for 12–14 h at 37 °C and 5% CO$_2$. Collected culture supernatants were frozen at −80 °C until measurement of protein levels. Whole blood

was diluted with an equal volume of RPMI and 400 μl was plated in a 48-well flat bottom plate and incubated for 12–14 h at 37 °C and 5% $CO_2$ with or without stimulation with 1.6 μg ml$^{-1}$ of *M. tuberculosis* lysates. Plasma was then collected and stored at −80 °C until the measurement of protein levels. Cytokines/chemokine levels were quantified by using MCYTOMAG-70K mouse chemokine and cytokine detection kit (Millipore Sigma, Oakville, ON) according to the manufacturer's instructions. Plates were read on a MagPix reader and concentrations of cytokines/chemokines were determined by using xPONENT software (Thermo Fisher Scientific).

### Ex vivo mycobacterium phagocytosis and killing assays

Mycobacterial phagocytic and killing assay of AM from BAL and pooled CD11b$^+$ and CD11c$^+$ cells purified from lung using microbeads (Miltenyi Biotec) was performed as described previously[19]. Bacterial killing capacity (% killing) was calculated and CFUs determined in the culture plates by using the formula: % killing = (number of bacterial CFU at 4 h – number of bacterial CFU at 24/48 h)/(number of bacterial CFU at 4 h) ×100. Moreover, mycobacterial phagocytic and killing capacity of AM was also evaluated using a recombinant BCG expressing red fluorescence (BCG-dsRed) and flow cytometry. Apoptosis and necrosis among mycobacterium-infected cells (dsRed$^+$) were analyzed using a flow cytometer after staining the cells with Aqua dead cell staining kit and Annexin V-APC staining kit (BD Biosciences) according to the manufacturer's instruction.

### Ex vivo alveolar macrophage antigen presentation to CD4 T cells

Ag85B-specific transgenic CD4 T cells purified from the spleen and lymph nodes of P25-Tg mice were labeled with Carboxyfluorescein succinimidyl ester (CFSE) (Invitrogen)[39]. CFSE-labeled CD4 T cells were cocultured with AM obtained from BAL at a ratio of 2:1 ($2×10^5$ T cells to $1×10^5$ AM) at 37 °C and 5% $CO_2$ for 96 h. Cells were then washed and immunostained for CD3 and CD4 cell surface markers and CFSE dilution in CD3$^+$CD4$^+$ cells was analyzed by flow cytometry.

### In vitro macrophage training and epigenetic re-modeling inhibition

AM obtained from BAL of naive mice resuspended in RPMI 1640 with % L-glutamine, with or without 1% penicillin/streptomycin, were plated (100,000 cells per well in 96-well or 250,000 cells per well in 48-well plates) and rested for 3 h at 37 °C and 5% $CO_2$. For innate immune training, the serum obtained from 5-week BCG-immunized or placebo mice was added to each well at 2% of the final volume and incubated for 24 h. After training, cells were washed once with culture medium (RPMI 1640 with % L-glutamine, with or without 1% penicillin/streptomycin and 2% FBS) and rested in the culture medium for 24 h or 3 d. The medium was changed once on day 2 for those rested for 3 d. After resting, cells were stimulated with or without *M. tuberculosis* WCL (1.6 μg ml$^{-1}$) for 12–14 h. Cytokine/chemokine proteins in culture supernatants were measured by Luminex. Separately, cells lifted from the well were immunostained and analyzed by flow cytometry. To investigate whether inhibition of epigenetic modifying enzymes affects innate immune training, inhibitors of histone methyltransferase, 5′-deoxy-5′-MTA (Sigma-Aldrich) at the concentration of 1 mM or histone acetyltransferase, EGCG (Sigma-Aldrich) at the concentration of 15 μM were added to the cells incubated in BCG serum for 24 h[48].

### In vivo depletion of T cells and IFN-γ neutralization

To deplete CD4 and CD8 T cells in vivo, mice were injected i.p. with 200 mg of anti-CD4 mAb (clone GK1.5) and 200 mg of anti-CD8 mAb (clone 2.43)[14]. To achieve continuous T cell depletion, 2 d following the initial injection, repeated doses of 100 mg of anti-CD4 mAb and 100 mg of anti-CD8 mAb were administered i.p. at a 7-d interval as needed. For IFN-γ neutralization, mice were injected i.p. with 200 μl of rabbit antimurine IFN-γ serum or normal control rabbit serum or with 500 μg monoclonal anti-IFN-γ antibody (Clone XMG1.2) (Bio X Cell) or isotype control mAb at 2-week post-BCG immunization. This treatment was repeated once every 5 d for a total of 3 weeks.

### In vivo alveolar macrophage labeling with PKH26-PCL

For selective labeling of AM, PKH26-Phagocytic Cell Linker (PCL; Sigma-Aldrich) was diluted in Diluent B (20 mM) as per the manufacturer's instruction and instilled intranasally into the lungs of mice (50 μl per mouse) 1 d before BCG immunization[42]. Control mice received PBS as control. The next day, one of the mice that received PKH26 or PBS was sacrificed to ensure selective labeling of AM.

### In vivo intestinal permeability measurement

Tracer FITC-labeled dextran (4 kDa; Sigma-Aldrich) was used to assess in vivo intestinal permeability[57].

### In vivo metabolite supplementation

For treating mice with distinct metabolites identified in BCG-immunized mice, 40-mM sodium butyrate and 1.25-mg ml$^{-1}$ L-carnitine (both from Sigma-Aldrich) were added to DW for 3 weeks[49]. Mice were then supplied with DW without metabolites for a week before the examination of AM and monocytes. The water intake was monitored every 2 d to ensure comparable water intake by mice provided with metabolite-supplemented water.

### Adoptive transfer of AM

AM obtained by BAL from 5 weeks of subcutaneous BCG-immunized and unimmunized mice were transferred via the intratracheal route[14]. Four hours after transfer, mice were challenged with *M. tuberculosis*.

### Cecal microbial transplantation

Cecum harvested from BCG-immunized and unimmunized mice were snap-frozen in liquid $N_2$ and stored at −80 °C until use[58]. Before cecal microbial transfer, mice were administered a broad-spectrum antibiotic cocktail containing vancomycin, neomycin, ampicillin and metronidazole at 0.5 g l$^{-1}$ concentration each (Sigma-Aldrich) in sterile DW (ad libitum) for 10 d. During antibiotic administration, body weight was monitored daily. The cecal matter for transplantation was prepared after dilution in 10 ml PBS and incubating at 37 °C for 30 min. After cessation of antibiotics administration, freshly prepared 200 μl of the cecal matter was administered by oral gavage once daily for 3 d. The microbiota was allowed to colonize for 5 weeks postgavage.

### Histological analysis and microscopy

To assess histological changes in the colon post-BCG, tissue sections were stained with H&E. To assess colonic barrier function, the tight junction components ZO-1 and occludin were visualized by immunohistochemistry. Briefly, after deparaffinization, rehydration and antigen-retrieval, tissue sections were treated with 3% hydrogen peroxide for 10 min to block endogenous peroxidase activity. Tissue sections were then blocked with 5% goat serum (Sigma) and were incubated overnight with either ZO-1 Polyclonal Antibody (1:50) (Invitrogen Life Technologies) or Occludin Rabbit Polyclonal Antibody (1:50), (Proteintech Group) at 4 °C. Signals were visualized using the HRP-DAB enzyme-chromatic reporter system (Dako EnVision+system HRP labeled polymer anti-rabbit from Dako and DAB from Sigma-Aldrich). To assess the mucin produced in the colon epithelial surfaces, tissue sections were stained for MUC2 on the Leica Bond Rx automated Stainer (Leica Biosystems) at MIRC Core histology facility, Department of Medicine, McMaster University. Briefly, tissue sections were pretreated with Leica Epitope retrieval 2 and then stained with rabbit-monoclonal (EPR23479-47) anti-MUC2 (Abcam) (1:3000) and Leica Bond Refine detection kit. Histological examination and scoring were independently verified by two researchers blinded to the treatment groups.

Cell morphology after training, resting and stimulating, the cells was studied by bright-field microscopy-EVOS cell imaging system (Thermo Fisher Scientific) and pictures were taken at ×5 and ×20 magnification. Images of representative micrographs were taken under a Zeiss Axio Imager 2 Research Microscope using AxioVision digital imaging software (Carl Zeiss Microscopy GmbH).

## Metabolic assay of AM

Real-time cell metabolism of AM was determined by using the Seahorse XF Glycolysis Stress Test Kit (Agilent Technologies) according to the manufacturer's instructions and as previously described[14]. Extracellular acidification rate (ECAR) corresponding to glycolysis and oxygen consumption rates corresponding to mitochondrial respirations (Oxidative phosphorylation) were assessed by using a Seahorse XFe24 Analyser (Agilent Technologies). Glycolysis was represented by ECAR after the addition of 10-mM glucose. Glycolytic capacity was represented by maximum ECAR following the addition of 1-µM oligomycin. The glycolytic reserve was represented by the difference between glycolytic capacity and glycolysis. Data were analyzed using Wave Desktop software version 2.6 (Agilent Technologies) and normalized to protein.

## RNA isolation and RNA-Seq

AMs were obtained via BAL. To ensure sufficient RNA for sequencing, two mice were pooled per sample. Triplicate samples were set up per group/condition. Unstimulated and stimulated samples were paired. Isolated AMs were cultured with or without stimulation with 1.6 µg ml⁻¹ of *M. tuberculosis* WCL. Following 12-h incubation, total cellular RNA was isolated using an RNeasy mini kit (QIAGEN) containing RNase-free DNase kit according to the manufacturer's instructions. RNA samples were stored at −80 °C until use. The quality of RNA was verified, and subsequent RNA sequencing was carried out by Farncombe Metagenomic Facility at McMaster University. RNA integrity was checked using the Agilent bioanalyzer. It was ensured that all RNA samples had a RIN (RNA integrity number) of 7.0 or greater for the best-quality libraries. The RNA was then subject to a polyA bead enrichment (NEBNext_PolyA_mRNA) process to enrich for mRNA with polyA tail, ensuring that high-quality RNA was obtained as any degraded transcripts would not be sequestered at this point. This step also ensured the removal of ribosomal RNA. The isolated mRNA was then converted to cDNA and made into a library containing adaptors and unique indexes using a ligation-based library prep kit (NEBNext_Ultra_II_Directional_RNA). The libraries were run on the bioanalyzer for a check of size and distribution and the concentration was checked using qPCR. The libraries were then pooled and run on two lanes of an Illumina HiSeq 1500 using onboard clustering in Rapid Mode. Furthermore, 25 M clusters were obtained per sample.

## Bacterial profiling by deep sequencing analysis of 16sRNA

Cecum was collected sterilely, and the V34 region of 16S rRNA gene was amplified by PCR[59,60]. In total, 50 ng of extracted DNA/sample was used as template with 1U of Taq, 1× buffer, 1.5 mM MgCl₂, 0.4 mg ml⁻¹ BSA, 0.2 mM dNTPs and 5 pmol each of 341F (CCTACGGGNGGCWGCAG) and 806R (GGACTACNVGGGTWTCTAAT) Illumina adapted primers. The reaction was carried out at 94 °C for 5 min, five cycles of 94 °C for 30 s, 47 °C for 30 s and 72 °C for 40 s, followed by 25 cycles of 94 °C for 30 s, 50 °C for 30 s and 72 °C for 40 s, with a final extension of 72 °C for 10 min. The resulting PCR products were visualized on a 1.5% agarose gel. Positive amplicons were normalized using the SequalPrep Normalization kit (Thermo Fisher Scientific) and sequenced on the Illumina MiSeq platform at the McMaster Genomics Facility.

## Metabolic profiling

Metabolic profiling by capillary electrophoresis coupled to time-of-flight mass spectrometry (CE-TOF-MS). All chemical standards and calibrants were purchased from Sigma-Aldrich, including analytical grade amino acids, acylcarnitines, ammonium acetate, ammonium bicarbonate, ammonium hydroxide, butylated hydroxylated toluene (BH₄), fatty acids, formic acid, organic acids, sodium hydroxide and recovery/internal standards—4-fluoro-L-phenylalanine (F-Phe), 3-chloro-L-tyrosine (Cl-Tyr) and D-glucose-¹³C₆ (¹³C-glucose). All LC−MS grade solvents, including acetonitrile, isopropanol, methanol and water were obtained from Caledon Laboratories. Calibrant solutions for all analytes were prepared by serial dilution of stock solutions (50 mM) in LC−MS grade water and stored refrigerated (4 °C).

Pall Nanosep Omega 3-kDa ultrafiltration tubes (VWR International) used for sample preparation were first prerinsed with deionized water to remove residual additives and background contamination. Tubes and filters were rinsed with 500 µl of deionized water, which were subsequently centrifuged at 10,000g for 5 min using Eppendorf 5430 (VWR International) to remove residual water. A mixture of 70:30 (MeOH:H₂O) containing 40 µM of F-Phe and Cl-Try, as well as 2 mM ¹³C-glucose, was prechilled on ice and used as internal or recovery standards for metabolite quantification.

All frozen cecum, colon and serum samples were slowly thawed on ice before sample pretreatment steps. In total, 100 µl of the above mixture containing 40 µM standards was added to the tubes containing preweighted cecum samples. The samples were then vortexed 10 min at room temperature at 3,000 rpm. Subsequently, the cecum samples were centrifuged at room temperature at 10,000g for 15 min and the supernatants were saved. The above-mentioned step was repeated, and the supernatants from both steps were combined. Subsequently, the supernatants were filtered through prerinsed ultrafiltration tubes at 10,000g for 15 min. Colon samples followed the similar sample pretreatment steps as cecum, but the ultrafiltration step was excluded. Serum samples were diluted fourfold. Briefly, 20 µl of the internal standard mixture containing 200 µM of F-Phe, Cl-Try and 2 mM ¹³C-glucose in water and 55 µl of water was added to the thawed 25-µl serum samples. The resulting mixtures were vortexed for 5 s at room temperature at 3,000 rpm. Subsequently, the mixtures were transferred to prerinsed ultrafiltration tubes and centrifuged at 10,000 g for 15 min. All extracted samples were frozen at −80 °C until analysis, and a 20 µl aliquot was transferred into a polypropylene vial for CE-TOF-MS analysis[61]. Lung samples were processed by adding 120 µl of chloroform and 200 µl of the mixture (50:50, methanol:water) containing 5 µM recovery standards (F-phenylalanine, choline-d9) to the vials containing ~5 mg of freeze-dried lung samples. After shaking for 15 min at room temperature at 3,000 rpm, samples were centrifuged at 3000g at 4 °C for 15 min to sediment protein at the bottom of the vial followed by a biphasic chloroform and water/methanol (top) layer. A fixed volume (150 µl) was collected from the upper aqueous layer into a new vial. The above-mentioned step was repeated, and the combined upper aqueous layer was collected into a new vial. Combined upper aqueous layers were then dried under a gentle stream of nitrogen gas at room temperature. Lung extracts were then stored at −80 °C and before analysis reconstituted in 50 µl of water/methanol (70:30) with 40-µM chloro-tyrosine and naphthalene monosulfonic acid (internal standards).

## Measurement of LPS in serum samples

Pierce Chromogenic Endotoxin Quant Kit from Thermo Fisher Scientific was used to measure the LPS levels in serum samples.

## Prediction of functional profiles of the bacterial communities

PICRUSt2 was used to infer the functional metagenomic contents of each sample (in the unrarefied OTU table). For quality assurance purposes, Nearest Sequenced Taxon Index scores were examined, and they were <0.15 for all samples. Differential abundance of the predicted metagenomes between the experimental groups was analyzed using the KO (K identifiers in KEGG pathway maps) metagenome unstratified predictions obtained from PICRUSt2. Metagenomes

with the predicted abundance of 0 in at least 10 of 11 samples were excluded from further analysis. Additionally, only metagenomes with the predicted abundance of at least ten in at least eight samples were included in further analysis. The differential abundance analysis was performed using limma package based on the limma–voom approach[62]. Specifically, the abundance values were rounded and then underwent VOOM transformation and TMM normalization before the differential analysis. Metagenomes exhibiting adjusted $P$ value of <0.05 were considered to be significantly regulated between the groups, and KEGG Orthology database (https://www.genome.jp/kegg/ko.html) was used to annotate and categorize these terms, using the second and the third levels of the database, including terms from BRITE hierarchy, and pathways.

## Quantification and statistical analysis

Statistical parameters including the exact value of $n$, the definition of center, dispersion and precision measures and statistical significance are reported in Figs. 1–8. The same samples were not repeatedly measured, and no data points were excluded from the analysis. All analyses were performed by using GraphPad Prism software (version 9.3.1, GraphPad Software). The confidence interval was set at 95% for statistical analysis. No statistical methods were used to predetermine sample sizes, but our sample sizes are similar to those reported in the previous publications[10,15,29]. Data distribution was assumed to be normal, but this was not formally tested. Data collection and analysis, except histological analysis, were not performed blind to the conditions of the experiments.

For RNAseq analysis, the reads were filtered by quality (at least 90% of the bases must have a quality score of 20 and higher) and then the remaining reads were aligned with mm10 (UCSC) reference using HISAT2. Next, the reads were counted by using HTSeq count. Genes, showing less than 10 counts in more than 30% of the samples per group, were removed using filterByExpr function in EdgeR package in R, resulting in 11,697 genes. Counts for these remaining genes were normalized with the TMM normalization method and then transformed using voom transformation. Differential expression between the groups of interest was examined using limma package in R. $P$ values were corrected with BH correction for multiple testings[63]; corrected values of <0.05 were considered to be significant. PCA was performed, and the results were visualized using rgl package in R (https://cran.r-project.org/web/package/rgl/index.html). Heatmaps were obtained using gplots package in R (https://cran.r-project.org/web/packages/gplots/index.html) and Volcano plots were obtained using limma package.

For microbiome analysis, reads were processed using DADA2. First, Cutadapt was used to filter and trim adaptor sequences and PCR primers from the raw reads with a minimum quality score of 30 and a minimum read length of 100 bp (ref. [64]). Sequence variants were then resolved from the trimmed raw reads using DADA2, an accurate sample inference pipeline from 16S amplicon data. DNA sequence reads were filtered and trimmed based on the quality of the reads for each Illumina run separately, error rates were learned and sequence variants were determined by DADA2. Sequence variant tables were merged to combine all information from separate Illumina runs. Bimeras were removed, and taxonomy was assigned using the SILVA database version 1.3.2. The most abundant bacterial taxa were recognized at the genus level. Comprehensive statistical analysis of the microbiome was performed using a web-based platform MicrobiomeAnalyst (www.microbiomeanalyst.ca)[65]. Data filtering was performed using the criteria that feature containing all zeros or appearing in only one sample was excluded from the analysis. Data were total sum scaled (count data are divided by the total number of reads in each sample). Community diversity was profiled as "alpha diversity" and "beta-diversity". Differentially abundance of operational taxonomical units at the genus level was identified using DESeq2. The statistical significance level was set at 5%. $P$ values were calculated and adjusted by the FDR.

For metabolite analysis, raw CE-TOF-MS data (d format) were processed using MassHunter Workstation Qualitative Analysis software (version B.06.00, Agilent Technologies, 2012). A comprehensive study of all detectable molecular features from the raw data was performed using MassHunter Molecular Feature Extractor, Molecular Formula Generator tools and an in-house compound database. Molecular features were extracted using a symmetric 10 ppm mass window, and all ions were annotated using their accurate mass (m/z), relative migration time (RMT) normalized to an internal standard (Cl-Tyr), and ionization mode of detection (p, positive; n, negative). RMTs are reported because they are an important parameter used to exclude redundant adducts and/or fragment ion peaks, which exhibit identical RMTs as the parent compound. Peak smoothing was performed using a quadratic/cubic Savitzky–Golay function (7 points) before peak integration. Peak areas and migration times for all molecular features were transferred to an Excel worksheet (Microsoft Office), and relative peak areas (RPA) for each unique molecular feature were saved as.csv file. RPAs were used for all statistical analyses. Pathway analysis (targeted) and multivariate data analysis, including PLS-DA, were performed using Metaboanalyst 5.0 (www.metaboanalyst.ca). In all cases, missing values were replaced with the default setting (one-fifth of the lowest detected value) and metabolomic data sets were (generalized) log-transformed and autoscaled unless otherwise stated.

### Reporting summary

Further information on research design is available in the Nature Portfolio Reporting Summary linked to this article.

## Data availability

RNA-seq data are deposited in the NCBI's GEO under accession code GSE213343 https://www.ncbi.nlm.nih.gov/geo/query/acc.cgi?acc=GSE213343. RNA microarray analysis of intranasally adenoviral (Ad)-vaccinated mice were previously published[14] and can be accessed under accession code GSE118512. Microbiome sequencing data and metabolomic data can be obtained from Mendeley Data at https://doi.org/10.17632/bvvfvz67z6.

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

## Acknowledgements

The work was supported by funds from the Canadian Institutes of Health Research (CIHR) Foundation Programme (FDN-154316 to Z.X.), National Sanitarium Association of Canada (to Z.X.), Canadian Foundation for Innovation, Genome Canada (to P.B.-M.), Ontario Government and McMaster University. We thank M. Tarnopolsky for allowing us access to his Seahorse XFe24 Analyzer and Z. Hmama for his kind provision of BCG-dsRed strain.

## Author contributions

M.J. and Z.X. conceived and designed the study. M.J., M.V.-S., S.A., J.A.G., A.K., M.R.D., Y.Y., S.J., A.Z., Z.K., M.S. and R.S. performed experiments. M.J., M.V.-S., M.S. and A.D.-G. analyzed data. P.B.-M. and W.I.K. supervised metabolic and microbiomic studies. M.J. and Z.X. wrote the paper.

## Competing interests

The authors declare no competing interests.

## Additional information

**Extended data** is available for this paper at https://doi.org/10.1038/s41590-022-01354-4.

**Correspondence and requests for materials** should be addressed to Zhou Xing.

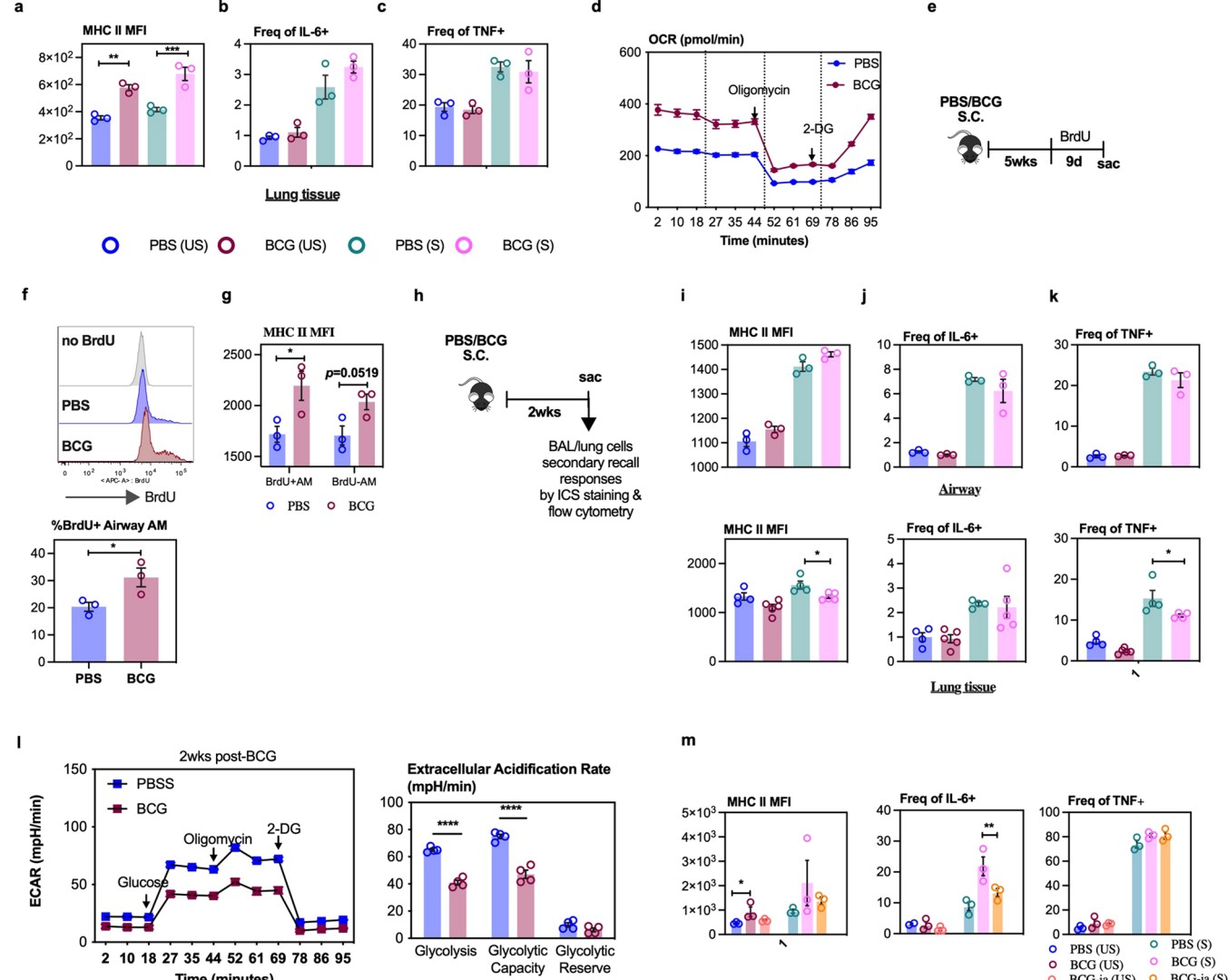

**Extended Data Fig. 1 | Characterization of innate memory phenotype of alveolar macrophages following s.c. BCG vaccination.** (**a**) MFI of MHC II expression on lung tissue AM with (S) or without (US) stimulation (**p = 0.0012, ***p = 0.0004). (**b & c**) Frequencies of IL-6- and TNF-producing lung tissue AM with (S) or without (US) stimulation. (**d**) Real-time oxygen consumption rate (OCR) in airway AM at 8-wks post-BCG vaccination. (**e**) Experimental schema. (**f & g**) Histograms of incorporation of BrdU by airway AM and frequencies of BrdU-incorporated airway AM (*p = 0.0475) (f) & measure of MFI of MHCII on BrdU + (*p = 0.0430) and BrdU- (p = 0.0519) airway AM (g). n = 3 mice/group. (**h**) Experimental schema. (**i, j & k**) Measure of MFI of MHCII on airway and lung tissue AM (*p = 0.0365) (i), frequencies of IL-6- (j) and TNF- (k) producing airway and

lung tissue AM (*p = 0.0249). n = 3 mice/group/airway, 4 mice/group/lung tissue. (**l**) Real-time extracellular acidification rate (ECAR) in airway AM and glycolysis, glycolytic capacity and glycolytic reserve in airway AM (****p = 4 mice/group. (**m**) Measure of MFI of MHCII expression on airway AM (*p = 0.0497) and frequencies of IL-6- (**p = 0.0082) and TNF-producing airway AM at 5-wks post-vaccination with viable BCG or inactivated BCG (BCG-ia) or PBS with (S) or without (US) re-stimulation. n = 3 mice/group. Data in **a-c** are representative of two independent experiments (n = 3 mice/group). Data in bar graphs are presented as mean ± SEM. One-way ANOVA was used for multiple comparison followed by Bonferroni's test for data in **a-c** and **i-k**, two-tailed t test for data in **f, g & l** and one-way ANOVA, followed by Dunnett's multiple comparison test for data in **m**.

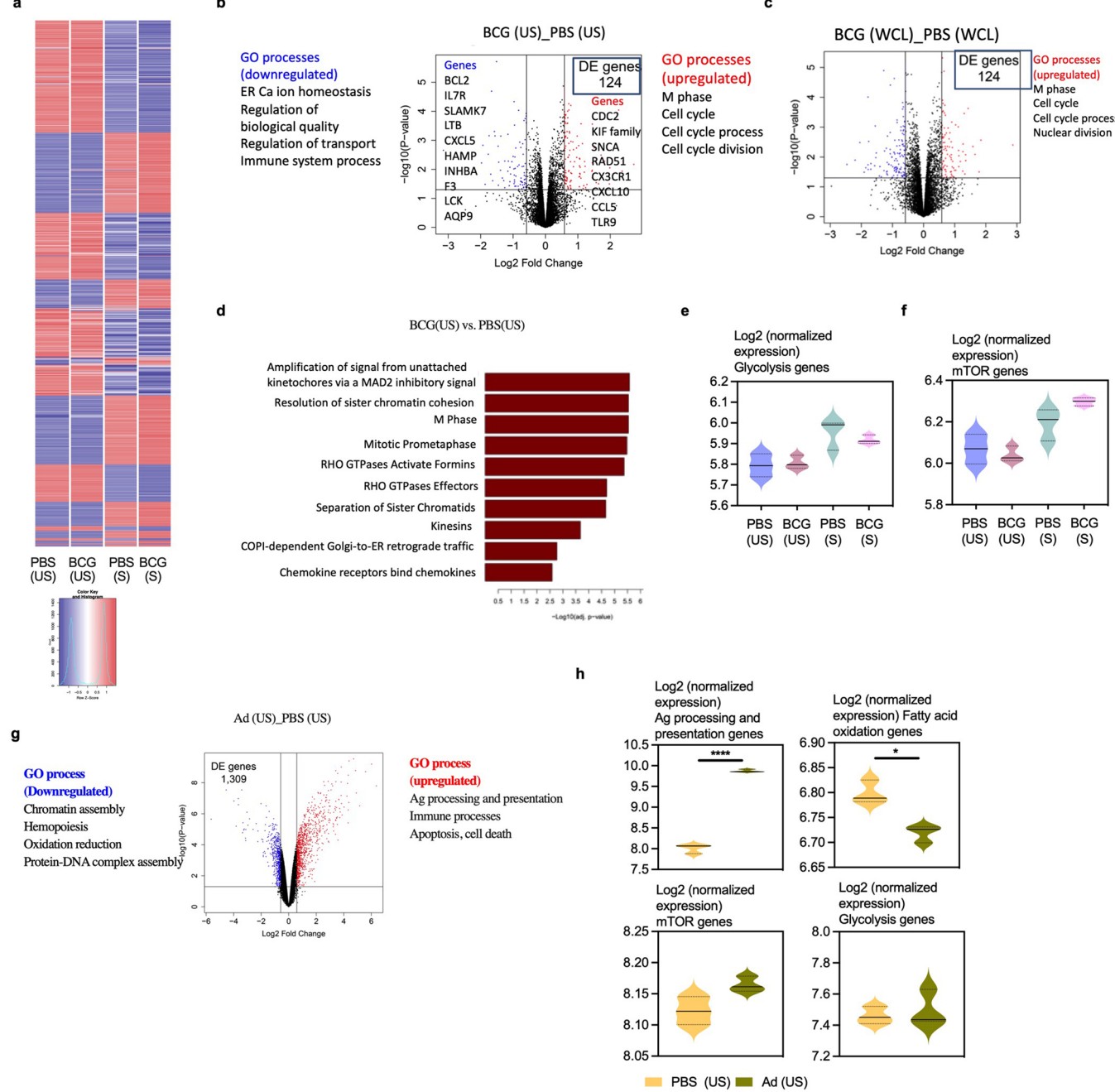

**Extended Data Fig. 2 | Transcriptional signatures of memory alveolar macrophages induced by s.c. BCG vaccination and respiratory mucosal viral-vectored vaccine.** (a) Heatmap of genes differentially expressed in airway AM by at least one of the comparisons. The difference was called when an absolute fold change was at least 1.5. Data are average levels of expression for each gene in each group. (**b**) Volcano plot for the comparison of airway AM from BCG-vaccinated vs. PBS animals without stimulation (US), with a threshold of absolute fold change set at 1.5. (**c**) Volcano plot for the comparison of airway AM from BCG-vaccinated vs. PBS animals with *M.tb* WCL stimulation (S), with a threshold of absolute fold change set at 1.5. (**d**) Pathways of the genes significantly up-regulated in un-stimulated (US) airway AM from BCG-vaccinated animals compared to those from PBS-treated animals. (**e & f**) Comparison of signature scores of the genes involved in glycolysis (**e**) and mTOR (**f**) in airway AM of BCG-vaccinated animals with those from control PBS animals, with (S)

or without (US) stimulation. (**g**) Volcano plot for the comparison of airway AM from intranasal adenoviral-vectored vaccine (Ad)-immunized vs. PBS control animals without stimulation (US) with a threshold of absolute fold change set at 1.5. (**h**) Comparison of signature scores of the genes involved in Ag processing/presentation (****$p = 6.15×10^{-7}$), fatty acid oxidation (*$p = 1.6×10^{-2}$), glycolysis and mTOR pathway in airway AM of intranasal adenoviral-vectored vaccine (Ad)-immunized animals with those from PBS control animals without (US) stimulation. Horizontal lines in violin plots denote medians and dotted lines denote lower and upper quartiles. Data in **a-g** are from experiments with 3 samples/group/condition. Vertical lines in volcano plots (**b, c & g**) denote the threshold of absolute fold change 1.5 and the horizontal line denotes adjusted $p$ value set at 0.05. Significant genes are marked in colors with up-regulated ones in red and down-regulated ones in blue. Adjusted $p$ values are presented for violin plots (**h**) and obtained using limma package and BH correction (see Methods).

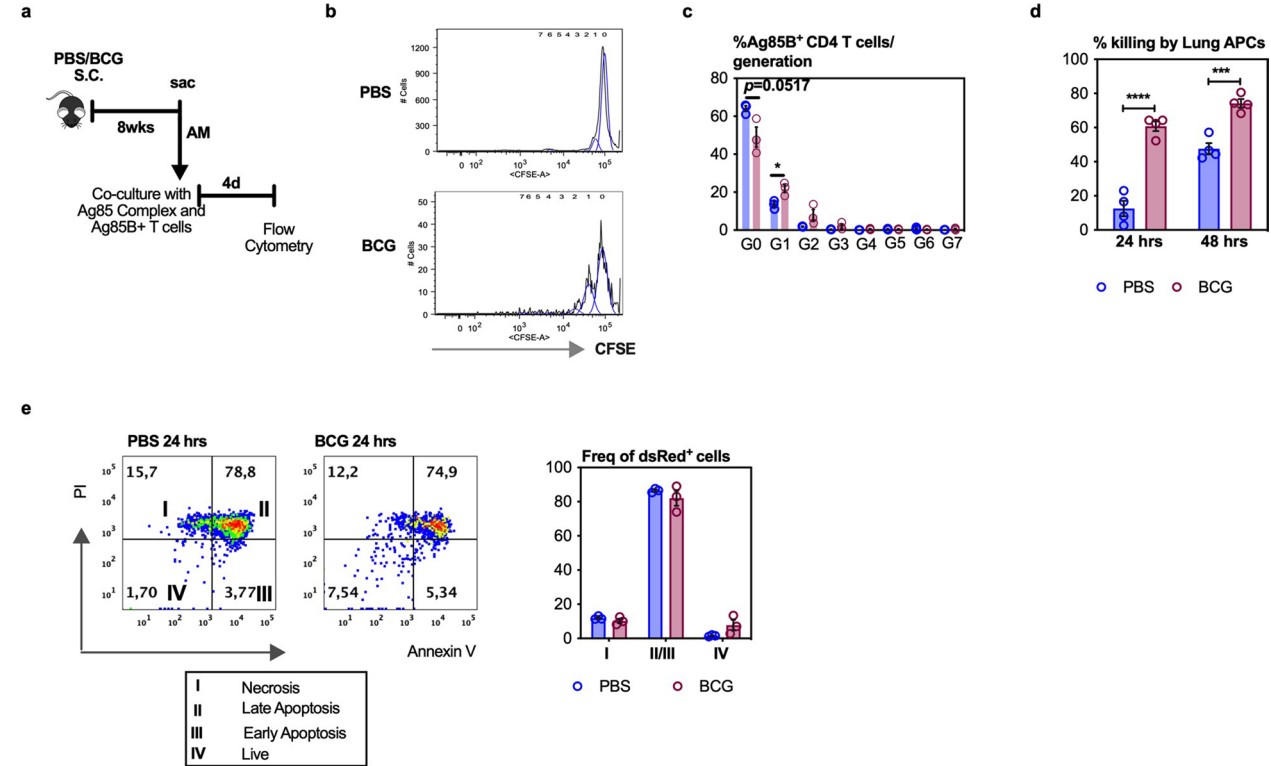

**Extended Data Fig. 3 | Characterization of antigen presentation and anti-microbial capabilities of memory alveolar macrophages.** (**a**) Experimental schema. **b**, Experimental schema. (**b**) Representative histograms of proliferating Ag85B-specific CD4 T cells in response to Ag85-loaded airway AM from BCG-vaccinated or PBS-treated animals, calibrated as the extent of CFSE label dilution. (**c**) Frequencies of Ag85B+ CD4 T cells in each generation (G) of proliferation. $p = 0.0517$ for G0, *$p = 0.0232$ for G1. n = 3 mice/group. (**d**) Percent of phagocytosed bacteria killed at 24 (****$p$ (***$p = 0.0006$) post-*M.tb* infection by lung tissue CD11C+CD11b+ antigen-presenting cells (APC) from BCG-vaccinated or PBS-treated animals. n = 4 mice/group. (**e**) Representative flowplots and frequencies of necrotic and apoptotic BCG-dsRed-infected airway AM of BCG-vaccinated and PBS-treated animals at 24 hrs post-infection. n = 3 mice/group. Data in bar graphs are presented as mean ± SEM. Statistical analysis was determined for data in **c & d** by two-tailed t test.

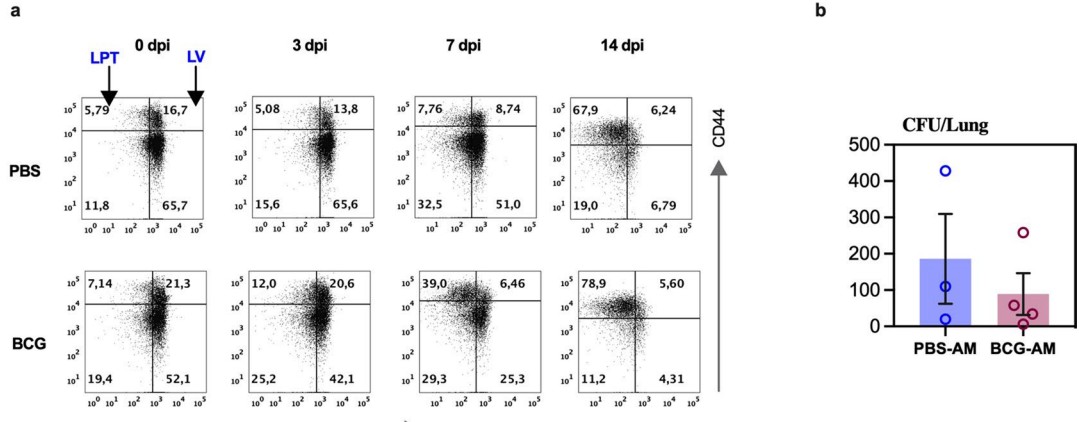

**Extended Data Fig. 4 | Kinetics of CD4 T cell responses in the lung tissue following pulmonary *M.tb* infection.** (**a**) Representative dotplots of lung mononuclear cells immunostained for surface markers CD4 and CD44 and gated to distinguish bona fide lung parenchymal T cells (LPT) from lung intravascular cells (LV) using CD45.2 antibody administered intravenously 3-5 min before sacrificing 5-wk PBS-treated or BCG-vaccinated mice. Data is from one experiment. (**b**) Mice were s.c. BCG-vaccinated or PBS-treated for 5-weeks. Airway AM harvested by bronchoalveolar lavage from PBS-treated (PBS-AM) and BCG-vaccinated (BCG-AM) were then adoptively transferred to naïve mice and challenged with *M.tb*. *M.tb* CFU in the lung at 3 days post-infection (n = 4 mice per group) were assessed. Data is from one experiment.

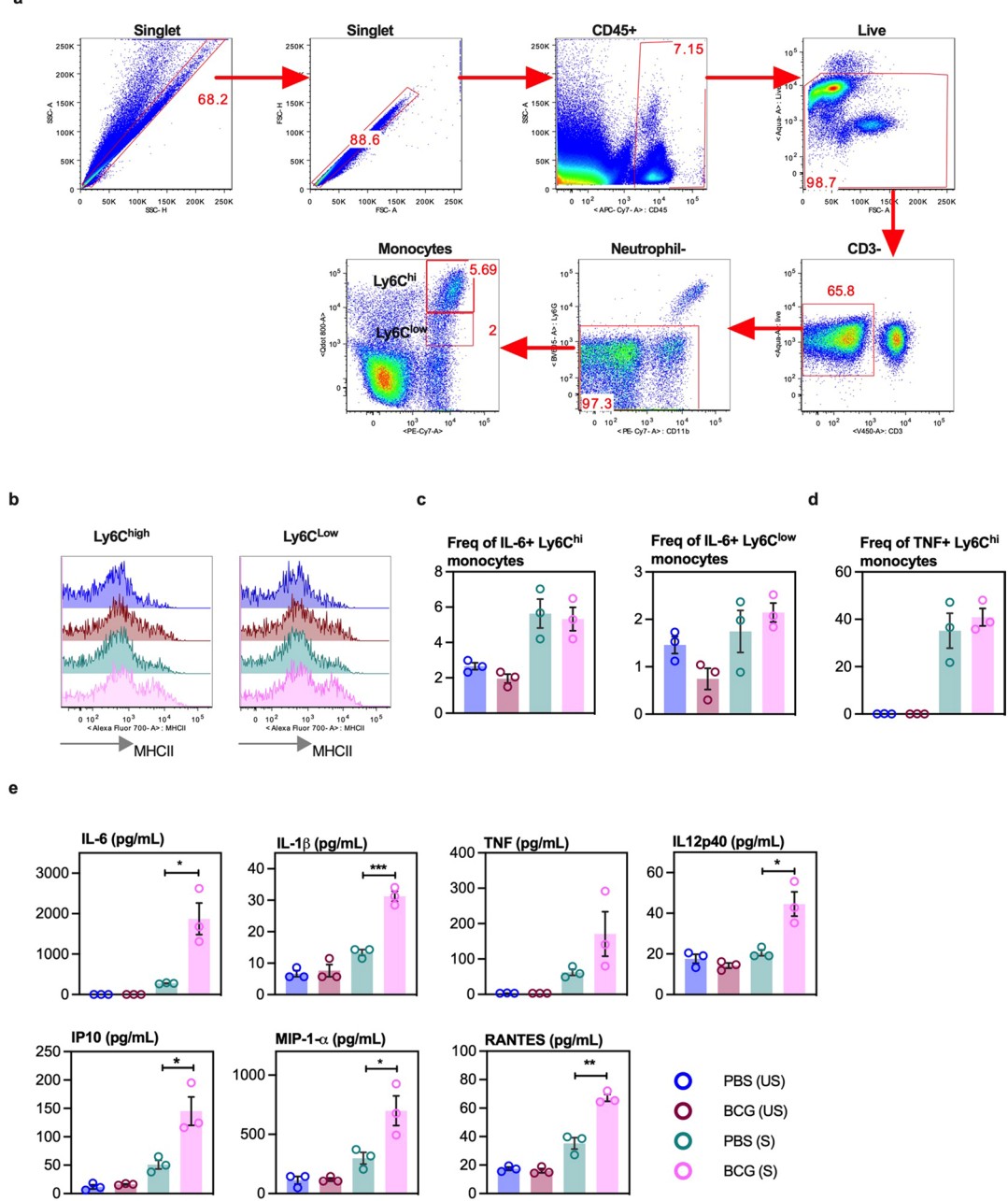

**Extended Data Fig. 5 | Immune characterization of circulating monocytes.**
(**a**) Gating strategy used for identification of circulating monocytes in the
peripheral blood. Live CD45+ cells were gated to remove CD3+ T cells (CD3-) and
CD11b+Ly6G+ neutrophils (neutrophil-). The remaining cell population was gated
to identify classical CD11b+Ly6C<sup>high</sup> and nonclassical CD11b+Ly6C<sup>low</sup> circulating
monocytes. (**b**) Representative histograms (n = 3 mice/group) comparing levels
of MHCII expression on Ly6C<sup>high</sup> and Ly6C<sup>low</sup> monocytes in the peripheral blood of
BCG-vaccinated and PBS-treated hosts cultured with (S) and without stimulation
(US). (**c & d**) Frequencies of IL-6-producing Ly6C<sup>high</sup> and Ly6C<sup>low</sup> monocytes and

frequencies of TNF-producing Ly6C<sup>high</sup> monocytes with (S) and without (US)
stimulation. n = 3 mice/group. (**e**) IL-6 (*$p$ = 0.015), IL-1β (***$p$ = 0.0007), TNF,
IL12p40 (*$p$ = 0.0173), IP10 (*$p$ = 0.0229), MIP-1-α (*$p$ = 0.0407) and RANTES
(**$p$ = 0.0024) protein contents in the plasma (pg/ml) of BCG-vaccinated or PBS
control animal-derived whole blood samples cultured with (S) and without (US)
stimulation. n = 3 mice/group. Data are presented as mean ± SEM, representative
of two independent experiments. Statistical analysis was determined by two-
tailed t test for data in **e**.

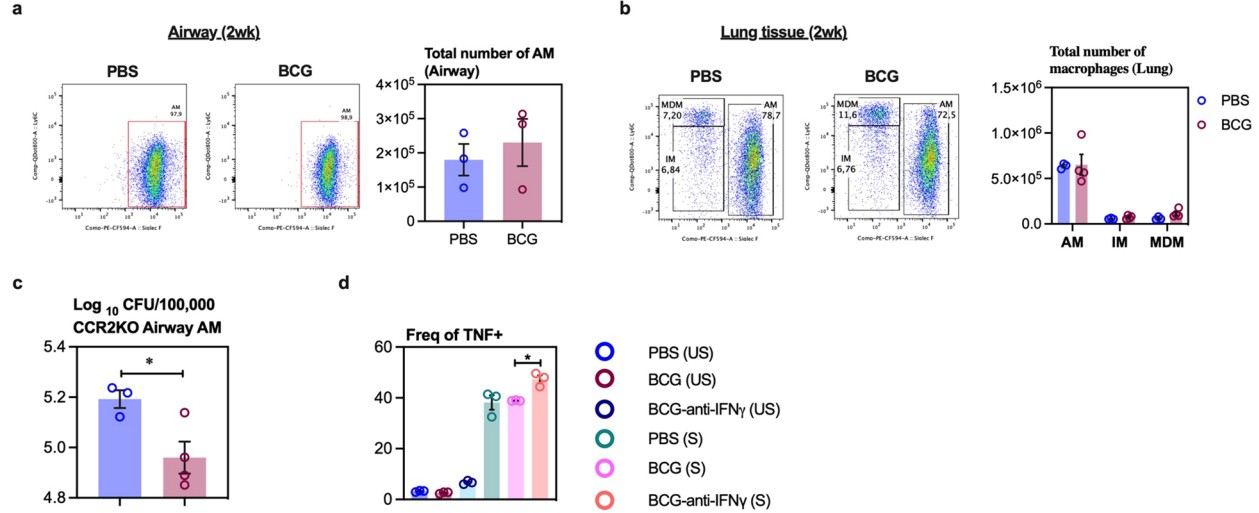

**Extended Data Fig. 6 | Macrophage subsets in the lung, *ex vivo M.tb* killing by alveolar macrophages from CCR2KO mice, and role of IFN-γ in induction of memory alveolar macrophages. (a & b)** Representative dotplots of SiglecF⁺Ly6C⁻ alveolar macrophages (AM) gated on CD64⁺CD24⁻ macrophages in the airway and lung tissue at 2-wk post-PBS treatment or BCG vaccination, & Monocyte-derived (MDM) and interstitial (IM) macrophages in lung tissue were identified as SiglecF⁻Ly6C⁺ and SiglecF⁻Ly6C⁻, respectively. Total numbers of different macrophage subsets in the airway and lung tissues are presented in the bar graphs. **(c)** Mycobacterial killing capacity of airway AM from 5-wk BCG⁻vaccinated or PBS-treated CCR2KO animals was assessed by *ex vivo* killing

assay. Total numbers of intracellular *M.tb* CFU were assessed at 24 hr after *ex vivo* infection. Each point represents biologically independent samples (*$p$ = 0.0351). n = 3 wells/PBS; n = 4 wells/BCG. Data are presented as mean ± SEM. **(d)** Frequencies of TNF-producing airway AM (unstimulated-US vs stimulated -S) from PBS control, BCG-vaccinated or BCG-vaccinated/IFNγ-depleted animals (*$p$ = 0.0031). n = 3 mice/group. Data are presented as mean ± SEM. and data in **d** are analyzed by one-way ANOVA for multiple comparison followed by Bonferroni's test. Statistical analysis for data in **c** was determined by two-tailed t test.

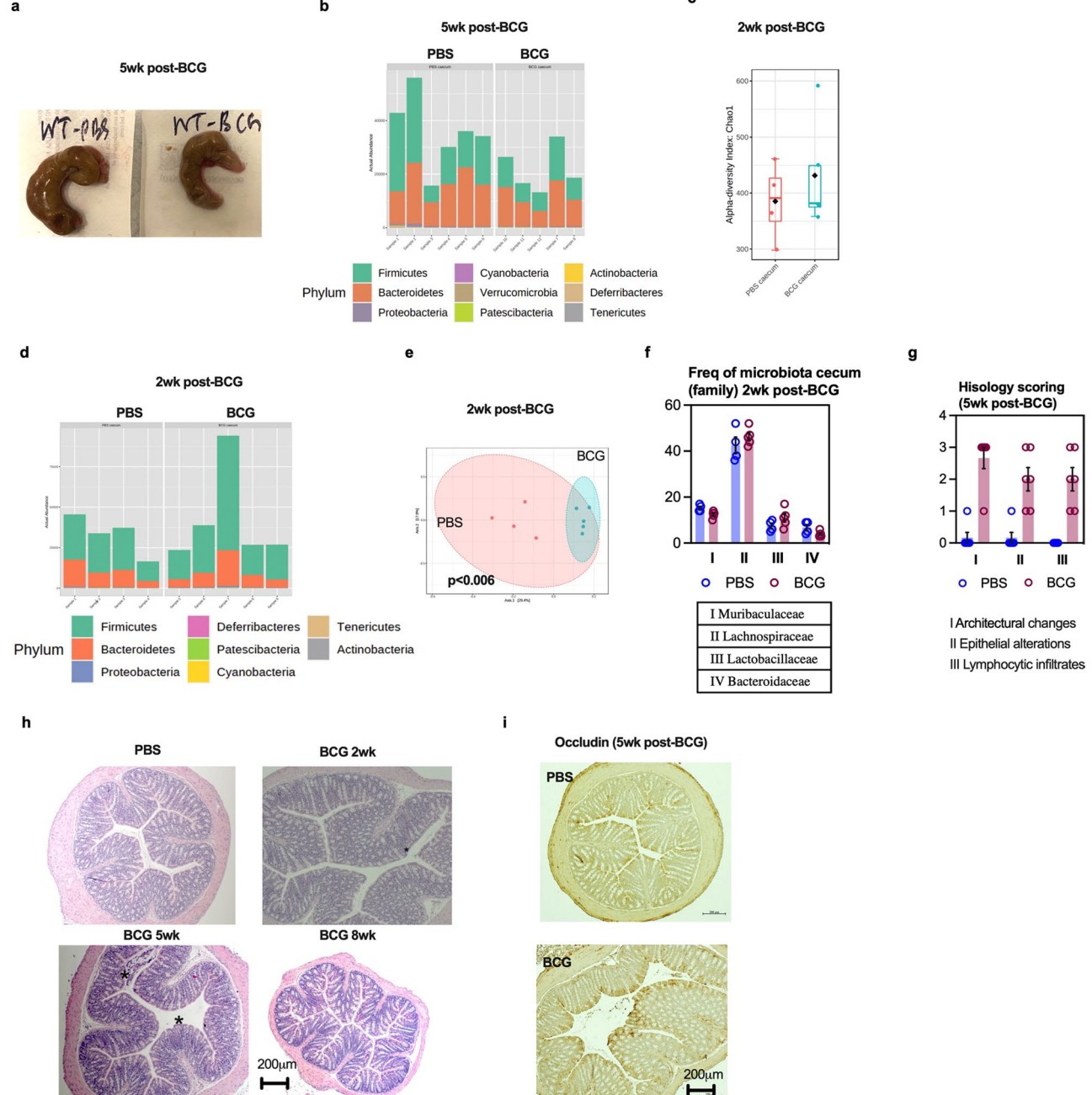

**Extended Data Fig. 7 | Altered gut microbiome, histomorphology and epithelium junction protein distribution following BCG vaccination.**
(**a**) Different gross appearance of the cecum from control PBS-treated and BCG-vaccinated animals. Representative of three independent experiments. n = 5 mice/group (**b**) Relative abundance profiles of top 9 bacterial phyla based on OTU data in 5-wk BCG-vaccinated (n = 5) and PBS control (n = 6) animals. Representative of two independent experiments. (**c**) Alpha diversity comparison based on operational taxonomic unit (OTU) richness in cecum microbiota using Chao1 diversity measure (t-test) between 2-wk BCG-vaccinated (n = 5) and control PBS (n = 4) animals. Horizontal lines in box plot denote medians and the length of the box denotes lower and upper quartile, and the whiskers denote minimum and maximum values. (**d**) Relative abundance profiles of top 8 bacterial phyla based on OTU data in 2-wk BCG-vaccinated and control PBS animals. n = 4 mice/

PBS, 5/BCG. (**e**) Beta diversity comparison of cecal microbial communities in 2-wk BCG-vaccinated and PBS animals using PCoA ordination method and Jenson-Shannon divergence distance method (PERMANOVA). (**f**) Bar graph comparing frequencies of top 4 abundant bacterial families in cecal microbiota of 2-wk BCG-vaccinated hosts with those in PBS control hosts. (**g**) Histologic scoring of architectural changes, epithelium alterations and inflammatory infiltrates of the colon from 5-wk BCG-vaccinated and PBS control animals. Representative of two independent experiments. n = 6 mice/group. (**h**) Histologic comparison of colon tissues from 2-wk, 5-wk, 8-wk BCG-vaccinated and PBS control hosts. Representative of n = 5 mice/2wk, 6 mice/5wk, 4 mice/8wk post-BCG. (**i**) Immunohistochemical staining of gut epithelium junction occludin protein in 5-wk BCG-vaccinated and PBS control animals. Representative of n = 6 mice/PBS, 5 mice/BCG. Data in bar graphs are presented as mean ± SEM.

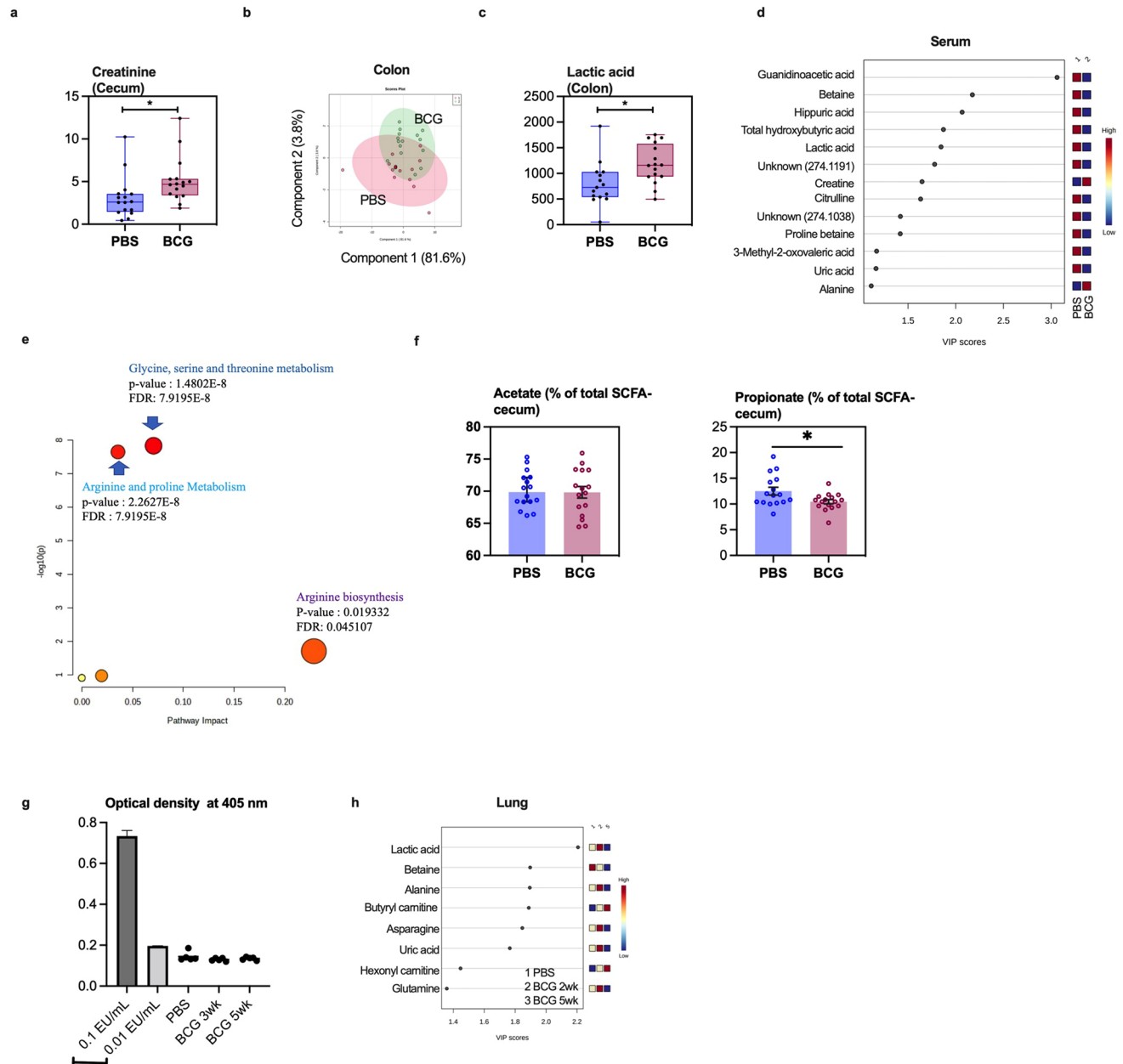

**Extended Data Fig. 8 | Altered microbial metabolic profile in gut, serum and lung, and undetectable LPS in serum following BCG vaccination.** (**a**) Comparison of creatinine levels in cecal tissue samples of 5-wk BCG-vaccinated and PBS animals. *p = 0.0330 (**b**) PLS-DA analysis on metabolic profiles in the colon tissue samples of 5-wk BCG-vaccinated and PBS animals. (**c**) Comparison of lactic acid levels in colon tissue samples of 5-wk BCG-vaccinated and PBS animals. *p = 0.0122 (**d**) Importance of the serum metabolites to the whole model identified by variable importance in projection (VIP) analysis. Colored boxes on the right denote comparative levels of serum metabolite between 5-wk BCG-vaccinated and PBS control serum samples. (**e**) Pathway analysis of serum metabolites of 5-wk BCG-vaccinated and PBS control animals with VIP score greater than 1.5 analyzed by Metaboanalyst 5.0, "pathway analysis-targeted" using Mus musculus (KEGG) pathway library. Three pathways were

identified as significant involving identified serum metabolites with VIP score greater than 1.5. (**f**) Proportions of acetate and propionate SCFAs calculated out of total concentration (mM/g) of SCFAs in cecal tissue samples of 5-wk BCG-vaccinated and PBS animals. *p = 0.0234 (**g**) LPS levels in the serum of 3-wk and 5-wk BCG-vaccinated and PBS control animals. n = 5 mice/group. (**h**) Lung tissue metabolites in 2-wk and 5-wk BCG vaccinated animals and PBS control animals (n = 10 mice/group) identified by variable importance in projection (VIP) analysis. Colored boxes on the right denote comparative levels of metabolite between BCG-vaccinated and control samples. Data in **a-f** is from two pooled experiments (n = 16 mice/group). Horizontal lines in box plots denote medians and the length of the box denotes lower and upper quartile, and the whiskers denote minimum and maximum values. Statistical analysis was determined by two-tailed t test for data in **a, c & f**.

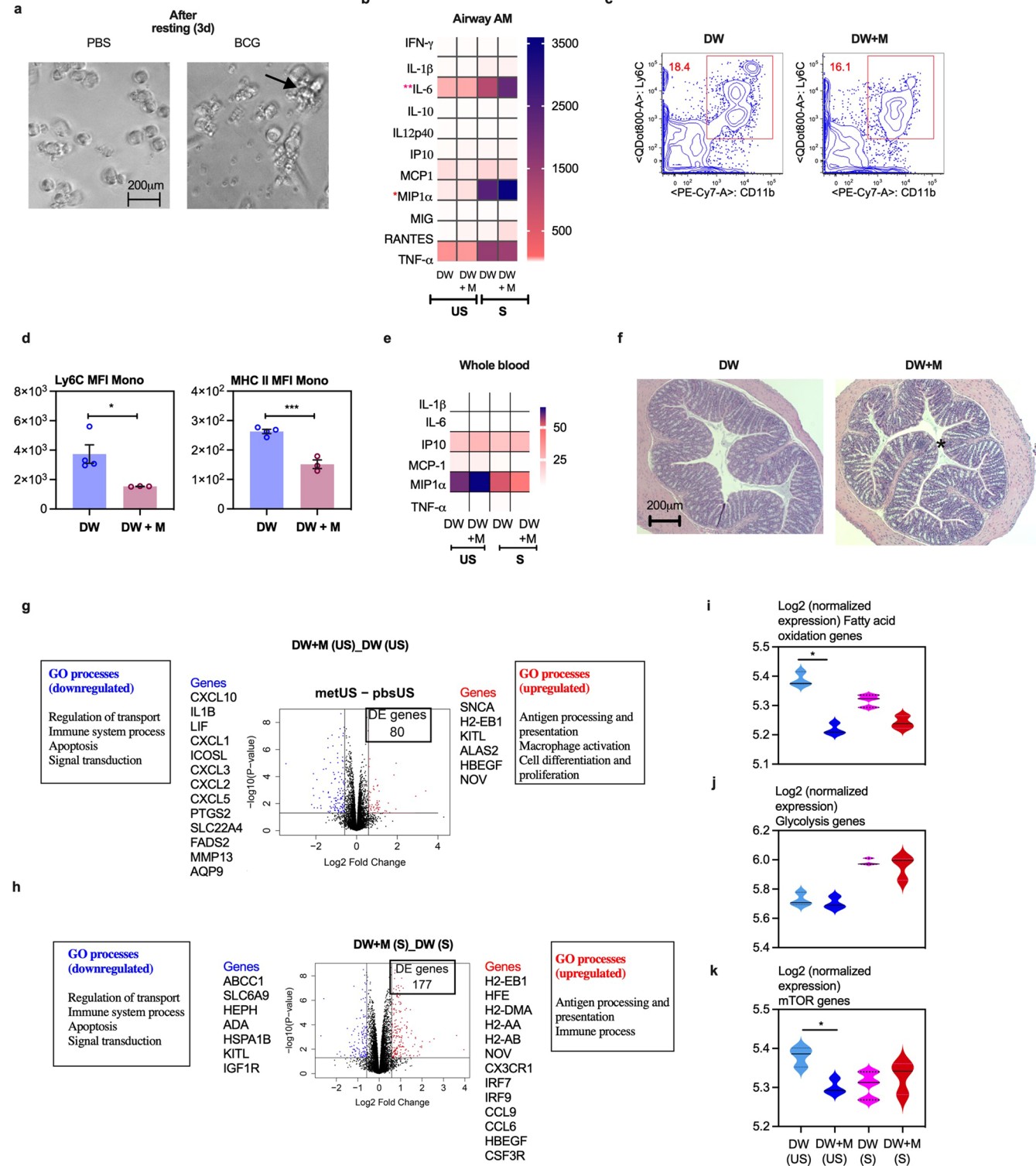

**Extended Data Fig. 9 | See next page for caption.**

**Extended Data Fig. 9 | Innate immune training of alveolar macrophages by circulating microbial metabolites in BCG-vaccinated animals. (a)** Representative bright-field microscopic images of AM after training with BCG-conditioned or control serum and resting before re-stimulation. The arrow points to the clusters and increased size and spreading of AM resulting from BCG-conditioned serum-mediated training and cell division. **(b)** Heatmap of cytokine/chemokine protein levels (geometric mean) in the culture supernatant of unstimulated (US) or re-stimulated (S) airway AM from animals on DW or DW + M. Asterisks denote significantly increased cytokine/chemokine production upon stimulation by airway AM of DW + M hosts. **(c)** Representative dotplots showing reduced classical CD11b⁺Ly6C^high monocytes in the circulation of animals on DW + M compared to those on DW. **(d)** MFI of Ly6C or MHCII expression on circulating monocytes of DW vs. DW + M animals (*$p$ = 0.0314, ***$p$ = 0.0007). Data are presented as mean ± SEM. n = 4 mice/DW, 3 mice/DW + M. **(e)** Heatmap of cytokine/chemokine protein levels (geometric mean) in the plasma of unstimulated (US) and stimulated (S) whole blood cultures of DW and DW + M animals. n = 4 mice/DW, 3 mice/DW + M. **(f)** Representative micrographs of H&E-stained colonic tissue sections from animals on DW and DW + M. **(g)** Volcano plot for the comparison of expression of select genes in airway AM from animals on DW + M vs. those on DW without stimulation (US), with a threshold of absolute fold change set at 1.5. **(h)** Volcano plot for the comparison of expression of select genes in airway AM from animals on DW-M vs. those on DW after stimulation (S), with a threshold of absolute fold change set at 1.5. **(i-k)** Comparison of signature scores of the genes involved in fatty acid oxidation (*$p$ = 0.0209), glycolysis and mTOR (*$p$ = 0.0209) in airway AM of animals on DW + M or DW, with (S) or without (US) stimulation. n = 3 mice/group. Horizontal lines in violin plots denote medians and dotted lines denote lower and upper quartiles. Data in **a-c** are representative of two independent experiments (n = 4 wells/condition). Vertical lines in volcano plots (**g & h**) denote the threshold of absolute fold change 1.5 and the horizontal line denotes adjusted p value set at 0.05. Significant genes are marked in colors with up-regulated ones in red and down-regulated ones in blue. Statistical analysis was determined by two-tailed t test for data in **d**. Adjusted p values are presented for violin plots (**i & k**) and obtained using limma package and BH correction (see Methods).

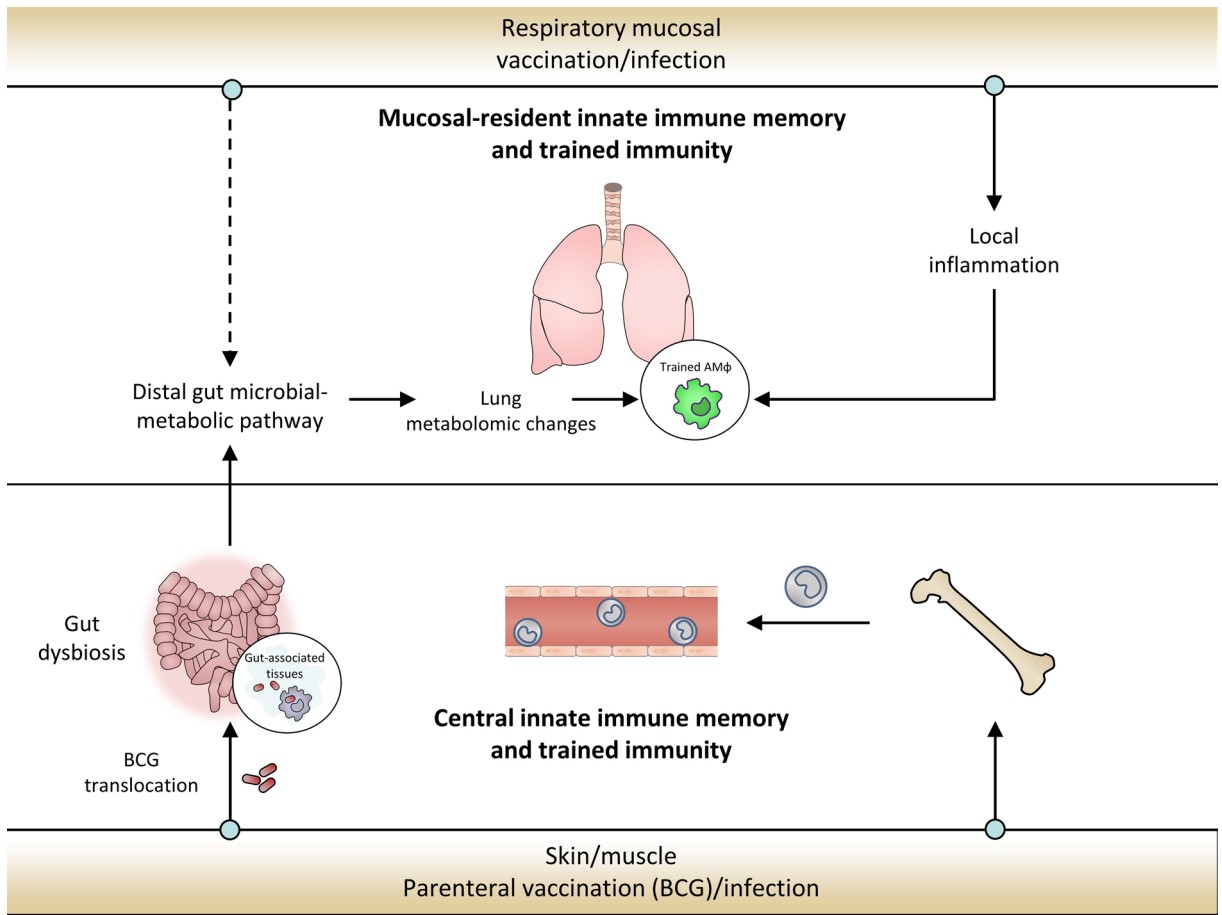

**Extended Data Fig. 10 | Illustration of the major findings from the current study in the context of what has previously been observed in the relevant field.** Recent evidence indicates that parenteral/systemic exposure to vaccine or microbe may lead to centrally induced innate immune memory and trained immunity (TII) via imprinting the myeloid progenitors in the bone marrow and subsequent releasing into the blood stream of trained monocytes. On the other hand, respiratory mucosal exposure to vaccine or microbe may induce mucosal tissue-resident memory macrophages and TII in the lung. There is also evidence that respiratory infection may cause gut microbiota dysbiosis. However, it has remained unclear whether parenteral vaccination could lead to gut dysbiosis and altered gut barrier function and metabolome, thus going on to induce mucosal-tissue resident memory macrophages and TII in a distal mucosal organ, the lung. Indeed, our current study finds that following s.c. BCG vaccination, initially BCG bacilli spread to the gut-associated tissues and lead to a time-dependent remarkable changes in gut microbiota and barrier permeability, and metabolomic changes not only in the gut but also in the serum and lung tissue. Thus, such a widespread immunological alert across a number of tissue sites threaded through a microbial metabolomic pathway ultimately results in a time-dependent induction of trained tissue-resident macrophages and innate immunity in the lung.

# Reporting Summary

## Statistics

For all statistical analyses, confirm that the following items are present in the figure legend, table legend, main text, or Methods section.

| n/a | Confirmed | |
|---|---|---|
| ☐ | ☒ | The exact sample size (*n*) for each experimental group/condition, given as a discrete number and unit of measurement |
| ☐ | ☒ | A statement on whether measurements were taken from distinct samples or whether the same sample was measured repeatedly |
| ☐ | ☒ | The statistical test(s) used AND whether they are one- or two-sided *Only common tests should be described solely by name; describe more complex techniques in the Methods section.* |
| ☐ | ☒ | A description of all covariates tested |
| ☐ | ☒ | A description of any assumptions or corrections, such as tests of normality and adjustment for multiple comparisons |
| ☐ | ☒ | A full description of the statistical parameters including central tendency (e.g. means) or other basic estimates (e.g. regression coefficient) AND variation (e.g. standard deviation) or associated estimates of uncertainty (e.g. confidence intervals) |
| ☐ | ☒ | For null hypothesis testing, the test statistic (e.g. *F*, *t*, *r*) with confidence intervals, effect sizes, degrees of freedom and *P* value noted *Give P values as exact values whenever suitable.* |
| ☒ | ☐ | For Bayesian analysis, information on the choice of priors and Markov chain Monte Carlo settings |
| ☒ | ☐ | For hierarchical and complex designs, identification of the appropriate level for tests and full reporting of outcomes |
| ☒ | ☐ | Estimates of effect sizes (e.g. Cohen's *d*, Pearson's *r*), indicating how they were calculated |

*Our web collection on statistics for biologists contains articles on many of the points above.*

## Software and code

Policy information about availability of computer code

| Data collection | Immunostained cell were collected using BD LSR II or BD LSRFortessa flow cytometer by FACSdiva (Version 8.0) Luminex plates were read on a MagPix reader Cell morphology was studied by bright field microscopy-EVOS cell imaging system Images of representative micrographs were taken under a Zeiss Axio Imager 2 Research Microscope using Zen digital imaging software (Version 2.3) Real-time cell metabolism of alveolar macrophages was determined by using the Seahorse XF Metabolic profiling by capillary electrophoresis coupled to time-of-flight mass spectrometry (CE-TOF-MS). |
|---|---|
| Data analysis | Flow cytometry data were analyzed by FlowJo (Version 10.8.1). Concentrations of cytokines/chemokines determined by using a xPONENT software (Version 4.2) Data of metabolic assay of alveolar macrophages were analyzed using Wave Desktop software version 2.6 Prediction of functional profiles of the bacterial communities by PICRUSt2 KEGG Orthology Database (https://www.genome.jp/kegg/ko.html) was used to annotate and categorize these terms, using the 2nd and the 3rd levels of the database, including terms from BRITE hierarchy, and pathways. RNAseq data were analyzed by Edge R (Version 3.34.0), limma package (Version 3.48.1), rgl package (Version 0.106.8) (https://cran.r-project.org/web/package/rgl/index.html), gplots (Version 3.1.1) (https://cran.r-project.org/web/packages/gplots/index.html). Microbiome data were analyzed by DADA2 and taxonomy was assigned using the SILVA database (Version 1.3.2) Statistical analysis of microbiome was performed using a web-based platform MicrobiomeAnalyst (www.microbiomeanalyst.ca) Metabolite raw CE-TOF-MS data were processed using Mass-Hunter Workstation Qualitative Analysis software (version B.06.00, Agilent Technologies, 2012) |

Pathway analysis (targeted) and multivariate data analysis, including partial least-squares-discriminant analysis (PLS-DA) were performed using Metaboanalyst 5.0 (www.metaboanalyst.ca).
Statistical tests were performed using Graphpad Prism (Version 9.3.1)

For manuscripts utilizing custom algorithms or software that are central to the research but not yet described in published literature, software must be made available to editors and reviewers. We strongly encourage code deposition in a community repository (e.g. GitHub). See the Nature Portfolio guidelines for submitting code & software for further information.

## Data

Policy information about availability of data

All manuscripts must include a data availability statement. This statement should provide the following information, where applicable:

- Accession codes, unique identifiers, or web links for publicly available datasets
- A description of any restrictions on data availability
- For clinical datasets or third party data, please ensure that the statement adheres to our policy

Raw RNA-seq data are available in the NCBI's GEO (http://www.ncbi.nlm.nih.gove/geo/) under accession No GSE213343. Raw data files for the RNA microarray analysis of intranasally adenoviral (Ad)-vaccinated mice were previously published (Yao, Y et al. Cell175, 1634-1650.e17 (2018) and can be obtained under accession No GSE118512. Microbiome sequencing data and metabolomic data can be obtained from Mendeley Data at https://doi.org/10.17632/bvvfvz67z6. The further data supporting the main findings are provided in accompanying Extended Data figures. Additional information is available via the correspondence author upon reasonable request.

## Human research participants

Policy information about studies involving human research participants and Sex and Gender in Research.

| Reporting on sex and gender | Use the terms sex (biological attribute) and gender (shaped by social and cultural circumstances) carefully in order to avoid confusing both terms. Indicate if findings apply to only one sex or gender; describe whether sex and gender were considered in study design whether sex and/or gender was determined based on self-reporting or assigned and methods used. Provide in the source data disaggregated sex and gender data where this information has been collected, and consent has been obtained for sharing of individual-level data; provide overall numbers in this Reporting Summary. Please state if this information has not been collected. Report sex- and gender-based analyses where performed, justify reasons for lack of sex- and gender-based analysis. |
|---|---|
| Population characteristics | Describe the covariate-relevant population characteristics of the human research participants (e.g. age, genotypic information, past and current diagnosis and treatment categories). If you filled out the behavioural & social sciences study design questions and have nothing to add here, write "See above." |
| Recruitment | Describe how participants were recruited. Outline any potential self-selection bias or other biases that may be present and how these are likely to impact results. |
| Ethics oversight | Identify the organization(s) that approved the study protocol. |

Note that full information on the approval of the study protocol must also be provided in the manuscript.

# Field-specific reporting

Please select the one below that is the best fit for your research. If you are not sure, read the appropriate sections before making your selection.

☒ Life sciences ☐ Behavioural & social sciences ☐ Ecological, evolutionary & environmental sciences

For a reference copy of the document with all sections, see nature.com/documents/nr-reporting-summary-flat.pdf

# Life sciences study design

All studies must disclose on these points even when the disclosure is negative.

| Sample size | No statistical methods were used to pre-determine sample sizes, but our sample sizes are similar to those reported in previous publications (Moorlag, S. J. C. F. M et al. Cell Rep. 31, 1076634 (2020); Guillon, A et al. JCI insight 5, (2020); Trompette, A et al. Immunity 48, 992-1005.e8 (2018) . Data distribution was assumed to be normal, but this was not formally tested. Data collection and analysis, except histological analysis, were not performed blind to the conditions of the experiments. |
|---|---|
| Data exclusions | no data points were excluded from analysis. |
| Replication | Most of the key experiments were repeated at least once more to establish the reproducibility of the findings, and such details were provided in individual figure legend. |

| Randomization | Animals were assigned to experimental groups at random. |
|---|---|
| Blinding | Since all data were included and objective analysis was carried out, data collection and analysis were not performed by researchers blinded to the conditions of the experiments except for histological analysis of the gut sections which was scored independently by two researchers blinded to the experimental groups. |

# Reporting for specific materials, systems and methods

We require information from authors about some types of materials, experimental systems and methods used in many studies. Here, indicate whether each material, system or method listed is relevant to your study. If you are not sure if a list item applies to your research, read the appropriate section before selecting a response.

## Materials & experimental systems

| n/a | Involved in the study |
|---|---|
| ☐ | ☒ Antibodies |
| ☒ | ☐ Eukaryotic cell lines |
| ☒ | ☐ Palaeontology and archaeology |
| ☐ | ☒ Animals and other organisms |
| ☒ | ☐ Clinical data |
| ☒ | ☐ Dual use research of concern |

## Methods

| n/a | Involved in the study |
|---|---|
| ☒ | ☐ ChIP-seq |
| ☐ | ☒ Flow cytometry |
| ☒ | ☐ MRI-based neuroimaging |

## Antibodies

| Antibodies used | Following antibodies were used. The details on their supplier names, catalog numbers, clone names etc are provided in Supplementary Table 5 of Methods section.<br><br>All antibodies for FACS were purchased from BD Bioscience unless otherwise specified: anti-CD45-APC-Cy7 (clone 30-FII), anti-CD11b-PE-Cy7 (clone Ml/70), anti-CDllc-APC (clone HL3), anti-MHC II-Alexa Flour 700 (clone MS/114.15.2; eBioscience), anti-CD3-V450 (clone 17A2), anti-Ly6C-Biotin (clone HKl.4; Biolegend), Streptavidin-Qdot800 (Invitrogen), anti-CD24-BV650 (clone Ml/69), anti-CD64-PE (clone X54-5/7.I; Biolegend), anti-Ly6G-BV605 (clone 1A8), anti-Siglec-F-PE-CF594 (clone ES0-2440), anti-CD11c-BV711 (clone HL3), anti-F4/80-e-Flour450 (clone BM8), anti-CD282(TLR2)-BV421 (clone CB225), anti-TNFalpha-PerCPCyS.5 (clone MP6-XT22 ) and anti-IL6-PE (clone MP5-20F3), anti-CD3e (clone 145-2CII, anti-Ly6G &Ly6C (cloneRB6-8C5), anti-CDllb (Clone Ml/70), anti-CD45R/B220 (clone RA3-6B2) and anti-Ly76 (erythroid cells) (clone-mTer-119), anti-c-kit-APC-Cy7 (clone 2B8), anti-Sca-l-PE-Cy7 (clone EB-161.7), anti-CD150-e-Fluor450 (clone mShadlS0), anti-CD48-PerCP-Efluor710 (clone HM48-l), anti-CD34-FITC (clone RAM34) and anti-Flt3-PE (clone A2F10.l), anti-CD3-V450 (clone 17A2), anti-CD4-PE-Cy7 (clone RM4-5), anti-CD8-APC-Cy7 (clone RAM34) and anti-Flt3-PE (clone A2F10.l), anti-CD3-V450 (clone 17A2), anti-CD4-PECy7 (clone RM4-5), anti-CD8-APC-Cy7 (clone 53-6.7), anti-gamma-APC (clone XMGl.2), and anti-CD44-V500 (clone IM7). |
|---|---|
| Validation | All antibodies are commercially available and validated by the manufacturer and/or titrated in the lab. Details of validation methods can be found in Supplementary Table 5. |

## Animals and other research organisms

Policy information about studies involving animals; ARRIVE guidelines recommended for reporting animal research, and Sex and Gender in Research

| Laboratory animals | Wild type female C57BL/6 mice were purchased from Charles River Laboratories (Saint Constant, QC, Canada) or the Jackson Laboratory (Bar Harbor, ME). Female Chemokine (C-C motif) receptor 2 (CCR2) (B6.129S4-Ccr2tm1Ifc/J), TLR2 (B6.129-Tlr2tm1Kir/J), TLR4 (B6(Cg)-Tlr4tm1.2Karp/J), NOD2 (B6. 129S1-Nod2tm1Flv/J) knock-out and P25 TCR-Tg transgenic mice containing CD4 T cells expressing Ag85B receptor (H2-Kb-Tcra,-Tcrb)P25Ktk/J) on a C57BL/6,  background were purchased from the Jackson Laboratory. All mice were 6-8-week of age upon arrival. Mice were housed in a specific pathogen-free level B facility or at the Biosafety Level 3 facility with ad libitum access to food and water, 12hrs light cycle, 50-60% humidity and at 20-250C room temperature at McMaster University, Hamilton. Aged-matched mice and housed in the same room were used in each experiment. Control mice were administered s.c. with PBS used for the preparation of BCG. Animals were assigned experimental groups at random. |
|---|---|
| Wild animals | no wild animals were used in this study. |
| Reporting on sex | the findings were applied only to female animals as only the female animals were used. |
| Field-collected samples | no field collected samples were used in this study. |
| Ethics oversight | All animal experiments were reviewed NS approved by the Animal Ethics Board (AREB) within McMaster University, Hamilton, Canada on behalf of the laboratory for Dr. Zhou Xing under the animal utilization protocol (AUP) number 210822. |

Note that full information on the approval of the study protocol must also be provided in the manuscript.

# Flow Cytometry

## Plots

Confirm that:

☒ The axis labels state the marker and fluorochrome used (e.g. CD4-FITC).

☒ The axis scales are clearly visible. Include numbers along axes only for bottom left plot of group (a 'group' is an analysis of identical markers).

☒ All plots are contour plots with outliers or pseudocolor plots.

☒ A numerical value for number of cells or percentage (with statistics) is provided.

## Methodology

**Sample preparation**

Mice were euthanized by exsanguination. In some instances, intravascular staining was carried out three minutes before exsanguination by injecting i.v. anti-CD45.2 antibody (clone 104) (BD Pharmingen, San Jose, CA). Cells in bronchoalveolar lavage and lung tissue were isolated. The peritoneal cavity was lavaged. Briefly, 3 ml of total wash solution (PBS containing 2mM EDTA, 1mM HEPES) was injected into the peritoneal cavity and the peritoneum was massaged gently for 30 sec. Lavage fluid was collected with a pipette tip after making small cut in the body wall. Spleen mononuclear cells were obtained. Bone marrow cells were obtained by crushing the spine, femur and tibia bones in a mortar in PBS. Bone marrow cells were then filtered through a 40 μm basket filter (BD Biosciences, San Jose, CA). After lysing red blood cell, isolated cells were resuspended in either complete RPMI 1640 medium (RPMI 1640 supplemented with 10% FBS and 1% L-glutamine, with or without 1% penicillin/streptomycin) for ex vivo culture or in PBS for flow cytometry staining. When the BAL and lung cells were stimulated for intracellular cytokine staining or cultured to measure cytokine/chemokine levels in culture supernatants, cells were resuspended in complete RPMI 1640 medium containing 2% FBS.

To determine alveolar macrophage activation levels and intracellular cytokine production, 250,000 mononuclear cells from BAL and 2x106 mononuclear cells from lung tissue were plated in flat-bottom 48-well plate and incubated for 3h at 37°C for macrophage adherence and tempering the irrelevant pro-inflammatory activities of freshly isolated alveolar macrophages. At the end of incubation non-adherent cells were washed off and fresh media was added with and without Mtb whole cell lysates (WCL) at a concentration of 1.6⊡g/ml. To determine levels of trained circulating monocytes and intracellular cytokine production, whole blood was collected into EDTA blood tubes (Sarstedt, Newton, NC) via cardiac puncture and diluted with equal volume of RPMI 1640. Diluted whole blood was aliquoted to 300⊡l and incubated with or without Mtb whole cell lysates at a concentration of 1.6⊡g/ml. GolgiPlug (5mg/ml) (BD Biosciences, San Jose, CA) was added to BAL and lung cells and to diluted whole blood cultures 1h after adding the stimulant. Cells were incubated for further 12-14h. To determine activation levels of peritoneal macrophages, 1x106 mononuclear cells were plated in U-bottom 96-well plate and with and without WCL at a concentration of 1.6⊡g/ml55. GolgiPlug was added 1h after adding the stimulant and the cells were incubated for further 5h. At the end of stimulation, BAL and lung cells were lifted from the wells by incubating in ice cold FACS buffer (0.5% bovine serum albumin) (Sigma Aldrich, Oakville, ON) in the fridge for 15 min. Whole blood mononuclear cells were obtained for immunostaining after incubating with EDTA (Sigma-Aldrich) (30⊡l of 20 mM EDTA/tube) for 15 min at room temperature and lysing red blood cells using BD Pharm Lyse™ (BD Biosciences, San Jose, CA). After staining with Aqua dead cell staining kit (ThermoFisher Scientific Waltham, MA), cells were washed and blocked with anti-CD16/CD32 (clone 2.4G2) and then fixed and permeabilized with BD Cytofix/Cytoperm (BD Biosciences, San Jose, CA) according to manufacturer's instructions.

**Instrument**

Sample were run on the BD LSR II or BD LSRFortessa flow cytometer at the McMaster Immunology Research Centre.

**Software**

Flowcytometry data was collected using FACSDiva (Version 8.0)
Flowcytometry data was analyzed using FlowJo (Version 10.8.1)

**Cell population abundance**

Abundance of CD11b+ and CD11c+ cells purified from lung using microbeads (Miltenyi Biotec, Auburn, CA) was 85-90%. Abundance of CD4 T cells from the spleen and lymph nodes of P25-Tg mice were 60-80%.

**Gating strategy**

Gating strategies used in the study were as per previously published studies (referred in the method section). Briefly, after removing debris, live cells were gated and immune cells were identified using CD45 surface marker. Cell populations were identified using FMO as the guide.

☒ Tick this box to confirm that a figure exemplifying the gating strategy is provided in the Supplementary Information.

