## [Peer Review File · Nature Immunology]

Peer Review Information

Journal: Nature Immunology

Manuscript Title: Parenteral BCG vaccine induces respiratory mucosal-resident memory macrophages and trained innate immunity via the gut-lung axis

Corresponding author name(s): Zhou Xing

Reviewer Comments & Decisions:

Decision Letter, initial version:
--

Subject: Decision on Nature Immunology submission NI-A32027A

Message: 21st Dec 2021

Dear Professor Xing,

Your Article, "Parenteral BCG vaccine induces respiratory mucosal-resident memory macrophages and trained innate immunity via the gut-lung axis" has now been seen by 2 referees. You will see from their comments copied below that while they find your work of considerable potential interest, they have raised quite substantial concerns that must be addressed. In light of these comments, we cannot accept the manuscript for publication, but would be very interested in considering a revised version that addresses these serious concerns.

While there are a number of potentially important issues raised, I suspect an absolutely key one for both referees will be to strengthen the connection with the microbiota i.e. it seems unprecedented that the BCG should be altering this. Other potential mechanisms by which the BCG is operating to train the AMs need to be ruled out.

We hope you will find the referees' comments useful as you decide how to proceed. If you wish to submit a substantially revised manuscript, please bear in mind that we will be reluctant to approach the referees again in the absence of major revisions. Please do not hesitate to get in touch if you would like to discuss these issues further.

If you choose to revise your manuscript taking into account all reviewer and editor comments, please highlight all changes in the manuscript text file.

* If you have not done so already please begin to revise your manuscript so that it conforms to our Article format instructions at <http://www.nature.com/ni/authors/index.html>. Refer also to any guidelines provided in this letter.

The Reporting Summary can be found here:
<https://www.nature.com/documents/nr-reporting-summary.pdf>

You may use the link below to submit your revised manuscript and related files:
[REDACTED]

If you wish to submit a suitably revised manuscript we would hope to receive it within 6 months. If you cannot send it within this time, please let us know. We will be happy to consider your revision so long as nothing similar has been accepted for publication at Nature Immunology or published elsewhere.

Nature Immunology is committed to improving transparency in authorship. As part of our efforts in this direction, we are now requesting that all authors identified as 'corresponding author' on published papers create and link their Open Researcher and Contributor Identifier (ORCID) with their account on the Manuscript Tracking System (MTS), prior to

acceptance. ORCID helps the scientific community achieve unambiguous attribution of all scholarly contributions. You can create and link your ORCID from the home page of the MTS by clicking on 'Modify my Springer Nature account'. For more information please visit www.springernature.com/orcid.

Thank you for the opportunity to review your work.

Sincerely,

Zoltan Fehervari, Ph.D.
Senior Editor
Nature Immunology

The Macmillan Building
4 Crinan Street
Tel: 212-726-9207
Fax: 212-696-9752
z.fehervari@nature.com

Referee expertise:

Referee #1:

Referee #2:

Referee #3:

Reviewers' Comments:

Reviewer #2:

Remarks to the Author:

This an innovative study that assessed the mechanisms through which parenteral administration of BCG induces trained immunity. The authors show that induction of trained immunity is both mediated at central level in the bone marrow, as well as in the periphery at the level of mucosal macrophages. Moreover, the authors dissect a new mechanism through which a gut-lung axis is mediated by metabolites released by changes in microbiome induced by BCG vaccination. The experiments are well performed and the manuscript is clearly written.

Comments:

1. The authors present extensive arguments that bone marrow-derived trained monocytes and T-cells/IFN γ are not needed for the induction of the alveolar macrophages (AMs) with a trained immunity phenotype. Are trained monocytes and T-cells/IFN γ however important for the protection given by BCG vaccination? A recent study by Kaufmann et al showed an important role of BCG training at bone marrow level: what is the relative role of trained

myelopoiesis and trained AMs for the protection induced by BCG vaccination?

2. BCG vaccination induces also release of microbial structures in the circulation, such as peptidoglycans and MDP that can induce trained immunity at distant sites. Could this release play a role in the induction of trained immunity in AMs?

3. The changes in cecum size, gut permeability, microbiome richness and composition by BCG vaccination is remarkable. What are the potential mechanisms through which distant BCG vaccination can induce such strong effects in the architecture of the gut? Is there BCG spread in the gut mucosa? (this may explain the delay needed for the observed effects)

4. The changes on gut histology and function by BCG vaccination may seem deleterious: increased gut permeability, irregular villi, epithelial disruption, etc. Such changes are also described in low-grade chronic inflammatory processes (e.g. HIV infection in humans) and are associated with cardio-vascular and metabolic complications. Would these changes induced by BCG be expected to lead to similar long-term deleterious effects?

5. It is not only metabolites, but also microbial structures (LPS, bacterial DNA, peptidoglycans, etc) that can translocate from gut to blood in cases of increased gut permeability. For example, were there higher LPS circulation concentrations in mice vaccinated with BCG?

6. Interestingly, also the mice colonized with microbiota from BCG-vaccinated animals displayed some of the epithelial dysplasia and disruption. This opens the possibility of leakage in the circulation of microbial products: both cellular structures (cell wall components-e.g. LPS, bacterial DNA) and metabolites. The same question here: is circulating LPS (or other bacterial structures) concentration changed?

7. The authors elegantly show that a mix of L-carnitine and butyrate hydrochloride in drinking water of the animals, that maintain high SCFAs in the blood, also induced trained immunity in the AMs. Does this mix of L-carnitine and butyrate hydrochloride induce histological changes in the gut mucosa?

8. A number of recent studies (including in non-human primates) showed stronger effects of i.v. BCG compared to s.c. BCG for protection against *M. tuberculosis*. Can the authors speculate whether trained immunity in AMs would be stronger induced by i.v. BCG?

Reviewer #3:

Remarks to the Author:

I wish to thank you for offering me the possibility to read and comment on this interesting study by Jeyanathan et al., which investigates the mechanisms of parenteral BCG vaccine to induce trained mucosal resident memory macrophages via the gut-lung axis.

A. Summary of the key results

The key results could be summarized as follows:

- the training of resident alveolar macrophages requires long (5 weeks) in vivo proliferation of BCG bacilli after vaccination.
- Trained alveolar macrophages are characterized by low phagocytosis ability, pro-inflammatory cytokine activity, and high glycolytic activity.
- BCG vaccine increases the *M. tb* control during the early stages of infection, and CD4 CD8 T cells do not seem involved in this protection.
- The training of alveolar macrophages is independent of circulating monocytes and T cells
- BCG vaccine induces a loss of diversity in the gut microbiome, associated with increased butyrate concentration in the cecum contents.
- The in vivo transfer of the BCG-altered gut microbiota reproduces some of the features

of trained AM, and serum for BCG-vaccinated mice can alter the in vitro production of cytokines by macrophages.

B. Originality and significance: if not novel, please include reference

The concept of trained immunity is rapidly gaining in importance (DOI 10.1038/s41590-020-00845-6), and many data explaining the potentially life-long immunological scar left by BCG and inflammation, in general, have been produced in the systemic compartment such as bone marrow progenitors (DOI 10.1016/j.chom.2020.05.014) and even in gametes (DOI 10.1038/s41590-021-01052-7). However, the recognition of the roles of resident macrophages (DOI 10.1016/j.immuni.2016.02.024) and resident memory T cells (DOI 10.1038/nri.2015.3), which bone-marrow-derived progenitors do not replace, raises the question of the mechanism of trained immunity in peripheral tissues. While the role of the anatomical niche in the imprinting of macrophages has been recently reviewed (DOI 10.1038/s41590-020-00849-2), the role of systemic factors in the training of resident AMs has not been thoroughly investigated to date.

In this setting, the demonstration that gut microbial metabolites can regulate remote immune cells is now well established and not entirely new (DOI 10.1038/s41579-020-0438-4, DOI 10.1038/s41586-020-03116-4), notably for the gut-lung axis (DOI 10.1016/j.celrep.2020.02.013 and doi 10.1038/nm.4194). However, I am unaware of a demonstration that the BCG vaccine alters gut microbiota and that gut microbiota is a component of trained immunity. So as a summary, I think this study is original in the setting of trained resident macrophages.

The significance of the study could be potentially high since gut microbiota can be easily accessed as a biomarker and can be modulated for those aiming at reproducing trained AM.

C. Data & methodology: validity of approach, quality of data, quality of presentation

Figure 3N. The deletion of CD4 CD8 T cells increases the control of M.Tb. Is this result also observed in PBS-treated mice? This is an essential control since it suggests that CD4 CD8 T cells are deleterious after the BGG vaccine.

D. Appropriate use of statistics and treatment of uncertainties

The statistics are sounds, and the used tests are reported. I would recommend representing the Figure by individual plots rather than histograms.

E. Conclusions: robustness, validity, reliability

The experiments are all repeated and statistically significant. My main question here concerns the validity of the mice models. Indeed, the observation of gut dysbiosis in BCG-vaccinated mice is novel, but it is an intriguing result since a decrease in the alpha diversity (Fig 5A) and the histological finding (Fig 5E) demonstrated that BCG induced colitis in mice. This effect has not been described in humans, yet BCG has been used for decades in clinical practice to the best of my knowledge. I am thus concerned by the extrapolation to humans of the results gained in this model and the conclusion that BCG-induced colitis induces training (and not only BCG-vaccine). To reinforce this message, the authors may consider demonstrating that the BCG vaccine-induced training of ResAM is altered in germ-free mice.

F. Suggested improvements: experiments, data for possible revision

In my opinion, two sets of experiments need to be performed to demonstrate the

conclusion of the authors definitively:

1- The main message of the manuscript is the training of resident AM. Since monocyte-derived AM can replace resident AM and gain most of their phenotype (DOI 10.1016/j.immuni.2020.02.015), the phenotype of res AM developed in a viral mice model (Ly6G-CD11b-CD11chiCD64hiSiglec-Fhi) can be misleading here. The only result demonstrating the impact of resAM is the CCR2^{-/-} mice (Figure 4H). The demonstration that BCG-vaccine also increases MHC-II in CCR2^{-/-} mice is encouraging, but it would be essential to reinforce this data, and notably to demonstrate that ResAM of CCR2^{-/-} mice receive the complete training. The authors may consider to

- Provide the total number of ResAM and Mo-AM in the several conditions (is the deletion of monocyte-derived AM changing the proliferation of ResAM ?).
- Demonstrate that ResAM in CCR2^{-/-} also has cytokine and metabolic reprogramming.
- Demonstrate that CCR2^{-/-} mice are still protected against M Tb after BCG vaccination

2- The second main message is the role of metabolites derived from gut microbiota in the training of ResAM. The specific role of the proposed metabolites (butyrate and L-carnitine) reported is not specific to BCG (DOI 10.1016/j.celrep.2020.02.013), and the reported effects of treatment with these metabolic (pro-inflammatory IL6 / MIP1a production) are relatively non-specific. The increased concentration of metabolites is only described in the cecum. To reinforce this message, and respond to the question of the specificity of BCG vaccine, the authors may consider to

- Compare the transcriptomic activity of ResAM in mice treated for three weeks with butyrate and L-carnitine, with resAM of BCG vaccinated mice. This would reinforce the message that these metabolites are specific and play an essential role in the BCG-induced training of AM
- Compare the transcriptomic activity of trained ResAM of BCG vaccinated mice with their results gained in these cells after viral pneumonia (Yao et al. Cell 2018) or after bacterial pneumonia (Roquilly et al. Nature Immunol 2020)
- Demonstrate that the concentrations of butyrate and L-carnitine are increased in the lungs after BCG, and provide the timing of this increase. Potentially (maybe not possible) demonstrate by in vivo tracking of tagged-metabolites injected in the guts that their passage to the lungs is possible and increased.

Minor propositions

- Figure 3C and 3E. Is it possible to test the resistance to M. Tb of BCG-vaccinated mice after depletion of ResAM ? Given the results in Fig 3H-J (drastic increase in CD4 T cells) and Fig 3N (positive effects of T cell depletion), it is possible to conclude that T cells are more critical than macrophages here.
- Figure 3N. The deletion of CD4 CD8 T cells increases the control of M.Tb. Is this result also observed in PBS-treated mice? This is an essential control since it suggests that CD4 CD8 T cells are deleterious after the BGG vaccine.
- Figure 5. Did the authors predict metabolomic functions of microbiota to propose butyrate and L-carnitine (e.f. PICRUST, <http://picrust.github.io/picrust/>)? This tool may provide a more comprehensive view of the gut-derived metabolites altered by the BCG vaccine.
- Figure 5. The timing of the microbiota analyses is not clear to me. Was it assessed 5 weeks after the vaccine? Since the training of ResAM is not there at Weeks 2 (Fig S1I-L) but appears at weeks 5, the time course of the gut dysbiosis would be essential to

describe.

- Figure 7A-F. It is stated that M. Tb is still dividing in vivo in vaccinated mice (line 180). So the results gained from the culture of ResAM with serum can not exclude a role for metabolites (or other mediators) released by other tissue. This experiment may be reproduced in germ-free mice to rule out this possibility.

- Figure 7G. Does the treatment with butyrate and L-carnitine increase M. Tb control in the lungs?

G. References: appropriate credit to previous work?

No comment.

H. Clarity and context: lucidity of abstract/summary, appropriateness of abstract, introduction, and conclusions

The manuscript is well written, and the abstract is appropriate. In the discussion, the role of respiratory microbiota in the training of ResAM is not discussed, yet respiratory dysbiosis induced by BCG vaccine is potentially also involved here. (DOI 10.1164/rccm.201907-1487oc, DOI 10.1164/rccm.201711-2180oc)

However, I have raised some comments and limits that, in my opinion, do not enable definitive conclusions to be reached. The proposed experiments may enable the team to validate their conclusions definitively.

Author Rebuttal to Initial comments

Nat Immunol NI-A32027A

We're very pleased of the favorable opinions of our study from both the outside expert reviewers and Editor. We appreciate the constructive comments from the reviewers and Editor.

Following a carefully crafted revision plan that was initially shared and approved with Editor, we have now remarkably improved both the scientific quality and presentation of the manuscript via considering all of the comments from both reviewers and Editor, performing a number of new experiments wherever appropriate and feasible, and providing the wealth of new data and information which are summarized as follows:

- 1) We've now provided the new data that following s.c. BCG vaccination, mycobacterial bacilli disseminated to the peritoneal macrophages and gut-associated lymphoid tissues, thus offering a mechanistic link between distal subcutaneous BCG vaccination and alterations in the gut;
- 2) We've now provided the data to reveal the time-dependent alterations in gut microbiome, inflammation and metabolome, and lung metabolome, and subsequent induction of memory airway macrophages and trained innate immunity in the lung;

- 3) We've now offered the evidence that s.c. BCG vaccination-induced mild gut inflammation/colitis is self-limited (such changes resolved by 8-weeks), and this offers a potential explanation why it has not been reported in humans following s.c. BCG vaccination;
- 3) We've now revealed that this gut-lung axis is threaded primarily via a microbial metabolite linkage between the gut, blood and lung;
- 4) We've now demonstrated that systemic supplementation of the key BCG vaccination-altered metabolites alone could induce the trained phenotype of airway-resident macrophages including their transcriptomic profile and anti-microbial trained innate immunity, resembling that induced by s.c. BCG vaccination;
- 5) We've also now provided the evidence that it's mainly the microbial metabolites, but not microbial-derived molecular patterns (MDMPs)/structural molecules, that play a major role in induction of trained innate immunity in lung macrophages;
- 6) We've provided further evidence to indicate the independence of circulating monocytes for induction of airway-resident memory macrophages and trained *anti-Mtb* innate immunity in early stages of pulmonary *Mtb* infection;
- 7) Based on the new mechanistic data now provided, we've revised the previous Graphical Abstract and included it (Supplementary Figure 10) in Discussion/Supplementary Figure package; (also attached at the end of Point-by-Point)
- 8) We've also made all necessary editorial changes accordingly.

We hope both Reviewers and Editor now find our manuscript to be satisfactory for its acceptance for publication.

Point by Point Response

Reviewer #2: (all figures/panels are bold-faced in text below)

This innovative study has demonstrated induction of trained immunity is both mediated at central level in the bone marrow, as well as in the periphery at the level of mucosal macrophages following s.c. BCG vaccination. Moreover, dissected a new mechanism through which a gut-lung axis is mediated by metabolites released by changes in microbiome induced by BCG vaccination. The experiments are well performed, and the manuscript is well written.

1. *The authors present extensive arguments that bone marrow-derived trained monocytes and T-cells/IFN γ are not needed for the induction of the alveolar macrophages (AMs) with a trained immunity phenotype. Are trained monocytes and T-cells/IFN γ however important for the protection given by BCG vaccination? A recent study by Kaufmann et al showed an important role of BCG training at bone marrow level: what is the relative role of trained myelopoiesis and trained AMs for the protection induced by BCG vaccination?*

Response: The role of T cells/IFN γ in BCG-induced protection was already addressed in WT animals in Fig 3M/N. Depletion of T cells and T cell-derived IFN γ in BCG hosts (BCG Δ T cells) did not compromise enhanced protection, it rather led to a further reduced *M.tb* CFU in the lung (Figure 3N). Nor did we find T cells or IFN γ to be required for induction of trained AMs (Figure 4O-R). On the other hand, depletion of T cells in un-vaccinated (PBS) host did not affect the *M.tb* CFU in the lung (4.27 ± 0.06 in PBS vs. 4.16 ± 0.08 in PBS Δ T cells)(this **new data** is now described on **pg. 11 bottom**).

To address the relative role of trained monocytes and trained AMs in BCG-induced protection, we set up new experiments using CCR2 knock-out mice that lack classical Ly6C^{hi} monocytes, a major peripheral blood source of monocyte-derived macrophages upon infection. Mice were BCG-vaccinated or treated with PBS, and were infected with *Mtb* at 5 weeks post-immunization. At day 3 post-infection mice were sacrificed and bacterial burden in cell fraction of bronchoalveolar lavage (airways) and lung tissue were assessed for *Mtb* CFU (**Figure 3O**). Despite the absence of Ly6C^{hi} monocytes, BCG-vaccinated CCR2KO mice still significantly better controlled the infection in the airways (**Figure 3P**) and lung tissue (**Figure 3Q**) than the un-vaccinated CCR2KO counterparts. This indicates infection is controlled starting as early as from day 3 post-infection in BCGvaccinated hosts independent of circulating monocytes. Our findings are in line with the previous report that monocyte recruitment to lung upon *Mtb* infection in BCG-vaccinated host does not ensue until d10 post-infection (Delahaye JL et al, *J. Immunol.* 203, 807–812, 2019). These new data are now provided on **pg. 12 top**.

2. *BCG vaccination induces also release of microbial structures in the circulation, such as peptidoglycans and MDP that can induce trained immunity at distant sites. Could this release play a role in the induction of trained immunity in AMs?*

(Note: this comment is similar to comments #5 & #6 below about the role of other microbial ligands such as LPS, besides microbial metabolites, potentially translocated from the leaky gut to blood, resulting from BCG vaccination, in induction of trained AMs. Hence, comments #2/5/6 are addressed together below).

Response: We appreciate these comments. In response, we evaluated whether the gut epithelial dysplasia and disruption resulting from s.c BCG vaccination led to leakage to the circulation of gut microbial-associated molecular patterns (MAMPs). We specifically measured serum levels of LPS, a

TLR4 ligand, by ELISA, as a representative gut microbial structural molecule. LPS levels in PBS control (Ctl) and BCG host did not differ and were all below 0.01EU/ml (**Supplementary Figure 8G**)(pg.18).

To further investigate the role of potentially leaked MAMPs in BCG-vaccinated hosts, we pursued new experiments where AMs from Toll-like receptor (TLR)2, TLR4 and NOD2-deficient mice were trained *ex vivo* with sera either from BCG or PBS control hosts using our well-established *ex vivo* AM training model (Figure 7A). Since MAMPs peptidoglycan, LPS and muramyl dipeptide (MDP) are ligands for TLR2, TLR4, and NOD2, respectively, we would expect impaired training of AM deficient in the cognate TLR if the corresponding MAMP present in serum played a critical role. We examined MHC II expression on AMs as an innate immunity training marker. Similar to the data in Figure 7C, BCG-conditioned serum induced significantly increased MHC II on wild type (WT) AMs (**Figure 7Q**). Of note, BCG-conditioned serum also induced significantly increased MHCII expression on un-stimulated TLR2 or NOD2-deficient AMs (**Figure 7R/T**). Stimulation with whole cell lysate of *M.tb* enhanced MHCII expression further on NOD2-deficient AMs, particularly on those trained by BCGconditioned serum (**Figure 7Q/T**). Absence of further increased MHCII expression in stimulated AMs from TLR2-deficient host suggests that *M.tb* cell lysate primarily stimulates AMs via TLR2 signaling (**Figure 7R**). Interestingly, TLR4- deficient AMs expressed considerably reduced MHCII expression and did not differ between control and BCG serum before stimulation (**Figure 7S**). Though stimulation increased MHCII expression in these cells, expression level did not differ between control and BCG-conditioned serum, suggesting inherent requirement of TLR4 signaling for MHCII expression by AMs. Collectively these data suggest that even if present at heightened levels in blood, these MAMPs unlikely microbial metabolites, do not contribute significantly to induction of training in resident AMs following s.c BCG vaccination. These results are now fully presented on **pg.24 top**.

3. *The changes in cecum size, gut permeability, microbiome richness and composition by BCG vaccination is remarkable. What are the potential mechanisms through which distant BCG vaccination can induce such strong effects in the architecture of the gut? Is there BCG spread in the gut mucosa? (this may explain the delay needed for the observed effects)*

Response: The spread of BCG to gut mucosa was not apparent in our microbiota analysis based on 16s rRNA expression. However, this approach may not be specific/sensitive enough to identify tiny numbers of BCG bacilli among the natural gut flora of billions of bacteria. Thus, given our findings that besides lung AMs, s.c BCG vaccination had a global training effect on macrophage populations at other mucosal tissues such as peritoneal macrophages (Figure 1 O/P) and that it is now known that there exists a strong biologic connection between the peritoneal cavity, peritoneal macrophages, gut, and other tissue organs residing within the peritoneal cavity (Ref 56, 57), we performed new experiments to assess the potential BCG translocation both to the peritoneal cavity including peritoneal macrophages and gut-associated mesenteric lymph nodes by mycobacterial colony forming unit (CFU) assay at 2-wk post-BCG. Indeed, not only did we find the length of the colon to be significantly different between control and BCG hosts with the latter being shortened ~by 2 cm at this timepoint, suggesting mild colitis following vaccination (**Figure 5Q**), but importantly, small but significant BCG CFU were

detected in the mesenteric lymph nodes (MLN), cell-free peritoneal washes (PW), and particularly, total peritoneal macrophage fraction (PCL) (**Figure 5R**). CFU was further verified to be BCG mycobacteria by acid fast staining. In comparison, these tissue samples were completely sterile in unvaccinated control animals. These data establish a mechanistic linkage between distal s.c BCG vaccination and the marked alterations in the gut and microbiome, most likely resulting from BCG dissemination to the peritoneal cavity, biologically motile peritoneal macrophages, and intestine. These new observations are now described on **pg. 19 top**.

4. *The changes on gut histology and function by BCG vaccination may seem deleterious: increased gut permeability, irregular villi, epithelial disruption, etc. Such changes are also described in low-grade chronic inflammatory processes (e.g. HIV infection in humans) and are associated with cardio-vascular and metabolic complications. Would these changes induced by BCG be expected to lead to similar long-term deleterious effects?*

Response: This is a very relevant point. In response, we examined the histology of colon tissues at both 2 and 8wk post-BCG vaccination and compared it to that at 5-wk timepoint when the overt colitis was observed in BCG hosts. In keeping with limited changes in the gut microbiome at 2-wk post vaccination (**Supplementary Figure 7C-F**), gut histology remained of little change except the low degree of lymphocytic infiltration (**Figure 5G & Supplementary Figure 7H**). Contrast to 5-wk timepoint when mild but overt colitis was observed (Figure 5E/F), by 8-wk post-vaccination, however, such intestinal inflammation largely resolved and the gut architecture restored comparable to that in control hosts (**Figure 5G & Supplementary Figure 7H**). These findings indicate that parenteral BCG vaccination triggers a time-dependent but self-limited gut inflammatory response. These new data are now described on **pg. 16 & 17**.

5. *It is not only metabolites, but also microbial structures (LPS, bacterial DNA, peptidoglycans, etc) that can translocate from gut to blood in cases of increased gut permeability. For example, were there higher LPS circulation concentrations in mice vaccinated with BCG?*

Response: Please refer to the collective response to comment #2 above.

6. *Interestingly, also the mice colonized with microbiota from BCG-vaccinated animals displayed some of the epithelial dysplasia and disruption. This opens the possibility of leakage in the circulation of microbial products: both cellular structures (cell wall components-e.g. LPS, bacterial DNA) and metabolites. The same question here: is circulating LPS (or other bacterial structures) concentration changed?*

Response: Please refer to the collective response to comment #2 above.

7. *The authors elegantly show that a mix of L-carnitine and butyrate hydrochloride in drinking water of the animals, that maintain high SCFAs in the blood, also induced trained immunity in the AMs. Does this mix of L-carnitine and butyrate hydrochloride induce histological changes in the gut mucosa?*

Response: To address this, we pursued new experiments where sets of mice were treated with L-carnitine and butyrate for 3 weeks in drinking water or given regular drinking water. At 3rd week metabolite treatment was stopped and mice were given regular drinking water for a week. At this point all animals were sacrificed and the colon sections subject to histological analysis. Unlike in BCG-vaccinated hosts (Figure 5E/F & Supplementary Figure 7G), as expected the metabolite supplementation of naïve animals (DW+M) did not cause major architectural changes in the colon except a mild lymphocytic infiltration (**Supplementary Figure 9F**). This data is now described on **pg.22 bottom**.

8. *A number of recent studies (including in non-human primates) showed stronger effects of i.v. BCG compared to s.c. BCG for protection against M. tuberculosis. Can the authors speculate whether trained immunity in AMs would be stronger induced by i.v. BCG?*

Response: This is an interesting question. While i.v delivered BCG vaccine induced better protection in preclinical settings, it does not represent a clinically feasible route for mass vaccination. As we've now demonstrated in this study, s.c BCG vaccination-induced trained immunity in AMs involves mycobacterial translocation to gut-associated lymphoid tissues, peritoneal macrophages, and likely intestine, it's possible that i.v BCG inoculation will lead to even greater extent of and quicker BCG dissemination to these critical tissue sites, thus accelerating induction of AM training via a robustly established metabolomic gut-lung axis.

Reviewer #3: (all new data figures/panels are bold-faced in text below)

I wish to thank you for offering me the possibility to read and comment on this interesting study by Jeyanathan et al., which investigates the mechanisms of parenteral BCG vaccine to induce trained mucosal resident memory macrophages via the gut-lung axis.

1. *The experiments are all repeated and statistically significant. My main question here concerns the validity of the mice models. Indeed, the observation of gut dysbiosis in BCG-vaccinated mice is novel, but it is an intriguing result since a decrease in the alpha diversity (Fig 5A) and the histological finding (Fig 5E) demonstrated that BCG induced colitis in mice. This effect has not been described in humans, yet BCG has been used for decades in clinical practice to the best of my knowledge. I am thus concerned by the extrapolation to humans of the results gained in this model and the conclusion that BCG-induced colitis induces training (and not only BCG-vaccine).*

Response: This is an interesting and relevant point. Indeed, thus far there have not been any clinical reports on the colitis or gut inflammatory responses resulting from cutaneous BCG vaccination in humans. This could be due to a couple of reasons. **1)** this could have been overlooked or neglected. As we brought up and discussed (pg.26), only some, but not all, of cutaneous BCG-vaccinated humans had

enhanced innate immune protection in the lung and these individually never develop overt infection, let alone TB disease, following *M.tb* exposure. Furthermore, as we've pointed out in the manuscript, considering the effect of parenteral vaccination on gut and gut microbiota is a very infant or terribly under-investigated area of research and almost all of the research has clinically or preclinically been set up only in the reversed direction, i.e. studying the effect of changes in gut microbiota on the efficacy of parenteral vaccination. It's thus our hope that our current preclinical study will kindle clinical studies to investigate the role of parenteral vaccination in gut microbiome, permeability, and metabolome, and their effects on mucosal immunity. **2)** via the revision of our manuscript, we've now provided new data that parenteral BCG vaccination triggers a time-dependent but self-limited gut inflammatory response. Specifically, we now show that at 2-wk post-BCG vaccination, there was little inflammation in the gut (**Figure 5G & Supplementary Figure 7H**) (pgs.16 & 17). This is in contrast to 5-wk timepoint when mild but overt colitis was observed (Figure 5E-I & Supplementary Figure 7G-I)). However, by 8-wk post-vaccination, such intestinal inflammation largely resolved and the gut architecture restored to be comparable to that in control hosts (**Figure 5G & Supplementary Figure 7H**). These preclinical observations suggest that in humans (infants in most countries where parenteral BCG vaccine is routinely administered), it's likely the mild and transient nature of colitis could have been clinically insignificant/unnoticeable. Although two recently published studies (one preclinical and one clinical) have seen changes in gut microbiota following parenteral BCG vaccination (Ref 64/65), it's unclear whether there existed any degree of colitis, metabolomic changes, and altered immunity in the lung. The above considerations are now thoroughly discussed (**pg. 26**).

2. *To reinforce the message, the authors may consider demonstrating that the BCG vaccine-induced training of ResAM is altered in germ-free mice.*

Response: Unfortunately, given the nature of our project we do not believe the use of germ-free mice will provide the reliable information and allow us to adequately address the relationship of BCG vaccination-altered microbiota in the gut to the training of alveolar macrophages. This is because the lack of microbial signals/stimulation in germ-free mice leads to the pervasive impairments in the immune development, maturation and function including decreased numbers and functions of myeloid cells and lymphoid cells and defective cytokine signaling (Kennedy EA et al, *Front Physiol* 9:1534, 2018; Geuking MB et al, *Gut Microbes* 5:411-418, 2014). In other words, such mice are expected to develop aberrant and impaired innate and adaptive immune responses to BCG vaccination due to a multitude of mechanisms.

3. *In my opinion, two sets of experiments need to be performed to demonstrate the conclusion of the authors definitively:*

The main message of the manuscript is the training of resident AM. Since monocyte-derived AM can replace resident AM and gain most of their phenotype, relative contribution of circulating monocytes to the genesis of trained resident AM needs to be demonstrated. The only result demonstrating the impact of resAM is the CCR2^{-/-} mice. The demonstration that BCG-vaccine also increases MHC-II in CCR2^{-/-}

mice is encouraging, but it would be essential to reinforce this data, and notably to demonstrate that ResAM of CCR2^{-/-} mice receive the complete training. The authors may consider to

3.1. *Provide the total number of ResAM and Mo-AM in the several conditions (is the deletion of monocytederived AM changing the proliferation of ResAM?).*

Response: In response, we've now provided the data on both frequencies and absolute numbers of major macrophage subsets (AM, MDM/Mo-AM, IM) in the airway and lung tissue at both 2-wk (**Supplementary Figure 6A/B**) and 5-wk (**Figure 4G/H**) timepoints. These data clearly show that there was a lack of monocytederived macrophages (MDM/Mo-AM) in the airway of BCG-vaccinated animals either at 2- or 5-wk postvaccination, and although as expected, MDM are present in lung tissue but BCG vaccination did not increase this subset of macrophages in lung tissue. The lack of contribution of MDM to trained tissue-resident AM was supported further by our findings from CCR2KO and PKH models (**Figure 4K/L**) and the AM vs. Mo lineage gene signature analysis (**Figure 4M/N**). These observations are in keeping with the data from a previous report, also showing a lack of MDM recruitment to the lung following s.c. BCG vaccination (Delahaye, JL et al, *J. Immunol.* 203, 807–812, 2019). These new details are now described on **pgs.13 & 14**.

3.2. *Demonstrate that ResAM in CCR2^{-/-} also has cytokine and metabolic reprogramming.*

Response: We appreciate this point. In response, we carried out new experiments to provide evidence to support the trained phenotype of ResAM in BCG-vaccinated CCR2^{-/-} hosts. To demonstrate trained phenotype in the ResAM of CCR2^{-/-} (CCR2KO) mice, BCG-vaccinated and PBS-treated CCR2KO mice were scarified at 5-wks and airway BAL AM and lung mononuclear cells were obtained and cultured in the presence or absence of *M.tb* WCL antigens and subjected to MHCII surface immunostaining, cytokine release assay (BAL AM) and/or intracellular cytokine staining (lung AM). Indeed, even in the absence of the major circulating monocytes, compared to the control, MHC II expression remained elevated in both airway and lung tissue ResAM of BCGvaccinated animals with and without stimulation, which was more significant with the latter (**Figure 4I/J**). Furthermore, upon stimulation, compared to the control counterpart, airway AM from BCG hosts produced significantly higher levels of IL-6 and TNF α (**Figure 4I**). Like the cytokine profile in ResAM of BCG-vaccinated wild type B6 mice (**Figure 1D/E**), there were significantly higher frequencies of stimulated lung AM of BCGvaccinated CCR2KO hosts producing IL-6 while the levels of TNF- α -producing cells were comparable to those in control animals (**Figure 4J**). The trained phenotype/immunity of ResAM from BCG CCR2KO animals was further supported by the new functional data that airway ResAM isolated from BCG CCR2KO animals, upon ex vivo infection with *M.tb*, demonstrated significantly increased mycobacterial killing (**Supplementary Figure 6C**). These data together demonstrate the induction of a trained phenotype of ResAM in vaccinated CCR2KO hosts, independent of circulating monocytes and they are now described on **pg.13 bottom & pg.14 top**. This conclusion further corroborates with the new data described below showing enhanced in vivo protection against pulmonary *M.tb* infection in the lung of parenteral BCG-vaccinated CCR2KO mice (**Figure 3O-Q**).

3.3. Demonstrate that CCR2^{-/-} mice are still protected against *M. Tb* after BCG vaccination

Response: Again, this is a very relevant question shared in part with Comment #1 from Reviewer#2 above. Thus, to address the relative role of trained monocytes and trained AM in BCG-induced protection, we performed new experiments using CCR2 knock-out mice that lack classical Ly6C^{hi} monocytes, a major peripheral blood source of monocyte-derived macrophages upon infection. Mice were BCG-vaccinated or treated with PBS, and were infected with *Mtb* at 5 weeks post-immunization. At day 3 post-infection mice were sacrificed and bacterial burden in cell fraction of bronchoalveolar lavage (airways) and lung tissue were assessed for *Mtb* CFU (**Figure 3O**). Despite the absence of Ly6C^{hi} monocytes, BCG-vaccinated CCR2KO mice still significantly better controlled the infection in the airway AM (**Figure 3P**) and lung tissue (**Figure 3Q**) than the un-vaccinated CCR2KO counterparts. This indicates infection is better controlled starting as early as from day 3 post-infection in BCG-vaccinated hosts independent of circulating monocytes. Our findings are in line with the previous report that monocyte recruitment to lung upon *Mtb* infection in BCG-vaccinated host does not occur markedly until d10 post-infection (Delahaye JL et al, *J. Immunol.* 203, 807–812, 2019). These new data are now provided on **pgs. 12 top**.

4. The second main message is the role of metabolites derived from gut microbiota in the training of ResAM. The specific role of the proposed metabolites (butyrate and L-carnitine) reported is not specific to BCG, and the reported effects of treatment with these metabolic (pro-inflammatory IL6 / MIP1a production) are relatively non-specific. The increased concentration of metabolites is only described in the cecum. To reinforce this message, and respond to the question of the specificity of BCG vaccine, the authors may consider to

4.1. Compare the transcriptomic activity of ResAM in mice treated for three weeks with butyrate and L-carnitine, with resAM of BCG vaccinated mice. This would reinforce the message that these metabolites are specific and play an essential role in the BCG-induced training of AM

Response: In response to this suggestion, we set up new experiments and isolated AM from animals supplemented with butyrate and L-carnitine metabolites (DW+M) and from control animals given regular drinking water (DW) according to experimental schema (Figure 7K). AM were subjected to transcriptional analysis with (S) and without (US) stimulation. We found that each of the experimental groups was clustered into its own pattern, suggesting the transcriptional alternation in AM following metabolite treatment (**Figure 7N**). Subsequently, a total of 265 genes were found differentially expressed (DE) in AM from metabolite-treated animals compared to DW controls (**Supplementary Figure 9G/H**). Upon comparison with BCG vaccination, we found that like AM from BCG hosts (Figure 2C & Supplementary Figure 2D), the genes associated with cell differentiation and proliferation (*nov*, *hbegf*, *kitl*, *six5*) were also up-regulated in AM of DW+M hosts compared to controls (**Supplementary Figure 9G**). The gene, *snca*, a microphage/microglial activation gene required for inflammatory responses, was also up-regulated in AM from both BCG-vaccinated and DW+M animals (Figure 2D &

Supplementary Figure 9G). In keeping with their elevated cytokine and chemokine production upon restimulation, AM of DW+M animals up-regulated a number of genes including HLA genes that are involved in immune processes (**Supplementary Figure 9H**). Also similar to AM in BCG hosts (Figure 2E), the genes involved in antigen presentation and processing were also significantly increased particularly in stimulated AM of DW+M hosts (**Figure 7O**). Furthermore, the levels of the genes associated with fatty acid oxidation, glycolysis and mTOR pathway in AM of DW+M animals (**Supplementary Figure 9I-K**) were in general similar to those of AM of BCG hosts (Figure 2F & Supplementary Figure 2E/F). Of importance, the new data provided below shows significantly enhanced protection in early stages of pulmonary *M.tb* infection in the lung of Met-treated animals, even to a greater extent than that by BCG vaccination (**Figure 7P**). These findings together indicate significantly shared transcriptomic changes in trained AM between s.c BCG vaccinated and Met-treated animals, further supporting the critical role by altered circulating microbial metabolites in induction of trained mucosal tissue-resident macrophages and trained innate immunity (TII). These new data are now described on **pg. 22 bottom & pg. 23**.

4.2. *Compare the transcriptomic activity of trained ResAM of BCG vaccinated mice with their results gained in these cells after viral pneumonia (Yao et al. Cell 2018)*

Response: This is an interesting comment. In response, we used the data presented in the current study on differentially expressed genes (DEG) by AM in s.c. BCG-vaccinated mice vs. PBS counterparts and analyzed them against the DEG identified in AM in adenoviral-vectored vaccine immunized mice published in Yao Y et al. *Cell* 2018. We found about five times more genes (1,309 vs 248 genes) that were differentially expressed by resident AM from the mice vaccinated intranasally with adenovirus-vectored (Ad) vaccine (**Supplementary Figure 2G**) compared to AM from BCG-vaccinated hosts (Supplementary Figure 2B/C). However, both shared similar features in pre-defined gene sets related to antigen processing and presentation, glycolysis, mTOR pathway and fatty acid oxidation (**Supplementary Figure 2H**). The transcriptomic features distinguishing Ad AM from BCG AM were up-regulation of cell activation genes, *lck*, *bcl2*, *il7r* and *slamf7* and no enrichment for cell cycle and cell division associate genes. This transcriptomic dataset comparison thus suggests that although some features of TII in trained AM are shared following parenteral BCG organism-based vaccination and respiratory mucosal viral vector-based vaccination, the trained AM are unique depending on both the route of vaccination and the nature of vaccine. These new data are now described on **pg.8 bottom & pg.9 top**.

4.3. *Demonstrate that the concentrations of butyrate and L-carnitine are increased in the lungs after BCG, and provide the timing of this increase. Potentially (maybe not possible) demonstrate by in vivo tracking of tagged metabolites injected in the guts that their passage to the lungs is possible and increased.*

Response: We too very much appreciate this comment. In response, we pursued new experiments and collected lung tissue homogenates from 2- and 5-wk BCG-vaccinated mice and the control mice and subjected them to metabolomic analysis. PLS-DA model showed distinct clustering of lung tissue metabolites for controls only at 5-wk post-BCG vaccination while it overlapped at 2-wk post-BCG with controls (**Figure 5N**). Based on VIP scores (>1.5), 7 discriminating lung metabolites were rank-ordered (**Supplementary Figure 8H**). In keeping with increased deoxy carnitine levels in the cecum at 5-wk post-BCG vaccination (Figure 5K), significantly elevated levels of carnitine products, both butyryl carnitine and hexanoyl carnitine, were found in the lung tissue at 5-wk post-BCG (**Figure 5O/P**). These new data are now described on **pg.18**. These findings suggest a time-dependent association between the metabolite content of the gut and lung in BCG hosts which corroborates with the time-dependent alternations/induction in gut microbiome, gut inflammation, serum microbiome, and respiratory mucosal memory AM as well as trained innate immunity in the lung (Figures 1-7).

Minor propositions

- *Figure 3C and 3E. Is it possible to test the resistance to M. Tb of BCG-vaccinated mice after depletion of ResAM? Given the results in Fig 3H-J (drastic increase in CD4 T cells) and Fig 3N (positive effects of T cell depletion), it is possible to conclude that T cells are more critical than macrophages here.*

Response: Unfortunately, depletion of ResAM from the airway won't provide the meaningful outcome/information regarding the role of T cells in BCG-induced early anti-*Mtb* protection since ResAM are the primary target cell of *Mtb* bacilli within the airway (Cohen SB. *et al.*, *Cell Host Microbe* 24: 439-446, 2018), and furthermore, the interaction of Ag-specific T cells with infected macrophages represents one of the important mechanisms by which T cells control *Mtb* infection.

We've previously shown that BCG-activated T cells, when adoptively transferred to the airway of naive mice, enhance anti-*Mtb* protection (Horvath C *et al*, *Mucosal Immunol*, 5:420, 2012), supporting a protective role of T cells in the absence of trained ResAM. However, our current findings suggest a greater protective role of trained ResAM during early stages of infection and a regulatory role of T cells (in the presence of trained ResAM) since depletion of T cells led to further reduced *Mtb* infection (Fig 3N).

- *Figure 3N. The deletion of CD4 CD8 T cells increases the control of M.tb. Is this result also observed in PBStreated mice? This is an essential control since it suggests that CD4 CD8 T cells are deleterious after the BGG vaccine.*

Response: In response, we set up a new experiment similarly as in Figure 3N but only comparing 7-day *Mtb* infection levels in the lung between PBS-treated WT mice and PBS mice depleted of total T cells. We found that as expected, T cell depletion did not affect *Mtb* infection levels in the lung (4.27 ± 0.06 in PBS vs. 4.16 ± 0.08 Log₁₀ CFU in PBS Δ T cells), different from the BCG and BCG Δ T cells data shown in Figure 3N. This piece of new data is now described on **pg.11 bottom**.

- *Figure 5. Did the authors predict metabolomic functions of microbiota to propose butyrate and L-carnitine (e.f. PICRUST, <http://picrust.github.io/picrust/>)? This tool may provide a more comprehensive view of the gut-derived metabolites altered by the BCG vaccine.*

Response: In response, we have now described the differential functional prediction of gut microbiome of BCG host in **Supplementary Table 4**. A total of 19 predictive functional molecules significantly differed between BCG host and control (p-value <0.05) and categorized to energy metabolism, transporters, signaling and cellular processes and genetic information processing (**Supplementary Table 4**). Importantly, in keeping with elevated butyric acid levels in the cecum (Figure 5M), acetolactate synthase I/II/III large subunit involved in butonate (salts and esters of butyric acid) metabolism was significantly increased in BCG host. This piece of new data is now described on **pg.18**.

- *Figure 5. The timing of the microbiota analyses is not clear to me. Was it assessed 5 weeks after the vaccine? Since the training of ResAM is not there at Weeks 2 but appears at weeks 5, the time course of the gut dysbiosis would be essential to describe.*

Response: The previous gut dysbiosis data was from 5-wk timepoint. We agree that it's important to also provide the 2-wk data on gut microbiome and compare it with 5-wk. In response, we've performed the gut microbiome analysis at 2-wk post-BCG vaccination. Indeed, although as our new data have shown that BCG bacilli could be detected in gut-associated tissues at 2-wk timepoint (**Figure 5Q/R**), gut microbiome at 2-wk post-BCG did not change much, still to a large extent, resembling the gut microbiome of the control animals (**Supplementary Figure 7C-F**) regardless their partial overlap (**Supplementary Figure 7E**) and fourteen specific OTUs being differentially changed in the gut microbiota of 2-wk BCG hosts (**Supplementary Table 3**). Thus, these results together indicate that distal parenteral BCG vaccination-mediated gut dysbiosis occurs in a time-dependent manner and the most significant gut microbiota changes are not seen until 5-wk post-BCG (Figure 5A-J & Supplementary Figure 7A/B), which corroborate well with significant gut inflammation, metabolite changes both in the gut and lung, and induction of trained tissue-resident macrophages and trained innate immunity in the lung. These new data are now described on **pg. 16 bottom & pg.17 top**.

- *Figure 7A-F. It is stated that *M. tb* is still dividing in vivo in vaccinated mice (line 180). So the results gained from the culture of ResAM with serum cannot exclude a role for metabolites (or other mediators) released by other tissue. This experiment may be reproduced in germ-free mice to rule out this possibility.*

Response: As stated earlier, it is not technically possible to address this point since there are critical confounding issues associated with the lack of microbial signals/stimulation in germ-free mice, which leads to fundamental impairments in the immune development and maturation including decreased numbers and function of myeloid cells including tissue-resident macrophages and lymphoid cells and defective cytokine signaling (Kennedy EA et al, *Front Physiol* 9:1534, 2018; Geuking MB et al, *Gut Microbes* 5:411-418, 2014).

- *Figure 7G. Does the treatment with butyrate and L-carnitine increase *M. Tb* control in the lungs?*

Response: This is a good question. To address it, we set up new experiments and isolated AM from animals supplemented in drinking water with butyrate and L-carnitine metabolites (Met) and from control animals given regular drinking water (DW) according to experimental schema depicted in Figure 7K. At the end of 3rd week, metabolite treatment was stopped, and mice were given regular drinking water for a week. A set of 5-wk BCGvaccinated mice were also included as a comparison. At this point all animals were infected with *Mtb* and CFU in lung tissue was evaluated at 7 days post-infection. In keeping with the trained phenotype in their resident AM (**Figure 7K-O**), metabolites-treated animals showed significantly enhanced protection in the lung during early stages of pulmonary *M.tb* infection, even to a greater extent than that in BCG-vaccinated counterparts (**Figure 7P**). This new data is now described on **pg. 23 bottom**. These findings together indicate significantly shared phenotypic and transcriptomic changes in trained AM between s.c BCG vaccinated and Met-treated animals, further supporting the critical role by altered circulating microbial metabolites in induction of trained mucosal tissue-resident macrophages and trained innate immunity (TII).

Conceptualized Summary:

Decision Letter, first revision:

Subject: Your manuscript, NI-A32027B

Message: Our ref: NI-A32027B

2nd Sep 2022

Dear Dr. Xing,

Thank you for your patience as we've prepared the guidelines for final submission of your Nature Immunology manuscript, "Parenteral BCG vaccine induces respiratory mucosal-resident memory macrophages and trained innate immunity via the gut-lung axis" (NI-A32027B). Please carefully follow the step-by-step instructions provided in the attached file, and add a response in each row of the table to indicate the changes that you have made. Please also check and comment on any additional marked-up edits we have proposed within the text. Ensuring that each point is addressed will help to ensure that your revised manuscript can be swiftly handed over to our production team.

When you upload your final materials, please include a point-by-point response to any remaining reviewer comments and please make sure to upload your checklist.

If you have not done so already, please alert us to any related manuscripts from your group that are under consideration or in press at other journals, or are being written up for submission to other journals (see: <https://www.nature.com/nature-portfolio/editorial-policies/plagiarism#policy-on-duplicate-publication> for details).

In recognition of the time and expertise our reviewers provide to Nature Immunology's editorial process, we would like to formally acknowledge their contribution to the external peer review of your manuscript entitled "Parenteral BCG vaccine induces respiratory mucosal-resident memory macrophages and trained innate immunity via the gut-lung axis". For those reviewers who give their assent, we will be publishing their names alongside the published article.

Nature Immunology offers a Transparent Peer Review option for new original research manuscripts submitted after December 1st, 2019. As part of this initiative, we encourage our authors to support increased transparency into the peer review process by agreeing to have the reviewer comments, author rebuttal letters, and editorial decision letters published as a Supplementary item. When you submit your final files please clearly state in your cover letter whether or not you would like to participate in this initiative. Please note that failure to state your preference will result in delays in accepting your manuscript for publication.

Cover suggestions

As you prepare your final files we encourage you to consider whether you have any images or illustrations that may be appropriate for use on the cover of Nature Immunology.

Covers should be both aesthetically appealing and scientifically relevant, and should be supplied at the best quality available. Due to the prominence of these images, we do not

generally select images featuring faces, children, text, graphs, schematic drawings, or collages on our covers.

Nature Immunology has now transitioned to a unified Rights Collection system which will allow our Author Services team to quickly and easily collect the rights and permissions required to publish your work. Approximately 10 days after your paper is formally accepted, you will receive an email in providing you with a link to complete the grant of rights. If your paper is eligible for Open Access, our Author Services team will also be in touch regarding any additional information that may be required to arrange payment for your article.

Please note that *Nature Immunology* is a Transformative Journal (TJ). Authors may publish their research with us through the traditional subscription access route or make their paper immediately open access through payment of an article-processing charge (APC). Authors will not be required to make a final decision about access to their article until it has been accepted. [Find out more about Transformative Journals](https://www.springernature.com/gp/open-research/transformative-journals).

If you have any questions about costs, Open Access requirements, or our legal forms, please contact ASJournals@springernature.com.

Please use the following link for uploading these materials: [REDACTED]

Best regards,

Elle Morris
Senior Editorial Assistant
Nature Immunology
Phone: 212 726 9207
Fax: 212 696 9752
E-mail: immunology@us.nature.com

On behalf of

Nick Bernard, PhD
Senior Editor
Nature Immunology

Reviewer #2:

Remarks to the Author:

This a very interesting study and well written manuscript. The authors have provided adequate answers to my suggestions.

Reviewer #3:

Remarks to the Author:

A. Summary of the key results

Jeyanathan et al. investigated the mechanisms of the parenteral BCG vaccine to induce trained resident memory macrophages via the gut-lung axis. The main results are that after in vivo BCG vaccination, first, resident macrophages can be trained independently from recruited macrophages, second that gut microbiome-derived metabolites participate to the spatio-temporal adaptation of resident macrophages.

Most of the literature on trained immunity is focused on the epigenetic regulation of bone-marrow progenitors, while in vivo, resident macrophages are key to organ homeostasis. The current results fit with the recognition of the spatio-temporal adaptation of macrophages and bring new information by demonstrating that gut microbiome-derived metabolites regulate the anatomical niche. The results are thus original and significant. The methodology, statistics, and conclusions are sound and robust.

I wish to congratulate the authors for making a good job at revising the manuscript and providing a lot of new data supporting their conclusions. I have no more question for the authors.

Final Decision Letter:

Subject: Decision on Nature Immunology submission NI-A32027C

Message: In reply please quote: NI-A32027C

Dear Dr. Xing,

I am delighted to accept your manuscript entitled "Parenteral BCG vaccine induces lung-resident memory macrophages and trained immunity via the gut-lung axis" for publication in an upcoming issue of Nature Immunology.

Over the next few weeks, your paper will be copyedited to ensure that it conforms to Nature Immunology style. Once your paper is typeset, you will receive an email with a link to choose the appropriate publishing options for your paper and our Author Services team will be in touch regarding any additional information that may be required.

Please note that *Nature Immunology* is a Transformative Journal (TJ). Authors may publish their research with us through the traditional subscription access route or make their paper immediately open access through payment of an article-processing charge (APC). Authors will not be required to make a final decision about access to their article until it has been accepted. [Find out more about Transformative Journals](https://www.springernature.com/gp/open-research/transformative-journals).

Your paper will be published online soon after we receive your corrections and will appear in print in the next available issue. Content is published online weekly on Mondays and Thursdays, and the embargo is set at 16:00 London time (GMT)/11:00 am US Eastern time (EST) on the day of publication. Now is the time to inform your Public Relations or Press Office about your paper, as they might be interested in promoting its publication. This will allow them time to prepare an accurate and satisfactory press release. Include your manuscript tracking number (NI-A32027C) and the name of the journal, which they will need when they contact our office.

About one week before your paper is published online, we shall be distributing a press release to news organizations worldwide, which may very well include details of your work. We are happy for your institution or funding agency to prepare its own press release, but it must mention the embargo date and Nature Immunology. Our Press Office will contact you closer to the time of publication, but if you or your Press Office have any enquiries in the meantime, please contact press@nature.com.

Also, if you have any spectacular or outstanding figures or graphics associated with your manuscript - though not necessarily included with your submission - we'd be delighted to consider them as candidates for our cover. Simply send an electronic version (accompanied by a hard copy) to us with a possible cover caption enclosed.

Please note that we encourage the authors to self-archive their manuscript (the accepted version before copy editing) in their institutional repository, and in their funders' archives,

six months after publication. Nature Portfolio recognizes the efforts of funding bodies to increase access of the research they fund, and strongly encourages authors to participate in such efforts. For information about our editorial policy, including license agreement and author copyright, please visit www.nature.com/ni/about/ed_policies/index.html

Sincerely,

Nick Bernard, PhD
Senior Editor
Nature Immunology